# Separable Neural Networks: Approximation Theory, NTK Regime, and Preconditioned Gradient Descent

**Yisi Luo**[1], **Deyu Meng**[1*]
[1]Xi'an Jiaotong University
yisiluo1221@foxmail.com, dymeng@mail.xjtu.edu.cn

## Abstract

Separable neural networks (SepNNs) are emerging neural architectures that significantly reduce computational costs by factorizing a multivariate function into linear combinations of univariate functions, benefiting downstream applications such as implicit neural representations (INRs) and physics-informed neural networks (PINNs). However, fundamental theoretical analysis for SepNN, including detailed representation capacity and spectral bias characterization & alleviation, remains unexplored. This work makes three key contributions to theoretically understanding and improving SepNN. First, using Weierstrass-based approximation and universal approximation theory, we prove that SepNN can approximate any multivariate function with arbitrary precision, confirming its representation completeness. Second, we derive the neural tangent kernel (NTK) regimes for SepNN, showing that the NTK of infinite-width SepNN converges to a deterministic (or random) kernel under infinite (or fixed) decomposition rank, with corresponding convergence and spectral bias characterization. Third, we propose an efficient separable preconditioned gradient descent (SepPGD) for optimizing SepNN, which alleviates the spectral bias of SepNN by provably adjusting its NTK spectrum. The SepPGD enjoys an efficient $\mathcal{O}(nD)$ complexity for $n^D$ training samples, which is much more efficient than previous neural network PGD methods. Extensive experiments for kernel ridge regression, image and surface representation using INRs, and numerical PDEs using PINNs validate the efficiency of SepNN and the effectiveness of SepPGD for alleviating spectral bias.

## 1 Introduction

Separable neural networks (SepNNs) are a class of neural architectures that represent a multivariate function as a linear combination of univariate functions, each parameterized by a lightweight factor neural network (Liang et al., 2022; Cho et al., 2023; Luo et al., 2024). A key advantage of SepNNs lies in their ability to significantly reduce computational costs by reducing network propagations. The computational efficiency makes SepNN particularly advantageous and efficient in applications such as implicit neural representations (INRs) (Liang et al., 2022; Luo et al., 2024), physical-informed neural networks (PINNs) (Cho et al., 2023; Yu et al., 2024), and neural radiance fields (Chen et al., 2022). Compared to other efficient architectures for neural networks, SepNNs hold unique efficiency advantages. Especially, a classical line of work employs tensor decomposition to factorize the weights of networks (Liu & Parhi, 2023), thereby reducing the number of parameters. This efficient architecture is the most closely related to SepNNs. However, SepNNs are motivated by a fundamentally different idea. Rather than decomposing the network weights, the principle of SepNNs is to separate the input vector into multiple smaller inputs and process each input using factor networks. This design uniquely improves efficiency in scenarios involving coordinate-based neural networks and function evaluations on grids by reusing the factor outputs. Such a structure is particularly advantageous in applications like INRs (Liang et al., 2022), which map coordinates to pixel values, and PINNs (Cho et al., 2023), which map coordinates to physical

---

*Corresponding author. Code of SepPGD is in `https://github.com/YisiLuo/SepPGD`

fields. Moreover, SepNNs not only facilitate efficient training, but also offer improved interpretability and robustness by leveraging low-dimensional representations and interpretable factor modeling, demonstrating strong potential for scientific applications such as separable operator learning (Yu et al., 2024), radio map construction in wireless communication (Yuan et al., 2025), geophysical full waveform inversion in Earth science (Chen et al., 2025), and transcriptomics analysis in bioinformatics (Song et al., 2023). Therefore, the structure and efficiency of SepNNs hold important value across a variety of practical machine learning applications. We make more discussions on the efficiency advantage of SepNNs in Section A.1.

Formally, a $D$-variable SepNN $f_\Theta(x_1, \cdots, x_D)$ can be expressed as

$$f_\Theta(x_1, \cdots, x_D) = L(f_{\Theta_1}(x_1), \cdots, f_{\Theta_d}(x_D)) : \mathbb{R}^D \to \mathbb{R},$$

where $L$ denotes a type of linear combination that encodes the interactions between different univariate factor functions $\{f_{\Theta_d}(x_d)\}_{d=1}^D$ (the factor functions $\{f_{\Theta_d}(x_d)\}_{d=1}^D$ are parameterized by univariate neural networks, such as multi-layer perceptrons (MLPs)), and $\Theta = \{\Theta_1, \cdots, \Theta_D\}$ are learnable parameters. In the bivariate case $D = 2$, the inner product is a classical (if not only) choice for $L$ (see for example (Liang et al., 2022; Wang et al., 2025)):

$$f_\Theta(x_1, x_2) = f_{\Theta_1}(x_1)^\top f_{\Theta_2}(x_2) : \mathbb{R}^2 \to \mathbb{R}, \tag{1}$$

where $f_{\Theta_1}(x_1), f_{\Theta_2}(x_2) : \mathbb{R} \to \mathbb{R}^R$ are factor functions mapping separated inputs to $R$-dimensional latent vectors, and $R$ serves as a "rank" parameter that determines the representation capacity of the SepNN. In the multivariate case where $D > 2$, multiple alternatives exist for defining the linear combination $L$. A natural and widely-used option is the tensor canonical parafac (CP) decomposition (Kargas & Sidiropoulos, 2021), which will be the primary focus of this study:

$$\text{(CP)} \quad f_\Theta(x_1, \cdots, x_D) = \sum_{r=1}^R (f_{\Theta_1}(x_1))_r (f_{\Theta_2}(x_2))_r \cdots (f_{\Theta_D}(x_D))_r : \mathbb{R}^D \to \mathbb{R}, \tag{2}$$

which computes the inner product between $D$ factor functions $\{f_{\Theta_d}(x_d) : \mathbb{R} \to \mathbb{R}^R\}_{d=1}^D$, and $(f_{\Theta_d}(x_d))_r$ denotes the $r$-th component of $f_{\Theta_d}(x_d)$. The CP SepNN (2) degenerates into (1) when $D = 2$. In addition, other stochastic tensor decomposition formulations can be considered to construct the SepNN. For instance, the tensor-train (TT) decomposition represents multivariate functions by a sequence of lower-dimensional tensor functions (Gorodetsky et al., 2019; Zhou et al., 2025), and the tensor Tucker decomposition introduces an additional core tensor $\mathcal{C}$ to encode weighted multilinear relationships into SepNNs (Luo et al., 2024):

$$\text{(TT)} \quad f_\Theta(x_1, \cdots, x_D) = \sum_{r_1=1}^{R_1} \sum_{r_2=1}^{R_2} \cdots \sum_{r_{D-1}=1}^{R_{D-1}} (f_{\Theta_1}(x_1))_{1,r_1} (f_{\Theta_2}(x_2))_{r_1,r_2} \cdots (f_{\Theta_D}(x_D))_{r_{D-1},1}, \tag{3}$$

$$\text{(Tucker)} \quad f_\Theta(x_1, \cdots, x_D) = \mathcal{C} \times_1 f_{\Theta_1}(x_1) \times_2 \cdots \times_D f_{\Theta_D}(x_D) : \mathbb{R}^D \to \mathbb{R}.$$

Here, $\mathcal{C} \in \mathbb{R}^{R_1 \times \cdots \times R_D}$ denotes the core tensor in the Tucker decomposition model, $(R_1, \cdots, R_D)$ denotes the TT or Tucker rank of the model, $\times_d : \mathbb{R}^{R_1 \times \cdots \times R_D} \times \mathbb{R}^{R_d} \to \mathbb{R}^{R_1 \times \cdots \times R_{d-1} \times R_{d+1} \times \cdots \times R_D}$ denotes the tensor product between a tensor and a vector, i.e., $\mathcal{C} \times_d f_{\Theta_d}(x_d) := \texttt{fold}_d(\texttt{unfold}_d(\mathcal{C}) f_{\Theta_d}(x_d))$, where $f_{\Theta_d}(x_d) \in \mathbb{R}^{R_d}$ is the factor output in the Tucker model, $\texttt{unfold}_d : \mathbb{R}^{R_1 \times \cdots \times R_D} \to \mathbb{R}^{\prod_{d' \neq d} R_{d'} \times R_d}$ denotes the unfolding operator from a tensor to a matrix, and $\texttt{fold}_d : \mathbb{R}^{\prod_{d' \neq d} R_{d'}} \to \mathbb{R}^{R_1 \times \cdots \times R_{d-1} \times R_{d+1} \times \cdots \times R_D}$ denotes the folding operator from a vector to a tensor. The $(f_{\theta_d}(x_d))_{r_{d-1},r_d}$ denotes the $(r_{d-1}, r_d)$-th output component of the matrix-valued univariate function $f_{\theta_d}(x_d) : \mathbb{R} \to \mathbb{R}^{R_{d-1} \times R_d}$ in the TT model. The aforementioned linear combinations $L$ can be viewed as specific instances within the generalized linear Einstein summation for multi-dimensional array (Ahlander, 2002). Furthermore, nonlinear combinations (e.g., nonlinear activations (Li et al., 2025b)) can also be considered to enhance the stochasticity among factors.

The efficiency advantage of SepNNs comes from its separability nature. Especially, when training on a $D$-dimensional grid tensor with each dimension containing $n$ training samples (for instance, with training inputs of the form $\{(x_1, \cdots, x_D) \mid x_d = 1, 2, \cdots, n, \, d = 1, \cdots, D\}$, which contains $n^D$ training samples), the SepNN only needs to query $nD$ times of inputs by querying the separated input $x_d$ in each dimension via factor functions and then combining the outputs through the linear combination $L$. Hence, the computational complexity of a SepNN scales as $O(nD)$ in an epoch, compared to $O(n^D)$ for a conventional neural network (Cho et al., 2023; Luo et al., 2024). This

efficiency superiority makes SepNNs advantageous in downstream applications such as INRs (Liang et al., 2022; Luo et al., 2024) and PINNs (Cho et al., 2023; Vemuri et al., 2025).

While SepNNs have enabled a variety of promising applications, their theoretical foundations remain relatively underdeveloped, hindering a deeper understanding of their representation capacity and optimization behaviors. In this work, we seek to address the following fundamental questions regarding the theoretical aspects of SepNNs: 1) Do SepNNs possess sufficient representation capacity to approximate any continuous multivariate function in Euclidean space? 2) How can we characterize the training dynamics of SepNNs and identify any inherent spectral bias during optimization? 3) How can the spectral bias of SepNNs be mitigated to further enhance training efficiency? To address these fundamental questions, this work makes the following contributions:

- Using a novel combination of Weierstrass-based approximation and universal approximation theories, we rigorously establish an approximation theorem that SepNNs possess the capacity to approximate any continuous multivariate function with arbitrary precision, thereby confirming their representation completeness.

- We derive the neural tangent kernel (NTK) regimes for SepNNs under different asymptotic conditions. The SepNN's NTK converges to a deterministic kernel under infinite width and infinite rank, and converges to a random kernel under infinite width and fixed rank, providing new insights into the spectral bias characterization and training behavior of SepNNs.

- We further propose a scalable separable preconditioned gradient descent (SepPGD) method that provably adjusts the eigenvalue distribution of NTK matrix, effectively alleviating spectral bias of SepNNs. The SepPGD achieves a significantly lower computational complexity of $\mathcal{O}(nD)$ for $n^D$ training samples, which is much more efficient than existing neural network preconditioning methods. Extensive experiments across various downstream tasks including kernel ridge regression, image & surface representation using INRs, and PINNs demonstrate the improved efficiency and effectiveness of our SepPGD approach for alleviating spectral bias of SepNNs.

## 2 APPROXIMATION THEORY OF SEPNN

**Approximation Theory.** For any continuous multivariate function $f(x_1, \cdots, x_D)$ defined on a compact set, it is well-established that standard neural networks such as MLPs with suitable activation functions can approximate it to arbitrary accuracy (Leshno et al., 1993; Pinkus, 1999)—a result known as the universal approximation theorem. However, such a universal approximation theory remains lacking for SepNN structures. To fill this blank, our first contribution is to establish a universal approximation theorem of SepNNs (including CP, TT, and Tucker (2)-(3)) as follows.

**Theorem 1** (Universal approximation theorem of multivariate SepNNs). *Let $\mathcal{X}_1 \subset \mathbb{R}, \mathcal{X}_2 \subset \mathbb{R}, \cdots, \mathcal{X}_D \subset \mathbb{R}$ and $\mathcal{X} = \mathcal{X}_1 \times \mathcal{X}_2 \times \cdots \times \mathcal{X}_D \subset \mathbb{R}^D$ be any compact sets, and let $f(\boldsymbol{x}) : \mathcal{X} \to \mathbb{R}$ be a continuous multivariate function where $\boldsymbol{x} := (x_1, \cdots, x_D)$. For any $\epsilon > 0$, the following statements hold, in which all MLPs are drawn in the set $\{\boldsymbol{W}_2\sigma(\boldsymbol{W}_1 x + \boldsymbol{b}) : \boldsymbol{W}_2 \in \mathbb{R}^{\tilde{W} \times W}, \boldsymbol{W}_1 \in \mathbb{R}^{W \times 1}, \boldsymbol{b} \in \mathbb{R}^W, \tilde{W}, W \in \mathbb{N}_+\}$ with $\sigma$ a non-polynomial function.*

*(CP) There exist rank $R \in \mathbb{N}_+$ and univariate MLPs $f_{\Theta_d} : \mathcal{X}_d \to \mathbb{R}^R$ ($d = 1, \cdots, D$) such that*

$$\sup_{\boldsymbol{x} \in \mathcal{X}} \left| f(\boldsymbol{x}) - \sum_{r=1}^{R} (f_{\Theta_1}(x_1))_r (f_{\Theta_2}(x_2))_r \cdots (f_{\Theta_D}(x_D))_r \right| < \epsilon.$$

*(TT) There exist ranks $R_1, \cdots, R_{D-1} \in \mathbb{N}_+$ and univariate MLPs $f_{\Theta_d} : \mathcal{X}_d \to \mathbb{R}^{R_{d-1}R_d}$ ($d = 1, \cdots, D, R_0 = R_D = 1$) such that*

$$\sup_{\boldsymbol{x} \in \mathcal{X}} \left| f(\boldsymbol{x}) - \sum_{r_1=1}^{R_1} \sum_{r_2=1}^{R_2} \cdots \sum_{r_{D-1}=1}^{R_{D-1}} (f_{\Theta_1}(x_1))_{1,r_1} (f_{\Theta_2}(x_2))_{r_1,r_2} \cdots (f_{\Theta_D}(x_D))_{r_{D-1},1} \right| < \epsilon.$$

*(Tucker) There exist ranks $R_1, \cdots, R_D \in \mathbb{N}_+$, univariate MLPs $f_{\Theta_d} : \mathcal{X}_d \to \mathbb{R}^{R_d}$ ($d = 1, \cdots, D$), and a core tensor $\mathcal{C} \in \mathbb{R}^{R_1 \times \cdots \times R_D}$ such that*

$$\sup_{\boldsymbol{x} \in \mathcal{X}} \left| f(\boldsymbol{x}) - \mathcal{C} \times_1 f_{\Theta_1}(x_1) \times_2 \cdots \times_D f_{\Theta_D}(x_D) \right| < \epsilon,$$

*where $\times_d : (\mathbb{R}^{R_1 \times \cdots \times R_D} \times \mathbb{R}^{R_d}) \to \mathbb{R}^{R_1 \times \cdots \times R_{d-1} \times R_{d+1} \times \cdots \times R_D}$ denotes the mode-d specific product between a tensor and a vector.*

The theorem states that any continuous multivariate function on compact sets can be well approximated by either the CP, TT, or Tucker SepNNs. The detailed proof, which is based on the combination of Stone-Weierstrass theorem (Fedorova, 2002) and universal approximation theorem (Leshno et al., 1993), is placed in Appendix Section A.5. We go through the proof sketch as follows.

First, taking the CP SepNN (2) as an example, we consider the associated separable function class:

$$\mathcal{A} = \left\{ g : \mathcal{X} \to \mathbb{R} : g(x_1, \cdots, x_N) = \sum_{r=1}^{R} (g_1(x_1))_r (g_2(x_2))_r \cdots (g_D(x_D))_r, R \in \mathbb{N}, (g_d(x_d))_r \in C(\mathcal{X}_d) \right\}.$$

The function class $\mathcal{A}$ consists of all separable functions that can be expressed in the CP form, using the linear combination of vector-valued continuous univariate functions $g_d(x_d) : \mathcal{X}_d \to \mathbb{R}^R$ defined on compact sets $\mathcal{X}_d$. By slightly extending the classical universal approximation theorem (Leshno et al., 1993) to vector-valued functions, we show that each $g_d(x_d)$ can be approximated by an MLP with non-polynomial activation functions. Hence, for any function in $\mathcal{A}$, there exists a SepNN that approximates it up to arbitrary precision—a fact that can be formalized by bounding the total approximation error via the Cauchy-Schwarz inequality across the errors of the factor MLPs.

It therefore remains to show that $\mathcal{A}$ is dense in $C(\mathcal{X})$, the space of continuous functions over $\mathcal{X}$. This would imply that any continuous multivariate function can be approximated arbitrarily well by an element of $\mathcal{A}$, and consequently by a SepNN. To establish the density of $\mathcal{A}$, we leverage the Stone-Weierstrass theorem (Fedorova, 2002), which asserts that a function class defined on a compact set $\mathcal{X}$ is dense in $C(\mathcal{X})$ if it: (1) contains the identity function; (2) separates points in $\mathcal{X}$ (i.e., for any $\boldsymbol{a} \neq \boldsymbol{b}$ in $\mathcal{X}$, there exists $g \in \mathcal{A}$ such that $g(\boldsymbol{a}) \neq g(\boldsymbol{b})$); and (3) is closed under algebraic operations. We carefully examine that $\mathcal{A}$ meets these requirements. By combining this with standard universal approximation results, we conclude with the universal approximation capacity of SepNNs.

**Related Work.** (Cho et al., 2023) provided the approximation theory for the bivariate SepNN (1). Their proof is based on the orthogonal basis functions construction for the tensor product function space. Compared to this prior art, our approximation theory extends to any multivariate function approximation with $D \geq 2$ and more types of SepNNs such as TT and Tucker, and includes the result in (Cho et al., 2023) as a special case when $D = 2$ for CP SepNN (2). Furthermore, our proof offers a simpler alternative against (Cho et al., 2023) for the $D = 2$ case of CP SepNN. Another work (Yu et al., 2024) deduced the approximation theory for a separable physical-informed operator network, which is related to the CP SepNN (2). Their proof is based on the separability of the sine activation function using trigonometric angle addition and universal approximation theory. Compared to this work, we characterize a more general class of SepNNs using any non-polynomial activation functions and deduce the approximation error more systematically for various types of SepNNs. Our proof technique is unified and broadly applicable to these structures, such as separable operator network. The SepNN is also closely related to the Kolmogorov-Arnold network (KAN) (Liu et al., 2025), which represents learnable neurons as weighted summations of independent spline bases and serves as a fundamental architecture in scientific applications (Li et al., 2025a). It is a promising direction to extend the suggested proof technique to KAN approximation and its variants.

## 3 NTK REGIMES AND SPECTRAL BIAS CHARACTERIZATION

**NTK Regimes.** While the SepNN is demonstrated to be a universal approximator for multivariate function representation, its training dynamics remain poorly understood even for classical learning problems with gradient descent. We borrow the theoretical tools of NTK (Jacot et al., 2018; Arora et al., 2019a) and aim to characterize the training process of SepNNs via associated kernel regression. Our main contributions are as follows. First, under the asymptotic regime of infinite network width and infinite rank, we prove that the NTK of a SepNN—which can be expressed as the summation over factor MLPs' NTKs (Lemma 1)—converges to a deterministic kernel (Theorem 2), analogous to known results for standard MLPs (Jacot et al., 2018). This result allows us to char-

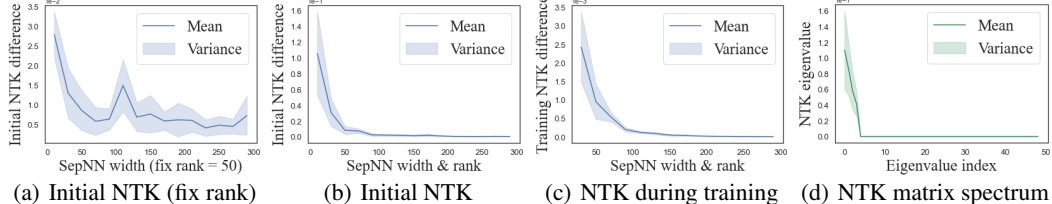

Figure 1: SepNN NTK verification using ten runs over multiple random seeds for square loss optimization using gradient descent. (a) The difference between NTK matrices in adjacent random seeds at initialization vs. SepNN width $W$ (under fixed rank $R = 50$). (b) The difference between NTK matrices in adjacent random seeds at initialization vs. SepNN width $W$ and rank $R$ (we set $W = R$). (c) The difference between NTK matrices at 100 training steps and at initialization over ten random seeds. (d) The NTK matrix eigenvalue distribution at initialization over ten random seeds.

acterize the convergence rate of wide SepNNs and to identify spectral bias related to the eigenvalue distribution of the NTK matrix[1].

**Lemma 1** (NTK of CP SepNN). *Let the multivariate CP SepNN be defined as $f_\Theta(x_1, \cdots, x_D) = \frac{1}{\sqrt{R}} \sum_{r=1}^{R} (f_{\Theta_1}(x_1))_r (f_{\Theta_2}(x_2))_r \cdots (f_{\Theta_D}(x_D))_r : \mathbb{R}^D \to \mathbb{R}$, where each $f_{\Theta_d} : \mathbb{R} \to \mathbb{R}^R$ is a parametric MLP with parameters $\Theta_d$, and $\Theta = (\Theta_1, \cdots, \Theta_D)$ is the collection of all parameters. Then the NTK of $f_\Theta$, defined as $K_\Theta(\boldsymbol{x}, \boldsymbol{x}') := \langle \nabla_\Theta f_\Theta(\boldsymbol{x}), \nabla_\Theta f_\Theta(\boldsymbol{x}') \rangle$ for $\boldsymbol{x} = (x_1, \cdots, x_D)$ and $\boldsymbol{x}' = (x_1', \cdots, x_D')$, is given by*

$$K_\Theta(\boldsymbol{x}, \boldsymbol{x}') = \frac{1}{R} \sum_{d=1}^{D} \boldsymbol{a}_d(\boldsymbol{x})^\top \boldsymbol{K}_{\Theta_d}(x_d, x_d') \boldsymbol{a}_d(\boldsymbol{x}'), \tag{4}$$

*where $\boldsymbol{K}_{\Theta_d}(x_d, x_d') \in \mathbb{R}^{R \times R}$ is the NTK matrix of the d-th factor MLP $f_{\Theta_d}$ with elements $(\boldsymbol{K}_{\Theta_d}(x_d, x_d'))_{r,s} = \langle \nabla_{\Theta_d} (f_{\Theta_d}(x_d))_r, \nabla_{\Theta_d} (f_{\Theta_d}(x_d'))_s \rangle$, and $\boldsymbol{a}_d(\boldsymbol{x})$ is a vector defined by $\boldsymbol{a}_d(\boldsymbol{x}) = \left( \prod_{d' \neq d} (f_{\Theta_{d'}}(x_{d'}))_1, \prod_{d' \neq d} (f_{\Theta_{d'}}(x_{d'}))_2, \cdots, \prod_{d' \neq d} (f_{\Theta_{d'}}(x_{d'}))_R \right)^\top \in \mathbb{R}^R$.*

**Theorem 2** (Deterministic NTK under infinite width and infinite rank). *Let the CP SepNN be defined as $f_\Theta(x_1, \cdots, x_D) = \frac{1}{\sqrt{R}} \sum_{r=1}^{R} (f_{\Theta_1}(x_1))_r (f_{\Theta_2}(x_2))_r \cdots (f_{\Theta_D}(x_D))_r : \mathbb{R}^D \to \mathbb{R}$, where each factor MLP $f_{\Theta_d} : \mathbb{R} \to \mathbb{R}^R$ has the architecture $f_{\Theta_d}(x_d) = \frac{1}{\sqrt{W}} \boldsymbol{W}_{2,d} \sigma(\boldsymbol{W}_{1,d} x_d + \boldsymbol{b}_d)$, with $\boldsymbol{W}_{1,d} \in \mathbb{R}^{W \times 1}$, $\boldsymbol{b}_d \in \mathbb{R}^W$, $\boldsymbol{W}_{2,d} \in \mathbb{R}^{R \times W}$, and $\sigma$ a differentiable activation function with derivative $\dot{\sigma}$. Let each element of $\boldsymbol{W}_{1,d}, \boldsymbol{b}_d, \boldsymbol{W}_{2,d}$ be independently initialized by $\mathcal{N}(0,1)$. Then, as both the width $W \to \infty$ and the rank $R \to \infty$, the NTK of $f_\Theta$ converges almost surely to a deterministic kernel*

$$K_\Theta(\boldsymbol{x}, \boldsymbol{x}') \xrightarrow{\text{a.s.}} \sum_{d=1}^{D} k(x_d, x_d') \prod_{d' \neq d} c_{d'}(x_{d'}, x_{d'}'),$$

*where $c_d(x_d, x_d') = \mathbb{E}_{w,b \sim \mathcal{N}(0,1)} (\sigma(wx_d + b)\sigma(wx_d' + b))$ and $k(x_d, x_d') = c_d(x_d, x_d') + \mathbb{E}_{w,b \sim \mathcal{N}(0,1)} (\dot{\sigma}(wx_d + b)\dot{\sigma}(wx_d' + b)(x_d x_d' + 1))$.*

**Remark 1.** *The proof of Lemma 1 and Theorem 2 are placed in Appendix Sections A.6-A.7. While we consider the two-layer MLP in Theorem 2, it is straightforward to extend to multi-layer MLPs or other network structures by utilizing the corresponding NTK formulations (Arora et al., 2019b).*

**Remark 2.** *Using similar arguments to standard NTK analysis (Jacot et al., 2018; Arora et al., 2019a), it is easy to show that the deterministic NTK of SepNN also stays fixed (in terms of a small error) during training when $W, R \to \infty$ (under square loss function and infinitely-small learning rate). This can be achieved by bounding the difference of NTK at two training time points using $O(\frac{1}{\sqrt{R}})$ and $O(\frac{1}{\sqrt{W}})$ related to the small movement of each weight. Such asymptotic analysis of NTK during training is formally analyzed in Appendix Section A.4.*

---

[1]For simplicity, the NTK analysis is primarily conducted for the CP SepNN (2), while we believe it can be readily extended to TT and Tucker SepNNs. For the CP SepNN, we introduce a scaling factor $\frac{1}{\sqrt{R}}$ in the SepNN to ensure the convergence of NTK, which does not affect the universal approximation Theorem 1.

**Spectral Bias Characterization.** Consider training pairs $\{\boldsymbol{x}_i, y_i\}_{i=1}^n$ and square loss minimization: $\min_\Theta \frac{1}{2}\sum_{i=1}^n (f_\Theta(\boldsymbol{x}_i) - y_i)^2$ with $f_\Theta$ a CP SepNN. Using the same argument as classical NTK analysis (Jacot et al., 2018; Arora et al., 2019a), under infinitely small learning rate, the dynamics of network predictions $\boldsymbol{u}(t) = (f_{\Theta(t)}(\boldsymbol{x}_1), \cdots, f_{\Theta(t)}(\boldsymbol{x}_n))^\top \in \mathbb{R}^n$ under standard gradient descent optimizer would follow $\frac{d\boldsymbol{u}(t)}{dt} = -\boldsymbol{K}(t)(\boldsymbol{u}(t) - \boldsymbol{y})$, where $\Theta(t)$ and $\boldsymbol{K}(t) \in \mathbb{R}^{n\times n}$ respectively denote the weights and NTK matrix of the SepNN at training time $t$ over training samples $\{\boldsymbol{x}_i\}$ (i.e., $(\boldsymbol{K}(t))_{i,j} = K_\Theta(\boldsymbol{x}_i, \boldsymbol{x}_j)$), and $\boldsymbol{y} = (y_1, \cdots, y_n)^\top$. If the NTK of the SepNN stays fixed during training (as is expected for SepNNs with sufficiently large width and rank), the dynamics become $\frac{d\boldsymbol{u}(t)}{dt} = -\boldsymbol{K}(\boldsymbol{u}(t) - \boldsymbol{y})$ where $\boldsymbol{K}$ denotes the fixed NTK matrix. Denote the eigenvalue decomposition of $\boldsymbol{K}$ as $\boldsymbol{K} = \sum_{i=1}^n \lambda_i \boldsymbol{v}_i \boldsymbol{v}_i^\top$, where $\lambda_1 \geq \cdots \geq \lambda_n \geq 0$ are eigenvalue and $\boldsymbol{v}_1, \cdots, \boldsymbol{v}_n$ are orthogonal eigenvectors. Multiplying both sides by $\boldsymbol{v}_i$, we obtain $\frac{d}{dt}\left(\boldsymbol{v}_i^\top \boldsymbol{u}(t)\right) = -\boldsymbol{v}_i^\top \boldsymbol{K}(\boldsymbol{u}(t) - \boldsymbol{y}) = -\lambda_i \left(\boldsymbol{v}_i^\top (\boldsymbol{u}(t) - \boldsymbol{y})\right)$. This differential equation has an analytical solution:

$$\boldsymbol{v}_i^\top (\boldsymbol{u}(t) - \boldsymbol{y}) = \exp\left(-\lambda_i t\right)\left(\boldsymbol{v}_i^\top (\boldsymbol{u}(0) - \boldsymbol{y})\right). \tag{5}$$

Hence, the training error $(\boldsymbol{u}(t) - \boldsymbol{y})$ and its convergence rate can be characterized by the eigenvalues of the NTK matrix. Note that $\{\boldsymbol{v}_i\}$ form an orthogonal basis, hence $(\boldsymbol{u}(t) - \boldsymbol{y}) = \sum_{i=1}^n \boldsymbol{v}_i^\top (\boldsymbol{u}(t) - \boldsymbol{y})$. Each term $\boldsymbol{v}_i^\top (\boldsymbol{u}(t) - \boldsymbol{y})$ decays exponentially due to the $\exp\left(-\lambda_i t\right)$ term in (5), driving the total training error $(\boldsymbol{u}(t) - \boldsymbol{y})$ converging to zero as well. The components of labels $\boldsymbol{y}$ that project onto the eigenvectors $\boldsymbol{v}_i$ with larger eigenvalue $\lambda_i$ converge faster than those correspond to smaller eigenvalues due to the $\exp\left(-\lambda_i t\right)$ term, a property known as the spectral bias of neural networks trained with gradient descent (Geifman et al., 2024; Shi et al., 2025; 2024). This indicates that convergence is slower along directions corresponding to smaller eigenvalues, often requiring more training steps if the condition number of NTK matrix is large. The NTK-based convergence rate applies to SepNNs as well, hence SepNNs also exhibit inherent spectral bias due to the uneven eigenvalue distribution of the NTK matrix (Fig. 1(d)). In Section 4, we introduce an efficient SepPGD method to alleviate spectral bias in SepNNs by adjusting the eigenvalue distribution.

**Random NTK Under Fixed Rank.** In practice, the rank $R$ of SepNNs is often chosen to be smaller compared to network width to promote low-dimensional representations and better generalization (Liang et al., 2022; Luo et al., 2024). Therefore, we further consider the fixed rank and infinite width asymptotic regime, and prove that the NTK converges to a stochastic kernel defined by Gaussian processes associated with the covariance of factor MLPs. This indicates that infinite rank $R$ is necessary to obtain a deterministic NTK by applying the large number law to vanish the covariance.

**Corollary 1** (Random NTK under infinite width and fixed rank). *Let the CP SepNN be defined as $f_\Theta(x_1, \cdots, x_D) = \frac{1}{\sqrt{R}}\sum_{r=1}^R (f_{\Theta_1}(x_1))_r (f_{\Theta_2}(x_2))_r \cdots (f_{\Theta_D}(x_D))_r : \mathbb{R}^D \to \mathbb{R}$ and the assumptions in Theorem 2 hold. Then, for a fixed rank $R$, as the width of the factor MLP $W \to \infty$, the NTK of the SepNN $f_\Theta$ converges in distribution to a stochastic kernel*

$$K_\Theta(\boldsymbol{x}, \boldsymbol{x}') \xrightarrow{\mathrm{d}} \sum_{d=1}^D k(x_d, x_d') V_d(\boldsymbol{x}, \boldsymbol{x}'),$$

*where $V_d(\boldsymbol{x}, \boldsymbol{x}') = \frac{1}{R}\sum_{r=1}^R \prod_{d'\neq d}\left(f_{\Theta_{d'}}(x_{d'})\right)_r \left(f_{\Theta_{d'}}(x_{d'}')\right)_r$, in which each factor $\left(f_{\Theta_{d'}}(x_{d'})\right)_r$ is a Gaussian process with covariance $\mathbb{E}\left((f_{\Theta_{d'}}(x_{d'}))_r (f_{\Theta_{d'}}(x_{d'}'))_r\right) = c_d(x_{d'}, x_{d'}')$.*

**Remark 3.** *Under the fixed rank condition, the training dynamic can not be characterized uniformly using a fixed NTK matrix as in (5) due to the randomness. However, the random NTK can at least characterize the training dynamic within a small range of training time around $t$ by using the potential stochastic differential equation and probability bound, which are promising future directions. We also empirically find that even with small rank, the proposed SepPGD method is effective in accelerating convergence and alleviating spectral bias in SepNNs (Appendix Table 3).*

All NTK theoretical results are empirically validated in Fig. 1. Fig. 1(a) shows that under a fixed rank $R$, the NTK at initialization does not converge to a fixed kernel and holds randomness even with larger network width (Corollary 1). Fig. 1(b) shows that the NTK at initialization tends to converge towards a deterministic kernel with joint increase of network width and decomposition rank (Theorem 2). Fig. 1(c) shows that the NTK tends to stay fixed during training with joint increase of network width and decomposition rank (Remark 2). Finally, Fig. 1(d) illustrates the

spectral bias that the eigenvalues of the SepNN's NTK matrix decay rapidly, resulting in slower convergence within components that project onto eigenvectors of these smaller eigenvalues.

Another positive property of SepNN is that its NTK matrix can be computed in parallel over multiple inputs, resulting in greater efficiency compared to computing the NTK matrices of other neural architectures (Novak et al., 2022; Mohamadi et al., 2023). In fact, the SepNN's NTK (12) admits an elegant form that allows the NTK matrix over grid inputs to be expressed as a Kronecker product of smaller NTK matrices (detailed in Appendix Section A.3), thereby improving efficiency.

## 4 EFFICIENT SEPARABLE PRECONDITIONED GRADIENT DESCENT

**Prior Arts.** The NTK-based preconditioning methods are proposed in recent works (Geifman et al., 2024; Shi et al., 2025), which adjust the eigenvalue distribution of the NTK matrix, thereby modulating the convergence rate. Especially, consider a $D$-dimensional data grid and $n^D$ training labels sorted in a vector $\boldsymbol{y} \in \mathbb{R}^{n^D}$, and the vector of network predictions $f_\Theta(\boldsymbol{X}) \in \mathbb{R}^{n^D}$, where $\boldsymbol{X} \in \mathbb{R}^{D \times n^D}$ are batched inputs. The training residual is $\boldsymbol{r} = f_\Theta(\boldsymbol{X}) - \boldsymbol{y}$ and the square loss is $\|\boldsymbol{r}\|_{\ell_2}^2$. Then, a gradient descent iteration with learning rate $\eta$ is given by $\Theta \leftarrow \Theta - \eta \nabla_\Theta f_\Theta(\boldsymbol{X})^\top \boldsymbol{r}$. According to (5), the convergence rate is related to the spectrum of NTK matrix, denoted by $\boldsymbol{K}$ here. (Geifman et al., 2024) proposed the following PGD with a preconditioning matrix $\boldsymbol{S}$: $\Theta \leftarrow \Theta - \eta \nabla_\Theta f_\Theta(\boldsymbol{X})^\top \boldsymbol{S} \boldsymbol{r}$. In this way, the convergence rate is related to the eigenvalues of $\boldsymbol{K} \boldsymbol{S}$. Hence, the preconditioner $\boldsymbol{S} \in \mathbb{R}^{n^D \times n^D}$ is constructed to modulate the spectrum of $\boldsymbol{K} \boldsymbol{S}$ so as to improve convergence (Geifman et al., 2024). For $n^D$ training samples, this method has complexity $O(n^D)$ by applying $\boldsymbol{S}$. Later, (Shi et al., 2025) leveraged this method to alleviate the spectral bias of INRs. They reduced the complexity of NTK matrix application by sampling a mini-batch, which enjoys $O(n^D/p)$ complexity with $p > 1$ denoting the number of mini-batches. Their preconditioning method can be expressed as $\Theta \leftarrow \Theta - \eta \nabla_\Theta f_\Theta(\boldsymbol{X}_i)^\top \boldsymbol{S}_i \boldsymbol{r}_i$, where $\boldsymbol{X}_i, \boldsymbol{S}_i, \boldsymbol{r}_i$ are sampled mini-batches. Different from these methods, our SepPGD applies smaller preconditioners separately for factor MLPs, further reducing computational complexity.

**SepPGD and Properties.** We first elaborate the training configuration of SepNNs for square loss optimization on grid points[2]. SepNNs exhibit efficiency benefits when applied to separable grid points (Liang et al., 2022; Cho et al., 2023). We therefore consider input points situated on a grid $\hat{\boldsymbol{x}}_1 \times \cdots \times \hat{\boldsymbol{x}}_D = \{(x_1, \cdots, x_D) \mid x_d \in \hat{\boldsymbol{x}}_d, d = 1, \cdots, D\} \subset \mathbb{R}^D$, where each $\hat{\boldsymbol{x}}_d$ is a discrete set (or say vector) of points in $\mathbb{R}$. For simplicity, we assume that each $\hat{\boldsymbol{x}}_d$ has cardinality $n$, meaning each dimension contains $n$ training samples, resulting in a total of $n^D$ training samples. Consider a SepNN $f_\Theta(\boldsymbol{x}) = \frac{1}{\sqrt{R}} \sum_{r=1}^R (f_{\Theta_1}(x_1))_r \cdots (f_{\Theta_D}(x_D))_r : \mathbb{R}^D \to \mathbb{R}$. Each factor MLP $f_{\Theta_d} : \mathbb{R} \to \mathbb{R}^R$ is applied in parallel to the input vector $\hat{\boldsymbol{x}}_d \in \mathbb{R}^n$, yielding a factor matrix $f_{\Theta_d}(\hat{\boldsymbol{x}}_d) \in \mathbb{R}^{R \times n}$. The corresponding optimization model is formulated as

$$\min_\Theta \frac{1}{2} \sum_{i=1}^{n^D} (f_\Theta(\boldsymbol{x}_i) - y_i)^2, \ \boldsymbol{x}_i \in \hat{\boldsymbol{x}}_1 \times \cdots \times \hat{\boldsymbol{x}}_D. \tag{6}$$

Owing to the grid structure of the training data, the labels $\{y_i\}_{i=1}^{n^D}$ can be naturally reshaped into a $D$-th order tensor $\mathcal{Y} \in \mathbb{R}^{n \times \cdots \times n}$. Let $\mathcal{R} := (\mathcal{Z}_\Theta - \mathcal{Y}) \in \mathbb{R}^{n \times \cdots \times n}$ denote the residual tensor during training, where $(\mathcal{Z}_\Theta)_{i_1, \cdots, i_D} = f_\Theta((\hat{\boldsymbol{x}}_1)_{i_1}, \cdots, (\hat{\boldsymbol{x}}_D)_{i_D})$ is the $(i_1, \cdots, i_D)$-th SepNN output.

The motivation behind SepPGD is to perform PGD for each factor MLP separately. We first compute $D$ factor preconditioning matrices $\{\boldsymbol{S}_d\}_{d=1}^D$ for the $D$ factor MLPs $\{f_{\Theta_d}\}_{d=1}^D$. This involves calculating a pseudo NTK matrix $\boldsymbol{K}_{\Theta_d} \in \mathbb{R}^{n \times n}$ for each $f_{\Theta_d}$ on the corresponding input data $\hat{\boldsymbol{x}}_d \in \mathbb{R}^n$ using sum-of-logits (Mohamadi et al., 2023), followed by eigenvalue modulation as described in (Geifman et al., 2024; Shi et al., 2025). Specifically, we calculate $\boldsymbol{S}_d = \boldsymbol{I} - \sum_{i=1}^k (1 - \frac{g(\lambda_i)}{\lambda_i}) \boldsymbol{v}_i \boldsymbol{v}_i^\top$, where $\{\lambda_i, \boldsymbol{v}_i\}$ are eigenvalues and eigenvectors of $\boldsymbol{K}_{\Theta_d}$. We set the modulation function $g(\lambda_i) = \lambda_k$ for $i \leq k$ and $g(\lambda_i) = \lambda_i$ for $i > k$ by following (Shi et al., 2025), which makes the eigenvalues

---

[2]Using grid inputs is a common training configuration for SepNNs, such as coordinate grids of images in INRs (Liang et al., 2022) and grid collocation points in PINNs (Cho et al., 2023), which facilitate efficient parallel training. For non-grid inputs, SepNNs remain applicable, though the computational complexity for NTK evaluation and SepPGD becomes equivalent to standard networks (Liang et al., 2022; Cho et al., 2023).

of $\boldsymbol{S}_d$ more evenly distributed for the first $k$ eigenvalues. Furthermore, denote $\oplus$ the concatenation of vectors. We are now ready to present the proposed SepPGD algorithm.

**Definition 1** (Separable PGD). *Consider the optimization problem (6) and the SepNN $f_\Theta(\boldsymbol{x}) = \frac{1}{\sqrt{R}} \sum_{r=1}^R (f_{\Theta_1}(x_1))_r \cdots (f_{\Theta_D}(x_D))_r : \mathbb{R}^D \to \mathbb{R}$. Given $D$ symmetric factor preconditioning matrices $\{\boldsymbol{S}_d \in \mathbb{R}^{n \times n}\}_{d=1}^D$ for factor MLPs $\{f_{\Theta_d}\}_{d=1}^D$, the SepPGD iteration is given by*

$$\Theta \leftarrow \Theta - \eta \bigoplus_{d=1}^D \underbrace{(\nabla_{\Theta_d} \langle f_{\Theta_d}(\hat{\boldsymbol{x}}_d), \boldsymbol{M}_d \rangle)}_{\text{gradient of factor MLP}}, \tag{7}$$

*where $\boldsymbol{M}_d$ is the mode-$d$ specific preconditioner defined by*

$$\boldsymbol{M}_d = \left(\underbrace{\bigoplus_{r=1}^R \bigotimes_{d' \neq d} f_{\Theta_{d'}}(\hat{\boldsymbol{x}}_{d'})_{r,:}}_{R \times n^{D-1}}\right) \left(\underbrace{\texttt{unfold}_d \left(\sum_{d=1}^D (\mathcal{R} \times_d \boldsymbol{S}_d)\right)}_{n^{D-1} \times n}\right) \in \mathbb{R}^{R \times n}, \ d = 1, \cdots, D, \tag{8}$$

*where the subscript $f_{\Theta_{d'}}(\hat{\boldsymbol{x}}_{d'})_{r,:}$ refers to the $r$-th row of the factor matrix $f_{\Theta_{d'}}(\hat{\boldsymbol{x}}_{d'}) \in \mathbb{R}^{R \times n}$. The $\bigotimes_{d' \neq d}$ refers to the outer product that calculates between $D-1$ vectors $\{f_{\Theta_{d'}}(\hat{\boldsymbol{x}}_{d'})_{r,:} \in \mathbb{R}^{1 \times n}\}$ and returns a single long vector of size $1 \times n^{D-1}$, and $\bigoplus_{r=1}^R$ denotes the concatenation between $R$ vectors and returns an $R \times n^{D-1}$ matrix. The $\texttt{unfold}_d : \mathbb{R}^{n_1 \times \cdots \times n_D} \to \mathbb{R}^{\prod_{d' \neq d} n_{d'} \times n_d}$ denotes the unfolding operator from a tensor to a matrix, and $\times_d$ denotes the mode-$d$ tensor-matrix product, a convention in tensor decomposition literature (Kolda & Bader, 2009; Luo et al., 2024).*

**Remark 4** (Computational complexity). *The computational complexity comparison of different preconditioning methods is given in Table 1. In terms of applying the preconditioner, SepPGD scales as $O(nD)$ by multiplying $D$ $n$-by-$n$ preconditioning matrices $\{\boldsymbol{M}_d\}$, while standard NTK-based method (Geifman et al., 2024) scales as $O(n^D)$ by multiplying an $n^D$-by-$n^D$ preconditioning matrix $\boldsymbol{S}$. Moreover, the preconditioner construction stage in SepPGD is also more efficient. Specifically, SepPGD constructs the preconditioner by calculating the NTK matrix and performing eigenvalue decomposition for $D$ factor NTK matrices $\{\boldsymbol{K}_{\Theta_d}\}$, each of size $n \times n$. In contrast, classical NTK-based PGD (Geifman et al., 2024) requires the same operations on a single large NTK matrix of size $n^D \times n^D$. Therefore, the preconditioner construction complexity for SepPGD scales as $O(D(n^3 + n^2 P))$ (note that NTK matrix calculation consumes $O(n^2 P)$ and eigenvalue decomposition consumes $O(n^3)$)[3], where $P$ is the number of network parameters, while the classical NTK-based PGD method scales as $O(n^{3D} + n^{2D} P)$. We therefore see that SepPGD is more efficient in both preconditioner construction and application.*

Table 1: Computational complexity comparison (in terms of applying the preconditioner) of several preconditioning methods for $n^D$ training samples using an over-parameterized SepNN $f_\Theta : \mathbb{R}^D \to \mathbb{R}$ with $P$ learnable parameters ($n^D < P$). Here, $\boldsymbol{r}$ denotes training residual, $\boldsymbol{H} \in \mathbb{R}^{P \times P}$ denotes the Hessian matrix, $\boldsymbol{S} \in \mathbb{R}^{n^D \times n^D}$ denotes the NTK-based preconditioning matrix, $\boldsymbol{S}_i \in \mathbb{R}^{\frac{n^D}{p} \times \frac{n^D}{p}}$ denotes the mini-batch version of the NTK preconditioning matrix, and $\boldsymbol{M}_d \in \mathbb{R}^{R \times n}$ denotes the proposed mode-$d$ specific preconditioner defined in (8).

| Preconditioning method | Gradient formulation | Complexity |
|---|---|---|
| Hessian-based methods | $\boldsymbol{H}^{-1} \nabla_\Theta f_\Theta(\boldsymbol{X})^\top \boldsymbol{r}$ | $O(P)$ |
| Modified NTK spectrum (Geifman et al., 2024) | $\nabla_\Theta f_\Theta(\boldsymbol{X})^\top \boldsymbol{S} \boldsymbol{r}$ | $O(n^D)$ $(n^D < P)$ |
| Inductive gradient adjustment (Shi et al., 2025) | $\nabla_\Theta f_\Theta(\boldsymbol{X}_i)^\top \boldsymbol{S}_i \boldsymbol{r}_i$ (mini-batch) | $O(n^D/p)$ $(p > 1)$ |
| **Separable PGD (Ours)** | $\bigoplus_{d=1}^D \nabla_{\Theta_d} \langle f_{\Theta_d}(\hat{\boldsymbol{x}}_d), \boldsymbol{M}_d \rangle$ | $O(nD)$ |

While the formulations in (7)-(8) seem complicated, we can more easily interpret the motivation and preconditioner construction $\boldsymbol{M}_d$ in the two-dimensional case $D = 2$ by connecting it with classical methods (Geifman et al., 2024; Shi et al., 2025).

---

[3]Note that constructing the preconditioner in (8) also involves a matrix product with complexity $O(n^{D-1})$. However, the primary computational cost of NTK-based PGD lies in the computation of the NTK matrix, which requires high-dimensional derivative calculations, and the eigenvalue decomposition of the NTK matrix, which involves inner numerical iterations (Geifman et al., 2024; Shi et al., 2025). In comparison, the matrix product in (8) is orders of magnitude less expensive in practice.

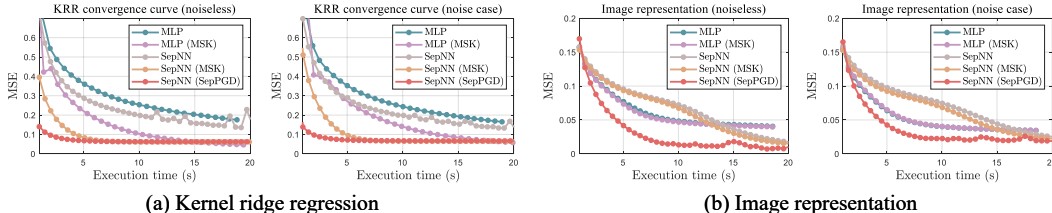

Figure 2: Convergence curves for KRR and image representation using INRs.

**Lemma 2.** *Let the SepNN be $f_\Theta(\boldsymbol{x}) = f_{\Theta_1}(x_1)^\top f_{\Theta_2}(x_2) : \mathbb{R}^2 \to \mathbb{R}$ and the suppositions in Definition 1 hold. Then the SepPGD iteration in (7) is equivalent to the classical NTK-based PGD as $\Theta \leftarrow \Theta - \eta \nabla_\Theta f_\Theta(\boldsymbol{X})^\top \tilde{\boldsymbol{S}} \boldsymbol{r}$, where $\boldsymbol{X} \in \mathbb{R}^{2 \times n^2}$ are batched inputs, $\boldsymbol{r} = \mathrm{vec}(\mathcal{R}^\top)$ is the vectored training residual, and $\tilde{\boldsymbol{S}} = (\boldsymbol{S}_1 \otimes \boldsymbol{I}_n + \boldsymbol{I}_n \otimes \boldsymbol{S}_2) \in \mathbb{R}^{n^2 \times n^2}$ is constructed by Kronecker product. Specifically, this implies $[\nabla_{\Theta_1} \langle f_{\Theta_1}(\hat{\boldsymbol{x}}_1), \boldsymbol{M}_1 \rangle, \nabla_{\Theta_2} \langle f_{\Theta_2}(\hat{\boldsymbol{x}}_2), \boldsymbol{M}_2 \rangle] = \nabla_\Theta f_\Theta(\boldsymbol{X})^\top \tilde{\boldsymbol{S}} \boldsymbol{r}$.*

The proof is provided in Appendix Section A.9. Lemma 2 establishes the connection between the proposed SepPGD method (7) and the classical NTK-based PGD (Geifman et al., 2024; Shi et al., 2025) under an appropriate choice of the large preconditioner $\tilde{\boldsymbol{S}}$. Although the two PGD methods are equivalent in this setting, SepPGD is computationally more efficient. This efficiency comes from the parallel computation of factor gradients. Specifically, the total gradient $\nabla_\Theta f_\Theta(\boldsymbol{X})^\top \tilde{\boldsymbol{S}} \boldsymbol{r}$ can be written in the form of outer products like $(\boldsymbol{C}^\top \otimes \boldsymbol{A}) \mathrm{vec}(\boldsymbol{B})$ (see Section A.9), whereas the factor gradients $\{\nabla_{\Theta_d} \langle f_{\Theta_d}(\hat{\boldsymbol{x}}_d), \boldsymbol{M}_d \rangle\}_{d=1}^D$ in SepPGD are computed via matrix products like $\mathrm{vec}(\boldsymbol{A}\boldsymbol{B}\boldsymbol{C})$ in the $O(n)$ dimensional space. This is more efficient than evaluating the matrix product $(\boldsymbol{C}^\top \otimes \boldsymbol{A}) \mathrm{vec}(\boldsymbol{B})$ in the $O(n^2)$ dimension. The key property we are using here is the equivalence between the Kronecker product expression and the vectorized matrix product, i.e., $(\boldsymbol{C}^\top \otimes \boldsymbol{A}) \mathrm{vec}(\boldsymbol{B}) = \mathrm{vec}(\boldsymbol{A}\boldsymbol{B}\boldsymbol{C})$. This allows us to decompose the large preconditioner $\tilde{\boldsymbol{S}} \in \mathbb{R}^{n^2 \times n^2}$ into smaller preconditioners $\boldsymbol{M}_d \in \mathbb{R}^{n \times n}$ and perform the SepPGD in an efficient manner. Lemma 2 can also be extended to non-grid inputs. Especially, if we construct the preconditioner as $\tilde{\boldsymbol{S}} = \sum_d \boldsymbol{S}_d$, then the SepPGD gradient $\nabla_{\Theta_d} \langle f_{\Theta_d}((\boldsymbol{X})_{d,:}), \boldsymbol{M}_d \rangle$ for some non-grid inputs $\boldsymbol{X} \in \mathbb{R}^{D \times n}$ ($n$ non-grid points in $D$-dimensional space) is equivalent to the classical NTK-based PGD update $\nabla_\Theta f_\Theta(\boldsymbol{X})^T \tilde{\boldsymbol{S}} \boldsymbol{r}$ under element-wise evaluation without Kronecker product structure. Here, $\boldsymbol{M}_d = \mathrm{einsum}(\odot_{d' \neq d} f_{\Theta'_d}((\boldsymbol{X})_{d',:}), \sum_d \boldsymbol{S}_d \boldsymbol{r}; (R, n) \times (n) \to (R, n))$ is the $d$-th preconditioner constructed by Einstein product, where $\odot$ denotes element-wise product. We use this formulation to test SepPGD for non-grid inputs; see Section A.2.

Given the equivalence between SepPGD and NTK-based PGD (Geifman et al., 2024), it therefore remains to show that the constructed preconditioner $\tilde{\boldsymbol{S}} = (\boldsymbol{S}_1 \otimes \boldsymbol{I}_n + \boldsymbol{I}_n \otimes \boldsymbol{S}_2)$ is indeed effective in adjusting the eigenvalue distribution of $\boldsymbol{K}\tilde{\boldsymbol{S}}$ to accelerate convergence. This can possibly be verified, because the eigenvalue of a Kronecker product matrix $\boldsymbol{S}_1 \otimes \boldsymbol{I}_n$ is the product of eigenvalues of $\boldsymbol{S}_1$ and $\boldsymbol{I}_n$. Therefore, $\tilde{\boldsymbol{S}}$ would have better spectrum (i.e., smaller condition number) than $\tilde{\boldsymbol{K}} = (\boldsymbol{K}_{\Theta_1} \otimes \boldsymbol{I}_n + \boldsymbol{I}_n \otimes \boldsymbol{K}_{\Theta_2})$ since $\boldsymbol{S}_d$ has better spectrum than $\boldsymbol{K}_{\Theta_d}$. Suppose that $\tilde{\boldsymbol{K}}$ is close to the true NTK matrix $\boldsymbol{K}$ (which can be verified using the NTK matrix formulation in Lemma 3). We can ultimately show that $\boldsymbol{K}\tilde{\boldsymbol{S}}$ has better spectrum than $\boldsymbol{K}$. Therefore, the proposed SepPGD could provably and efficiently adjust the spectrum of the SepNN's NTK matrix during training, effectively alleviating spectral bias. In practice, SepPGD allows the preconditioner $\{\boldsymbol{M}_d\}$ to be efficiently updated every ten iterations, which is computationally expensive in previous methods. It is believed that the result in Lemma 2 (and the analysis following) can be readily extended to multivariate cases $D > 2$. Also, based on the convergence result of NTK-based preconditioning algorithm (Geifman et al., 2024), we can also deduce the convergence and solution consistency (w.r.t. standard gradient descent) of our SepPGD algorithm by using the equivalence between SepPGD and the corresponding kernel ridge regression with representer theorem. This is left for future research.

## 5 NUMERICAL EXPERIMENTS

The numerical results are presented to verify the effectiveness of SepPGD for improving convergence of SepNNs. Examples include kernel ridge regression (KRR), image and surface representation using INRs, and PINNs. Detailed experimental settings are placed in Appendix Section A.12.

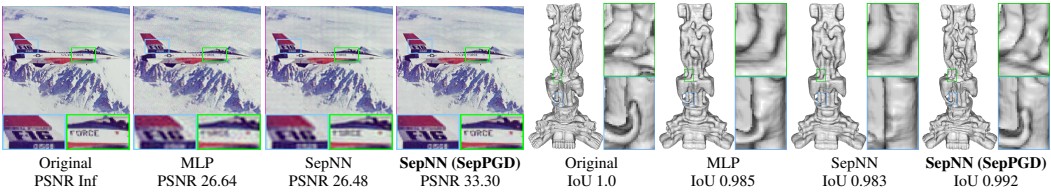

| Original | MLP | SepNN | **SepNN (SepPGD)** | Original | MLP | SepNN | **SepNN (SepPGD)** |
|----------|-----|-------|--------------------|----------|-----|-------|--------------------|
| PSNR Inf | PSNR 26.64 | PSNR 26.48 | PSNR 33.30 | IoU 1.0 | IoU 0.985 | IoU 0.983 | IoU 0.992 |

Figure 3: Visual results for image and 3D surface representation using SepPGD.

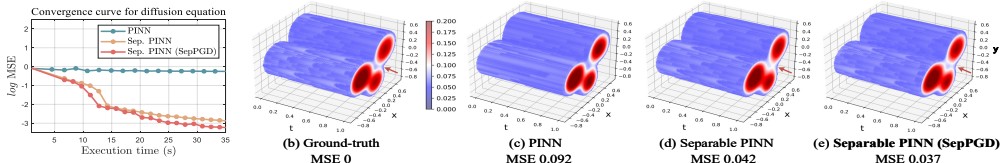

Figure 4: Convergence curves and results for the 3D diffusion equation using PINNs.

**KRR.** Following (Geifman et al., 2024), we perform KRR by using the gradients of neural network w.r.t. parameters as the feature function of kernel (see Appendix Section A.12 for detailed formulation and settings). We test both MLP and CP SepNN, and compare SepPGD with the classical NTK-based PGD, the modified spectrum kernel (MSK) (Geifman et al., 2024; Shi et al., 2025). Following (Geifman et al., 2024), we consider both noisy (standard deviation 0.01) and noiseless cases. The convergence behavior, measured by testing MSE during training, is shown in Fig. 2(a). Because the efficiency advantage of SepNN and SepPGD comes from the lower complexity in an iteration, we plot the convergence curve w.r.t. execution time rather than iteration number. In noiseless case, SepPGD achieves the fastest convergence. In the presence of noise, SepPGD remains robust, while MSK has slower convergence. SepPGD improves SepNN to a large margin as shown in Fig. 2.

**Image Representation and Recovery.** We leverage INR for image representation and recovery (inpainting) by following the settings in (Sitzmann et al., 2020; Shi et al., 2025). Convergence curves and representation visual examples are shown in Fig. 2(b) and Fig. 3's left. SepPGD effectively improves the convergence of SepNN by alleviating spectral bias and better capturing image fine details. Moreover, SepPGD accelerates the convergence without affecting the model's generalization (in most cases improving generalization); see image inpainting results in Appendix Fig. 10.

**Surface Representation.** We perform surface representation by representing the volumetric occupancy grids of a 3D surface using INRs (Shi et al., 2025). The results (under the same iteration number) with intersection over union (IoU) are shown in Fig. 3's right. SepPGD effectively improves the ability of SepNN to capture surface textures and details by alleviating spectral bias during training.

**PINNs.** We use the separable PINN (Cho et al., 2023) (CP SepNN) and perform the tests on 3D diffusion, Klein-Gordon, and Helmholtz equations using grid samples. Convergence curves using testing MSE vs. time and visual examples (under the same iteration) are shown in Fig. 4. More results are shown in Appendix Figs. 13-14. The separable PINN (Cho et al., 2023) is more efficient than classical PINN, and SepPGD further enhances the convergence speed of separable PINN.

## 6 CONCLUSIONS AND DISCUSSIONS

We established the universal approximation theory for SepNNs, deduced their NTK regimes, and proposed an efficient separable PGD to alleviate spectral bias of SepNNs. The algorithm was shown to be effective in various applications such as image & surface representation and PINNs. We further elaborate on the potential impact of our proposed theory and method. First, SepNNs have been attracting growing attention in various applications due to their efficient structure, such as INRs and PINNs. Numerical experiments demonstrate that SepPGD can effectively speed up convergence in applications involving INRs and PINNs (e.g., in image and surface representation and numerical PDEs). Therefore, we believe that our theory and the corresponding algorithm have the potential to benefit these fields and address related challenges. Moreover, SepNNs are also increasingly being adopted across diverse scientific domains by leveraging their efficient structure and interpretability (Cho et al., 2023; Song et al., 2023; Chen et al., 2025). Therefore, understanding the theoretical approximation ability, training behavior (e.g., NTK regime), and addressing the potential optimization challenge of SepNNs are believed to be valuable and important for these practical applications.

ACKNOWLEDGMENTS

This work is supported by the Fundamental and Interdisciplinary Disciplines Breakthrough Plan of the Ministry of Education of China (JYB2025XDXM101), the NSFC (No. 124B2029, 62476214), and the Tianyuan Fund for Mathematics of the National Natural Science Foundation of China (Grant No. 12426105). We thank the anonymous reviewers for the constructive and valuable comments.

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

## A APPENDIX

### A.1 MORE DISCUSSIONS

#### A.1.1 EFFICIENCY ADVANTAGE OF SEPNNS

We make more discussions on the efficiency advantage of SepNNs. The computational advantage of SepNNs is most relevant in certain structured settings, and does not directly imply faster training in all scenarios compared to architectures like convolutional neural networks (CNNs). We would like to clarify that SepNNs are primarily designed for problems where the input is a set of coordinate points on a grid, such as in INRs and PINNs. In these cases, the input is not a high-dimensional signal like an image, but rather a low-dimensional coordinate space. CNNs, while highly efficient for image-like data through localized convolution and parallelism, are less naturally suited to such

coordinate-based inputs, as they rely on spatial locality and translation invariance, which may not align with the structure of the problem. In contrast, SepNNs exploit the separable structure of the grid to reduce the number of unique function evaluations from $O(n^D)$ (as in a standard MLP applied naively to all grid points) to $O(nD)$, resulting in faster training on grid coordinate points. This makes SepNNs particularly advantageous in training coordinate networks on grids, such as image representation using INRs and evaluating grid collocation points in PINNs. Therefore, the benefits of SepNNs are most pronounced in these structured settings.

From a mathematical perspective, the theory may inspire several promising future research directions. SepNNs are inherently related to decomposition-based algebraic structures such as nonnegative matrix factorization (NMF) and tensor decomposition. Therefore, our NTK analysis for SepNNs provides a theoretical tool to bridge neural network optimization with classical NMF, tensor decomposition, and related scientific computing topics. Furthermore, the NTK theory can be naturally extended to separable operator learning (Cho et al., 2023), providing a theoretical tool for learning mappings between function spaces. Moreover, based on the NTK analysis, another future research line is to study adaptive feature learning regime (Haobo Zhang, 2024) induced by SepNNs, which is particularly interesting for SepNNs since they can learn disentangled representations, making them suited to extract interpretable features from multimodal data.

### A.1.2 DISCUSSIONS ON THEORETICAL ANALYSIS

Our current theoretical analysis establishes the universal approximation capability of SepNNs and does not yet provide explicit approximation error rates in terms of network rank or width. In future work, we can consider function spaces spanned by Fourier bases and leveraging the decay properties of Fourier coefficients (Herman, 2016) to derive approximation error bounds with respect to the rank, in order to provide theoretical guidance for practical rank selection. This could be achieved by formulating the Fourier expansion in the SepNN form (Kargas & Sidiropoulos, 2021). Moreover, we can try to characterize the approximation ability of SepNNs w.r.t. width by borrowing approximation theory for narrow neural networks (Kidger & Lyons, 2020) in future research.

While this work considers the CP-type SepNN (which is the most classical SepNN type) to perform the NTK analysis, we believe the NTK analysis can be extended to TT- and Tucker-type SepNNs in a similar fashion. For TT-type SepNNs, the derivation would closely follow CP, relying on tensor products of NTKs corresponding to factor MLPs. For Tucker-type SepNNs, analogous NTK results are also expected to hold, provided the asymptotic behavior and boundedness of the core tensor $\mathcal{C}$ are appropriately treated. We regard this extension as a valuable direction for future research.

Our current theoretical analysis does not yet provide comprehensive convergence guarantees or generalization error bounds for the fixed-rank regime. Such analysis is crucial for a deeper understanding of the training and generalization behaviors of SepNNs. Still, we believe the present theoretical framework offers a valuable initial insight, as it establishes a convergence analysis for SepNNs under infinite rank regime. In fact, by applying probability bounds related to the law of large numbers with respect to the rank parameter $R$, we anticipate that an $\epsilon$-convergence bound on the training residual could be derived for a fixed rank $R = O(1/\epsilon^2)$, in a manner analogous to classical probability convergence bound analyses for wide MLPs (see, e.g., Theorem 4.1 in (Arora et al., 2019a)). In future work, we can leverage more advanced tools from stochastic NTK analysis and explore training dynamics beyond the convergent NTK regime (Liu et al., 2020), to provide a more comprehensive theoretical understanding on convergence and generalization of SepNNs under small rank. Moreover, in the fixed-rank regime, the factor MLPs of the SepNN converge to random Gaussian processes. This implies that their outputs serve as random feature maps for each input. The SepNN then performs a tensor product of these random features. This connection allows us to consider the well-established theoretical framework of random feature models (Rahimi & Recht, 2007) to analyze the potential property of SepNNs under fixed-rank settings, providing a way to derive non-asymptotic convergence rates and generalization bounds for fixed-rank SepNN.

The proposed preconditioner in this work is designed to modulate the eigenvalue distribution of the NTK matrix $\boldsymbol{K}$ to accelerate training convergence. By applying the modulation function $g(\lambda)$ to smooth the top $k$ eigenvalues, the resulting preconditioned product matrix $\boldsymbol{K}\tilde{\boldsymbol{S}}$ is guaranteed to have a condition number no larger than $\boldsymbol{K}$. This is because the dominant large eigenvalues are reduced, leading to a more uniform spectral distribution. In future work, we can derive quantitative bounds on the condition number of the modulated NTK matrix by leveraging its eigenvalue decay properties.

For instance, tools from (Li et al., 2024) on the eigenvalue decay rates of kernel functions could be employed. We regard this as a promising direction for further research. Regarding the effect of the preconditioner on generalization, existing studies (e.g., (Geifman et al., 2024; Tsigler & Bartlett, 2023)) suggest that slowing the eigenvalue decay of the kernel matrix can introduce an implicit regularization effect, helping prevent overfitting and improving generalization. While the current manuscript does not include a rigorous theoretical analysis of generalization, the above references provide a way for establishing formal generalization guarantees for SepPGD.

While we consider the two-layer MLP in Theorem 2, it is straightforward to extend to multi-layer MLPs or other network structures by utilizing the corresponding NTK formulations (Arora et al., 2019b). Especially, for SepNNs with multi-layer factor neural networks, we can use the corresponding NTK analysis for multi-layer neural networks to study their theoretical property (Arora et al., 2019b). Especially, (Arora et al., 2019b) has shown that the NTK of a multi-layer neural network (beyond two-layer) converges to a deterministic kernel under infinite width. Their results are based on the recursive formulation of the covariance matrix of hidden layers in a multi-layer network (cf. Theorem 3.1 in (Arora et al., 2019b)). By borrowing this type of technique, we believe the theoretical result for SepNN (Theorem 2 in our manuscript) can be readily extended to multi-layer MLPs by replacing the two-layer factor NTK $k(x_d, x'_d)$ with multi-layer NTK formulations.

Our analysis in this paper can be readily generalized to multi-output SepNN $f_\Theta : \mathbb{R}^D \to \mathbb{R}^{D_{\text{out}}}$ by applying multi-output NTK approximation (Mohamadi et al., 2023). Moreover, the detailed convergence and generalization bounds of SepNNs under fixed rank conditions warrants further discussions. It is also interesting to apply SepPGD to more applications, such as implicit full waveform inversion (Borcea et al., 2024; Chen et al., 2025) and diffeomorphic autoencoding (Dummer et al., 2024).

### A.1.3   RELATIONS TO KAN

The recently popular structure KAN (Liu et al., 2025) is closely related to the SepNN. Especially, each layer of KAN can be viewed as a type of SepNN. Specifically, consider the input vector $\boldsymbol{x} = [\boldsymbol{x}_{(1)}, \boldsymbol{x}_{(2)}, \cdots, \boldsymbol{x}_{(n)}]^T$. A single KAN layer (Liu et al., 2025) (take single-output as an example), denoted as $\Phi(\cdot) : \mathbb{R}^n \to \mathbb{R}$, is defined as $\Phi(\boldsymbol{x}) = \sum_{j=1}^n \phi_{1,j}(\boldsymbol{x}_{(j)})$, where each univariate function $\phi_{1,j}(\cdot) : \mathbb{R} \to \mathbb{R}$ is parameterized as $\phi_{1,j}(\boldsymbol{x}_{(j)}) = \omega^{1,j} \left[ b(\boldsymbol{x}_{(j)}) + \sum_{k=1}^K c_k^{1,j} B_k(\boldsymbol{x}_{(j)}) \right]$. Here, $\{c_k^{1,j}\}$ for $k = 1, \ldots, K$ are trainable parameters, and $\{B_k(\cdot)\}$ are spline basis functions as used in (Liu et al., 2025). We observe that the output $\Phi(\boldsymbol{x})$ is a linear combination (specifically, summation) of learnable univariate functions $\phi_{1,j}(\boldsymbol{x}_{(j)})$, which shares a similar spirit with SepNNs. Therefore, each KAN layer can be viewed as a SepNN, though it does not directly correspond to the CP, TT, or Tucker types. The distinction lies in how the linear combination is realized. In KAN, it is implemented via simple addition of univariate functions, while in the SepNNs discussed in our manuscript, the combination is realized via tensor product (e.g., CP, TT, or Tucker). The most significant commonality between KAN and SepNN is that both architectures learn univariate functions and compose multivariate functions through linear combinations of these univariate components. This characteristic fundamentally differentiates KAN from conventional MLPs, and similarly distinguishes SepNNs from their non-separable counterparts.

### A.1.4   MODULATION FUNCTION CHOICE

We conducted a sensitivity analysis on the modulation function $g(\lambda)$. Specifically, we compared two forms of $g(\lambda)$: $g_1(\lambda_i) = \lambda_k$ for $i \leq k$, and $g_1(\lambda_i) = \lambda_i$ for $i > k$ (the version used in the experiment); $g_2(\lambda_i) = \sqrt{\lambda_i}$ for $i \leq k$, and $g_2(\lambda_i) = \lambda_i$ for $i > k$. We also varied the number of modulated eigenvalues $k$ to assess sensitivity. The results in Table 2 below demonstrate that SepPGD remains effective under both modulation functions, and $g_1(\lambda)$ yields better performance. Moreover, the method shows robustness to the choice of $k$ within a certain range (e.g., $k \in (60, 90)$). These findings indicate that with a suitable modulation function that smooths the eigenvalue distribution, the proposed SepPGD method is not overly sensitive to the specific form of $g(\lambda)$.

Table 2: PSNR performance on image representation using INRs with different modulation functions.

| $k$ | 30 | 40 | 50 | 60 | 70 | 80 | 90 | 100 |
|---|---|---|---|---|---|---|---|---|
| w/o SepPGD | 26.48 | 26.48 | 26.48 | 26.48 | 26.48 | 26.48 | 26.48 | 26.48 |
| $g_1(\lambda)$ | 26.79 | 26.95 | 29.65 | 33.04 | 33.30 | 34.28 | 34.21 | 34.06 |
| $g_2(\lambda)$ | 26.92 | 27.70 | 27.91 | 30.81 | 31.58 | 31.74 | 31.95 | 30.23 |

Table 3: Sensitivity analysis (PSNR) for rank $R$ in the SepNN and $k$ in the eigenvalue modulation function for image representation on the original sized *Plane* image.

| Rank $R$ | 100 | 200 | 300 | 400 | 500 | 600 | 700 | $k$ | | 30 | 40 | 50 | 60 | 70 | 80 | 90 |
|---|---|---|---|---|---|---|---|---|---|---|---|---|---|---|---|---|
| SepNN | 26.36 | 26.74 | 26.54 | 26.61 | 26.48 | 26.49 | 26.45 | SepNN | | 26.48 | 26.48 | 26.48 | 26.48 | 26.48 | 26.48 | 26.48 |
| SepNN (SepPGD) | 31.59 | 32.69 | 33.41 | 33.65 | 33.30 | 33.54 | 33.40 | SepNN (SepPGD) | | 26.79 | 26.95 | 29.65 | 33.04 | 33.30 | 34.28 | 34.21 |

### A.1.5 HYPERPARAMETER SENSITIVITY AND MORE EXPERIMENTS

We conduct sensitivity analysis on two critical hyperparameters: the rank $R$ in SepNN and the parameter $k$ in the modulation function $g(\lambda)$ used in the SepPGD algorithm. The image representation results are summarized in Table 3. For varying ranks $R$, our SepPGD consistently enhances the performance of SepNN. We find that setting $k > 60$ in this case effectively mitigates the spectral bias in SepNN by adjusting the NTK spectrum. In particular, values of $k \in \{60, 70, 80, 90\}$ produce a desirable effect for enhancing convergence efficiency. Our SepPGD method is not overly sensitive to the choice of $k$.

The sensitivity analysis of our SepPGD w.r.t. the preconditioner update frequency is shown in the Table 4 below. SepPGD is relatively not sensitive to the preconditioner update frequency. This is because the NTK matrix would not change abruptly during training and thus the preconditioner is effective for NTK eigenvalue modulation across a number of iterations.

Table 4: PSNR performance on image representation using INRs with different preconditioner update frequency (number of iterations between preconditioner updates).

| Update frequency | 10 | 100 | 300 | 500 | 700 | 900 | 1100 |
|---|---|---|---|---|---|---|---|
| SepNN | 26.48 | 26.48 | 26.48 | 26.48 | 26.48 | 26.48 | 26.48 |
| SepNN (SepPGD) | 33.30 | 33.29 | 33.16 | 33.12 | 33.09 | 33.06 | 33.06 |

The performance of SepPGD with different activation functions as well as the Fourier feature mapping (Tancik et al., 2020) is shown in the Table 5 below. SepPGD improves the convergence speed of the SepNN baseline with different activation functions and Fourier encoding structures.

Table 5: PSNR performance on image representation using INRs with different activation function/network structure.

| Activation | Sin | Cos | ReLU | LeakyReLU | Fourier feature +ReLU | Fourier feature+LeakyReLU |
|---|---|---|---|---|---|---|
| SepNN | 26.48 | 21.83 | 18.20 | 18.20 | 30.89 | 30.94 |
| SepNN (SepPGD) | 33.30 | 30.06 | 20.02 | 20.74 | 40.49 | 40.51 |

The performance with different width is shown in the Table 6 below. With a sufficient width (e.g., width> 200), the SepPGD is effective in accelerating the convergence of SepNN.

The experimental results for the image representation under additive Gaussian noise are shown in the Table 7 below. Under moderate to low noise deviation, SepPGD demonstrates robustness and improves the denoising performance of SepNN. However, when the noise deviation is large, SepPGD tends to overfit to the noise components. This behavior is expected, as SepPGD accelerates convergence toward high-frequency details, which can cause it to fit high-frequency noise more rapidly

Table 6: PSNR performance on image representation using INRs with different network width.

| Width | 100 | 200 | 300 | 400 | 500 | 600 | 700 |
|---|---|---|---|---|---|---|---|
| SepNN | 22.46 | 23.88 | 25.21 | 26.19 | 26.48 | 27.08 | 27.20 |
| SepNN (SepPGD) | 22.79 | 31.46 | 32.75 | 33.22 | 33.30 | 33.89 | 34.35 |

when the noise level is high and no explicit regularizations are applied. In future work, we can introduce suitable regularizations (e.g., total variation) to enhance robustness in noisy scenarios.

Table 7: PSNR performance on image representation using INRs with Gaussian noise.

| Noise standard deviation | 0.01 | 0.03 | 0.05 | 0.07 | 0.09 |
|---|---|---|---|---|---|
| SepNN | 26.40 | 26.35 | 26.27 | 25.85 | 25.29 |
| SepNN (SepPGD) | 33.26 | 31.08 | 28.20 | 26.35 | 25.06 |

Moreover, we employ SVD truncation to synthesize images with varying ranks, and then apply a SepNN model with a fixed rank of $R = 100$ to represent these images via INRs. The relationship between the approximation error (MSE) and the true separation rank of the target image is given in the Table 8 below. It can be observed that as the image rank increases, the approximation error of SepNN also grows. In contrast, SepPGD effectively enhances convergence efficiency (and consequently reduces the approximation error) across different data ranks.

Table 8: Approximation error (MSE) vs. true separation rank of the target image data (Setting the model rank $R = 100$) using a bivariate CP SepNN.

| Data rank | 100 | 200 | 300 | 400 | 500 |
|---|---|---|---|---|---|
| SepNN | 0.052 | 0.061 | 0.063 | 0.064 | 0.066 |
| SepNN (SepPGD) | 0.021 | 0.030 | 0.035 | 0.036 | 0.037 |

## A.2 EXPERIMENTS ON NON-GRID INPUTS

We further conducted experiments on non-grid tasks using INRs. Specifically, we randomly sampled 500 non-grid points along with their corresponding function values $(f_1(x, y), f_2(x, y), f_3(x, y)$ from Eqs. (16)-(18)) to form the non-grid training dataset. We then train a SepNN to fit these non-grid samples, as shown in Table 9 below. For non-grid data, the proposed SepPGD gradient is formulated as $\nabla_{\Theta_d} \langle f_{\Theta_d}((\boldsymbol{X})_{d,:}), \boldsymbol{M}_d \rangle$, where $\boldsymbol{X} \in \mathbb{R}^{D \times n}$ are non-grid input points and $\boldsymbol{M}_d$ is constructed by some Einstein products between factor outputs, preconditioner $\boldsymbol{S}_d$, and training residual $\boldsymbol{r}$ (especially, $\boldsymbol{M}_d = \mathrm{einsum}(\odot_{d' \neq d} f_{\Theta'_d}((\boldsymbol{X})_{d',:}), \sum_d \boldsymbol{S}_d \boldsymbol{r}; (R, n) \times (n) \to (R, n))$, where $\odot$ denotes element-wise product and $\mathrm{einsum}$ denotes the Einstein product between matrix and vector). This formulation reduces to point-wise evaluations and yields similar efficiency and effectiveness to classical NTK-based preconditioning methods, such as the MSK (Geifman et al., 2024). Therefore, it is reasonable that SepNN (SepPGD) performs comparably to SepNN (MSK) on non-grid data, while both outperform the original SepNN. The efficiency advantage of SepPGD comes primarily from the separable structure of grid inputs, which is a prevalent training configuration in applications such as image grids in INRs and grid collocation points in PINNs.

Table 9: MSE performance of SepPGD for non-grid data $f_1(x, y), f_2(x, y), f_3(x, y)$ by learning function representations using INRs.

| Data | $f_1(x, y)$ | $f_2(x, y)$ | $f_3(x, y)$ |
|---|---|---|---|
| SepNN | 0.020 | 0.030 | 0.075 |
| SepNN (MSK) | 0.014 | 0.024 | 0.058 |
| SepNN (SepPGD) | 0.015 | 0.021 | 0.062 |

### A.3 EFFICIENT NTK MATRIX COMPUTATION

Another merit of the SepNN's NTK is that its NTK matrix can be computed in parallel over multiple inputs, resulting in greater efficiency compared to computing the NTK matrices of other neural architectures (Novak et al., 2022; Mohamadi et al., 2023). In fact, the SepNN's NTK (12) admits an elegant form that allows the NTK matrix over grid inputs to be expressed as a Kronecker product of smaller NTK matrices (Lemma 3), thereby improving efficiency. For simplicity, the NTK matrix derivation is presented for the bivariate case $D = 2$ because the NTK matrix formulation for multivariate cases $D > 2$ would be much more lengthy.

**Lemma 3** (Exact NTK matrix of SepNN). *Let the bivariate CP SepNN be* $f_\Theta(x_1, x_2) = \frac{1}{\sqrt{R}} \sum_{r=1}^{R} (f_{\Theta_1}(x_1))_r (f_{\Theta_2}(x_2))_r$, *where each* $f_{\Theta_d} : \mathbb{R} \to \mathbb{R}^R$ ($d = 1, 2$) *is a parametric MLP with parameters* $\Theta_d$, *and* $\Theta = (\Theta_1, \Theta_2)$. *Consider the grid input points* $\{(x_1, x_2) | x_d \in \hat{\boldsymbol{x}}_d, d = 1, 2\}$ *where* $\hat{\boldsymbol{x}}_d \in \mathbb{R}^n$. *The NTK matrix* $\boldsymbol{K} \in \mathbb{R}^{n^2 \times n^2}$ *of* $f_\Theta$ *over these grid points is given by:*

$$\boldsymbol{K} = \frac{1}{R} \sum_{r,s=1}^{R} \left( \boldsymbol{K}_1^{(r,s)} \otimes \left( (f_{\Theta_2}(\hat{\boldsymbol{x}}_2))_{r,:}^\top (f_{\Theta_2}(\hat{\boldsymbol{x}}_2))_{s,:} \right) + \left( (f_{\Theta_1}(\hat{\boldsymbol{x}}_1))_{r,:}^\top (f_{\Theta_1}(\hat{\boldsymbol{x}}_1))_{s,:} \right) \otimes \boldsymbol{K}_2^{(r,s)} \right), \quad (9)$$

*where* $(f_{\Theta_d}(\hat{\boldsymbol{x}}_d))_{r,:} \in \mathbb{R}^{1 \times n}$ *denotes the* $r$*-th row of the output factor matrix* $f_{\Theta_d}(\hat{\boldsymbol{x}}_d) \in \mathbb{R}^{R \times n}$ *(* $f_{\Theta_d}$ *takes each element in* $\hat{\boldsymbol{x}}_d$ *as input in parallel) and each smaller NTK matrix* $\boldsymbol{K}_d^{(r,s)} \in \mathbb{R}^{n \times n}$ *(* $d = 1, 2$ *) is defined by* $\left( \boldsymbol{K}_d^{(r,s)} \right)_{i,j} = \left\langle \nabla_{\Theta_d} \left( f_{\Theta_d}((\hat{\boldsymbol{x}}_d)_i) \right)_r, \nabla_{\Theta_d} \left( f_{\Theta_d}((\hat{\boldsymbol{x}}_d)_j) \right)_s \right\rangle$.

The proof is in Section A.10. Lemma 3 shows that computing the NTK matrix of the SepNN on grid points only requires computing some Kronecker products between smaller matrices of size $n \times n$, compared to computing the direct $n^2 \times n^2$ NTK matrix needed by other neural architectures for $n^2$ grid points. Furthermore, we can further decrease the complexity using pseudo NTK. While computing the exact NTK matrices $\{\boldsymbol{K}_d^{(r,s)}\}$ of multi-output factor MLPs $f_{\Theta_d}$ requires $O(nR)$ complexity due to the $R$ outputs, researchers usually consider computing the pseudo NTK matrix (Mohamadi et al., 2023; Novak et al., 2022) instead. In this work, we leverage the rank-one sum-of-logits approximation to approximate the NTK matrix of the multi-output factor MLP $f_{\Theta_d}$, reducing complexity to $O(n)$. The pseudo NTK of the bivariate SepNN $f_\Theta$ is formulated as

$$\boldsymbol{K} \approx \frac{1}{R} \left( \boldsymbol{K}_1 \otimes \left( f_{\Theta_2}(\hat{\boldsymbol{x}}_2)^\top f_{\Theta_2}(\hat{\boldsymbol{x}}_2) \right) + \left( f_{\Theta_1}(\hat{\boldsymbol{x}}_1)^\top f_{\Theta_1}(\hat{\boldsymbol{x}}_1) \right) \otimes \boldsymbol{K}_2 \right), \quad (10)$$

where $\boldsymbol{K}_1, \boldsymbol{K}_2 \in \mathbb{R}^{n \times n}$ are pseudo NTK matrices of the multi-output factor MLPs $f_{\Theta_1}, f_{\Theta_2} : \mathbb{R} \to \mathbb{R}^R$ on training points $\hat{\boldsymbol{x}}_1 \in \mathbb{R}^n$ and $\hat{\boldsymbol{x}}_2 \in \mathbb{R}^n$, computed using sum-of-logits (Mohamadi et al., 2023). The relative approximation error between the pseudo NTK matrix (10) and the exact NTK matrix (9), which is a simple weighted addition of the errors induced by $\boldsymbol{K}_1, \boldsymbol{K}_2$, would scale as $O(\sqrt{W})$ (network width) by using analogous arguments as in (Mohamadi et al., 2023).

### A.4 FIXED NTK DURING TRAINING

**Theorem 3** (SepNN NTK during training). *Let the multivariate CP SepNN be defined as* $f_\Theta(x_1, \cdots, x_D) = \frac{1}{\sqrt{R}} \sum_{r=1}^{R} (f_{\Theta_1}(x_1))_r (f_{\Theta_2}(x_2))_r \cdots (f_{\Theta_D}(x_D))_r : \mathbb{R}^D \to \mathbb{R}$, *where each* $f_{\Theta_d} : \mathbb{R} \to \mathbb{R}^R$ *is a parametric MLP with parameters* $\Theta_d$, *and* $\Theta = (\Theta_1, \cdots, \Theta_D)$ *is the collection of all parameters. Let each factor MLP* $f_{\Theta_d} : \mathbb{R} \to \mathbb{R}^R$ *has the architecture* $f_{\Theta_d}(x_d) = \frac{1}{\sqrt{W}} \boldsymbol{W}_{2,d} \sigma(\boldsymbol{W}_{1,d} x_d + \boldsymbol{b}_d)$, *with* $\boldsymbol{W}_{1,d} \in \mathbb{R}^{W \times 1}$, $\boldsymbol{b}_d \in \mathbb{R}^W$, $\boldsymbol{W}_{2,d} \in \mathbb{R}^{R \times W}$, *and* $\sigma$ *a differentiable activation function with derivative* $\dot{\sigma}$. *Let* $\sigma, \dot{\sigma}$ *be Lipschitz smooth. Consider* $n$ *training samples* $\{\boldsymbol{x}_i, y_i\}_{i=1}^{n}$. *Assume the weight (e.g.,* $\boldsymbol{W}_{1,d}$*) are initialized by i.i.d.* $\mathcal{N}(0, 1)$. *Let the input norm* $\|\boldsymbol{x}_i\|_{\ell_2}$ *and label* $y_i$ *being uniformly bounded by some constants* $O(1)$. *Consider standard gradient descent optimizer with infinitely small learning rate. Then, given a training time* $t > 0$, *as both the width* $W \to \infty$ *and the rank* $R \to \infty$, *the NTK of* $f_\Theta$ *during training converges to the NTK at initialization* $K_{\Theta(t)}(\boldsymbol{x}_i, \boldsymbol{x}_j) \to K_{\Theta(0)}(\boldsymbol{x}_i, \boldsymbol{x}_j)$ *for all* $\boldsymbol{x}_i, \boldsymbol{x}_j \in \{\boldsymbol{x}_i\}_{i=1}^{n}$.

The proof of Theorem 3 is placed in Section A.11. The NTK matrix of SepNN during training converges to the NTK matrix at initialization, meaning that the training dynamic of SepNN under infinite width and rank can be characterized by the corresponding linear kernel regression associated

with its NTK, allowing the convergence analysis in Eq. (5) to be true. It is also possible to bound the NTK change $|K_{\Theta(t)} - K_{\Theta(0)}|$ by Chebyshev-type inequality through leveraging the variance of $|K_{\Theta(t)} - K_{\Theta(0)}|$ related to $O(\frac{t}{R})$ with a fixed $R$ and small step sizes. Indeed, when $W, R \to \infty$, the variance vanishes and $K_{\Theta(t)}$ converges to $K_{\Theta(0)}$ almost surely under infinitely small step sizes. With a fixed $R$ and small step sizes, we can instead use the Chebyshev inequality to quantify the error $|K_{\Theta(t)} - K_{\Theta(0)}|$ using probability bounds related to $R$ and the step size.

### A.5  Proof of Theorem 1

*Proof.* The proof consists of two main steps, including demonstrating the density of the separable function classes (CP, TT, and Tucker), and approximating the univariate functions in the decompositions using neural networks.

We first demonstrate the density of separable function classes using the Stone-Weierstrass theorem. Consider the algebra $\mathcal{A}$ of functions on $\mathcal{X} = \mathcal{X}_1 \times \cdots \times \mathcal{X}_N$ defined by

$$\mathcal{A} = \left\{ g : \mathcal{X} \to \mathbb{R} : g(x_1, \cdots, x_N) = \sum_{r=1}^{R} (g_1(x_1))_r (g_2(x_2))_r \cdots (g_D(x_D))_r, R \in \mathbb{N}, (g_d(x_d))_r \in C(\mathcal{X}_d) \right\}.$$

We show that this algebra is closed under addition, multiplication, and scalar multiplication.

**Closed Under Multiplication.** Let two functions $g, h \in \mathcal{A}$ be given by

$$g(\boldsymbol{x}) = \sum_{r=1}^{R_g} \prod_{d=1}^{D} (g_d(x_d))_r, \ h(\boldsymbol{x}) = \sum_{s=1}^{R_h} \prod_{d=1}^{D} (h_d(x_d))_s \in \mathcal{A},$$

where $(g_d(x_d))_r, (h_{d,s}(x_d))_r \in C(\mathcal{X}_d)$ are continuous univariate functions. Their product is

$$g(\boldsymbol{x})h(\boldsymbol{x}) = \left( \sum_{r=1}^{R_g} \prod_{d=1}^{D} (g_d(x_d))_r \right) \left( \sum_{s=1}^{R_h} \prod_{d=1}^{D} (h_d(x_d))_s \right) = \sum_{r=1}^{R_g} \sum_{s=1}^{R_h} \prod_{d=1}^{D} (g_d(x_d))_r (h_d(x_d))_s.$$

Define new rank $R = R_g R_h$. For each $d$ and each index pair $t = (r, s)$ (where $t = 1, \cdots, R$), define new continuous functions

$$(c_d(x_d))_t = (g_d(x_d))_r (h_d(x_d))_s.$$

Then we have

$$g(\boldsymbol{x})h(\boldsymbol{x}) = \sum_{t=1}^{R} \prod_{d=1}^{D} (c_d(x_d))_t,$$

which is clearly in $\mathcal{A}$. Thus, the CP form is closed under multiplication.

Meanwhile, let two functions $g$ and $h$ have TT decompositions:

$$g(\boldsymbol{x}) = \sum_{r_1=1}^{R_1^g} \cdots \sum_{r_{D-1}=1}^{R_{D-1}^g} g_1(x_1)_{1,r_1} g_2(x_2)_{r_1,r_2} \cdots g_D(x_D)_{r_{D-1},1},$$

$$h(\boldsymbol{x}) = \sum_{s_1=1}^{R_1^h} \cdots \sum_{s_{D-1}=1}^{R_{D-1}^h} h_1(x_1)_{1,s_1} h_2(x_2)_{s_1,s_2} \cdots h_D(x_D)_{s_{D-1},1},$$

where $g_d(x_d) \in \mathbb{R}^{R_{d-1}^g \times R_d^g}$ and $h_d(x_d) \in \mathbb{R}^{R_{d-1}^h \times R_d^h}$ are matrix-valued continuous functions with $R_0^g = R_D^g = 1$ and $R_0^h = R_D^h = 1$. Their product is

$$g(\boldsymbol{x})h(\boldsymbol{x}) = \sum_{r_1, \cdots, r_{D-1}} \sum_{s_1, \cdots, s_{D-1}} (g_1(x_1)_{1,r_1} h_1(x_1)_{1,s_1}) (g_2(x_2)_{r_1,r_2} h_2(x_2)_{s_1,s_2}) \cdots$$

$$(g_D(x_D)_{r_{D-1},1} h_D(x_D)_{s_{D-1},1}).$$

Define new ranks $T_d = R_d^g R_d^h$ for $d = 1, \cdots, D-1$, and define new matrix-valued functions $f_d(x_d)$ as follows. For $d = 1$, $f_1(x_1)$ is a $1 \times T_1$ matrix with elements

$$f_1(x_1)_{1,t_1} = g_1(x_1)_{1,r_1} h_1(x_1)_{1,s_1},$$

where $t_1 = r_1 s_1$. For $d = 2, \cdots, D-1$, $f_d(x_d)$ is a $T_{d-1} \times T_d$ matrix with elements

$$f_d(x_d)_{t_{d-1}, t_d} = g_d(x_d)_{r_{d-1}, r_d} h_d(x_d)_{s_{d-1}, s_d},$$

where $t_{d-1} = r_{d-1} s_{d-1}$ and $t_d = r_d s_d$. For $d = D$, $f_D(x_D)$ is a $T_{D-1} \times 1$ matrix with elements

$$f_D(x_D)_{t_{D-1}, 1} = g_D(x_D)_{r_{D-1}, 1} h_D(x_D)_{s_{D-1}, 1}.$$

Then we have

$$g(\boldsymbol{x}) h(\boldsymbol{x}) = \sum_{t_1=1}^{T_1} \cdots \sum_{t_{D-1}=1}^{T_{D-1}} f_1(x_1)_{1, t_1} f_2(x_2)_{t_1, t_2} \cdots f_D(x_D)_{t_{D-1}, 1},$$

which is in TT form. Thus, the TT form is closed under multiplication.

Let two functions $g$ and $h$ have Tucker decompositions:

$$g(\boldsymbol{x}) = \mathcal{C}^g \times_1 g_1(x_1) \times_2 \cdots \times_D g_D(x_D) = \sum_{r_1, \cdots, r_D} \mathcal{C}^g_{r_1, \cdots, r_D} \prod_{d=1}^{D} (g_d(x_d))_{r_d},$$

$$h(\boldsymbol{x}) = \mathcal{C}^h \times_1 h_1(x_1) \times_2 \cdots \times_D h_D(x_D) = \sum_{s_1, \cdots, s_D} \mathcal{C}^h_{s_1, \cdots, s_D} \prod_{d=1}^{D} (h_d(x_d))_{s_d},$$

where $g_d(x_d) \in \mathbb{R}^{R_d^g}$, $h_d(x_d) \in \mathbb{R}^{R_d^h}$ are vector-valued continuous functions, and $\mathcal{C}^g \in \mathbb{R}^{R_1^g \times \cdots \times R_D^g}$, $\mathcal{C}^h \in \mathbb{R}^{R_1^h \times \cdots \times R_D^h}$ are core tensors. Their product is

$$g(\boldsymbol{x}) h(\boldsymbol{x}) = \sum_{r_1, \cdots, r_D} \sum_{s_1, \cdots, s_D} \mathcal{C}^g_{r_1, \cdots, r_D} \mathcal{C}^h_{s_1, \cdots, s_D} \prod_{d=1}^{D} (g_d(x_d))_{r_d} (h_d(x_d))_{s_d}.$$

Define new ranks $T_d = R_d^g R_d^h$ for $d = 1, \cdots, D$, and define a new core tensor $\mathcal{C}^{gh} \in \mathbb{R}^{T_1 \times \cdots \times T_D}$ with elements

$$\mathcal{C}^{gh}_{t_1, \cdots, t_D} = \mathcal{C}^g_{r_1, \cdots, r_D} \mathcal{C}^h_{s_1, \cdots, s_D},$$

where $t_d = r_d s_d$ for each $d$. Define new vector-valued functions $f_d(x_d) \in \mathbb{R}^{T_d}$ with elements

$$(f_d(x_d))_{t_d} = (g_d(x_d))_{r_d} (h_d(x_d))_{s_d}.$$

Then we have

$$g(\boldsymbol{x}) h(\boldsymbol{x}) = \mathcal{C}^{gh} \times_1 f_1(x_1) \times_2 \cdots \times_D f_D(x_D),$$

which is in Tucker form. Thus, the Tucker form is closed under multiplication.

**Closed Under Addition.** We verify that the function classes defined by CP, TT, and Tucker decompositions are closed under addition. Let two functions $g, h \in \mathcal{A}$ be

$$g(\boldsymbol{x}) = \sum_{r=1}^{R_g} \prod_{d=1}^{D} (g_d(x_d))_r, \quad h(\boldsymbol{x}) = \sum_{s=1}^{R_h} \prod_{d=1}^{D} (h_d(x_d))_s \in \mathcal{A},$$

where $(g_d(x_d))_r, (h_d(x_d))_s \in C(\mathcal{X}_d)$ are continuous univariate functions. Their sum is

$$g(\boldsymbol{x}) + h(\boldsymbol{x}) = \sum_{r=1}^{R_g} \prod_{d=1}^{D} (g_d(x_d))_r + \sum_{s=1}^{R_h} \prod_{d=1}^{D} (h_d(x_d))_s.$$

Define new rank $R = R_g + R_h$ and for each $d$, define new continuous functions $f_d : \mathcal{X}_d \to \mathbb{R}^R$ with components

$$(f_d(x_d))_r = \begin{cases} (g_d(x_d))_r, & r = 1, \cdots, R_g, \\ (h_d(x_d))_{r-R_g}, & r = R_g + 1, \cdots, R. \end{cases}$$

Then we have

$$g(\boldsymbol{x}) + h(\boldsymbol{x}) = \sum_{r=1}^{R} \prod_{d=1}^{D} (f_d(x_d))_r,$$

which is clearly in $\mathcal{A}$. Thus, the CP form is closed under addition.

Let two functions $g$ and $h$ have TT decompositions

$$g(\boldsymbol{x}) = \sum_{r_1=1}^{R_1^g} \cdots \sum_{r_{D-1}=1}^{R_{D-1}^g} g_1(x_1)_{1,r_1} g_2(x_2)_{r_1,r_2} \cdots g_D(x_D)_{r_{D-1},1},$$

$$h(\boldsymbol{x}) = \sum_{s_1=1}^{R_1^h} \cdots \sum_{s_{D-1}=1}^{R_{D-1}^h} h_1(x_1)_{1,s_1} h_2(x_2)_{s_1,s_2} \cdots h_D(x_D)_{s_{D-1},1},$$

where $g_d(x_d) \in \mathbb{R}^{R_{d-1}^g \times R_d^g}$ and $h_d(x_d) \in \mathbb{R}^{R_{d-1}^h \times R_d^h}$ are matrix-valued continuous functions with $R_0^g = R_D^g = 1$ and $R_0^h = R_D^h = 1$. Their sum is $g(\boldsymbol{x}) + h(\boldsymbol{x})$. Define new ranks $T_d = R_d^g + R_d^h$ for $d = 1, \cdots, D-1$, and define new matrix-valued functions $f_d(x_d)$ as follows. For $d = 1$, $f_1(x_1)$ is a $1 \times T_1$ matrix with elements

$$f_1(x_1)_{1,t} = \begin{cases} g_1(x_1)_{1,t}, & t = 1, \cdots, R_1^g, \\ h_1(x_1)_{1,t-R_1^g}, & t = R_1^g + 1, \cdots, T_1. \end{cases}$$

For $d = 2, \cdots, D-1$, $f_d(x_d)$ is a $T_{d-1} \times T_d$ matrix with elements

$$f_d(x_d)_{t,u} = \begin{cases} g_d(x_d)_{t,u}, & t \le R_{d-1}^g, u \le R_d^g, \\ h_d(x_d)_{t-R_{d-1}^g, u-R_d^g}, & t > R_{d-1}^g, u > R_d^g, \\ 0, & \text{otherwise}. \end{cases}$$

For $d = D$, $f_D(x_D)$ is a $T_{D-1} \times 1$ matrix with elements

$$f_D(x_D)_{t,1} = \begin{cases} g_D(x_D)_{t,1}, & t \le R_{D-1}^g, \\ h_D(x_D)_{t-R_{D-1}^g, 1}, & t > R_{D-1}^g. \end{cases}$$

Then we have

$$g(\boldsymbol{x}) + h(\boldsymbol{x}) = \sum_{t_1=1}^{T_1} \cdots \sum_{t_{D-1}=1}^{T_{D-1}} f_1(x_1)_{1,t_1} f_2(x_2)_{t_1,t_2} \cdots f_D(x_D)_{t_{D-1},1},$$

which is in TT form. Thus, the TT form is closed under addition.

Let two functions $g$ and $h$ have Tucker decompositions

$$g(\boldsymbol{x}) = \mathcal{C}^g \times_1 g_1(x_1) \times_2 \cdots \times_D g_D(x_D), \ h(\boldsymbol{x}) = \mathcal{C}^h \times_1 h_1(x_1) \times_2 \cdots \times_D h_D(x_D),$$

where $\mathcal{C}^g \in \mathbb{R}^{R_1^g \times \cdots \times R_D^g}$, $\mathcal{C}^h \in \mathbb{R}^{R_1^h \times \cdots \times R_D^h}$, and $g_d(x_d) \in \mathbb{R}^{R_d^g}$, $h_d(x_d) \in \mathbb{R}^{R_d^h}$ are vector-valued continuous functions. Their sum is $g(\boldsymbol{x}) + h(\boldsymbol{x})$. Define new ranks $T_d = R_d^g + R_d^h$ for $d = 1, \cdots, D$, and define a new core tensor $\mathcal{C} \in \mathbb{R}^{T_1 \times \cdots \times T_D}$ with elements

$$\mathcal{C}_{i_1, \cdots, i_D} = \begin{cases} \mathcal{C}_{i_1, \cdots, i_D}^g, & \text{if } i_d \le R_d^g \text{ for all } d, \\ \mathcal{C}_{i_1-R_1^g, \cdots, i_D-R_D^g}^h, & \text{if } i_d > R_d^g \text{ for all } d, \\ 0, & \text{otherwise}. \end{cases}$$

Defined new vector-valued functions $f_d(x_d) \in \mathbb{R}^{T_d}$ with elements

$$(f_d(x_d))_r = \begin{cases} (g_d(x_d))_r, & r = 1, \cdots, R_d^g, \\ (h_d(x_d))_{r-R_d^g}, & r = R_d^g + 1, \cdots, T_d. \end{cases}$$

Then we have

$$g(\boldsymbol{x}) + h(\boldsymbol{x}) = \mathcal{C} \times_1 f_1(x_1) \times_2 \cdots \times_D f_D(x_D),$$

which is in Tucker form. Thus, the Tucker form is closed under addition.

**Closed Under Scalar Multiplication.** For any CP, Tucker, or TT functions, it is easy to show that multiplying them with a scalar are still CP, Tucker, or TT functions. Hence the separable function classes are closed under scalar multiplication.

**Separate Points.** We further verify that the function classes defined by CP, TT, and Tucker decompositions separate points. That is, for any two distinct points $\boldsymbol{p} \ne \boldsymbol{q}$ in the compact set

$\mathcal{X} = \mathcal{X}_1 \times \cdots \times \mathcal{X}_D$, there exists a function $g$ in the respective class such that $g(\boldsymbol{p}) \neq g(\boldsymbol{q})$. Consider the CP form:

$$g(\boldsymbol{x}) = \sum_{r=1}^{R} \prod_{d=1}^{D} (g_d(x_d))_r.$$

Since $\boldsymbol{p} \neq \boldsymbol{q}$, there exists some dimension $d_0$ such that $p_{d_0} \neq q_{d_0}$. Choose $R = 1$ and define the functions such that for $d = d_0$, $g_{d_0} : \mathcal{X}_{d_0} \to \mathbb{R}$ to be a continuous function such that $g_{d_0}(p_{d_0}) \neq g_{d_0}(q_{d_0})$ (such a function exists since continuous functions can separate points in one dimension) and for $d \neq d_0$, $g_d(x_d) = 1$. Then we have

$$g(\boldsymbol{x}) = \prod_{d=1}^{D} g_d(x_d)$$

and

$$g(\boldsymbol{p}) = g_{d_0}(p_{d_0}) \prod_{d \neq d_0} 1 \neq g_{d_0}(q_{d_0}) \prod_{d \neq d_0} 1 = g(\boldsymbol{q}).$$

Thus, the CP form separates points.

Consider the TT form:

$$g(\boldsymbol{x}) = \sum_{r_1=1}^{R_1} \cdots \sum_{r_{D-1}=1}^{R_{D-1}} (g_1(x_1))_{1,r_1} (g_2(x_2))_{r_1,r_2} \cdots (g_D(x_D))_{r_{D-1},1}.$$

Choose all ranks $R_d = 1$ for $d = 1, \cdots, D - 1$. Then the TT form simplifies to

$$g(\boldsymbol{x}) = (g_1(x_1))_{1,1} (g_2(x_2))_{1,1} \cdots (g_D(x_D))_{1,1}.$$

Since $\boldsymbol{p} \neq \boldsymbol{q}$, there exists $d_0$ such that $p_{d_0} \neq q_{d_0}$. Choose $g_{d_0}$ such that $(g_{d_0}(p_{d_0}))_{1,1} \neq (gd_0(q_{d_0}))_{1,1}$, and for other $d$, choose $g_d$ to be the constant function $(g_d(x_d))_{1,1} = 1$. Then we have

$$g(\boldsymbol{p}) \neq g(\boldsymbol{q}). \tag{11}$$

Thus, the TT form separates points.

Note that CP is a special case of Tucker by setting the core tensor $\mathcal{C}$ as diagonal. Since the CP function class separates points, the Tucker function class separates points as well.

**Contain Identity Functions.** We verify that the function classes defined by CP, TT, and Tucker decompositions contain identity functions (constant functions). Consider the CP form:

$$g(\boldsymbol{x}) = \sum_{r=1}^{R} \prod_{d=1}^{D} (g_d(x_d))_r.$$

To obtain a constant function $g(\boldsymbol{x}) = c$ for some constant $c \in \mathbb{R}$, we can choose $R = 1$, $g_d(x_d) = c_d$ where $c_d$ is a constant for each $d = 1, \cdots, D$ such that $\prod_{d=1}^{D} c_d = c$. Then we have

$$g(\boldsymbol{x}) = \prod_{d=1}^{D} c_d = c.$$

Thus, the CP form contains identity functions.

Consider the TT form:

$$g(\boldsymbol{x}) = \sum_{r_1=1}^{R_1} \cdots \sum_{r_{D-1}=1}^{R_{D-1}} (g_1(x_1))_{1,r_1} (g_2(x_2))_{r_1,r_2} \cdots (g_D(x_D))_{r_{D-1},1}.$$

To obtain a constant function $g(\boldsymbol{x}) = c$, we can choose all ranks $R_d = 1$ for $d = 1, \cdots, D - 1$, and for each $d = 1, \cdots, D$, $g_d(x_d)$ to be a constant matrix with all elements equal to $c_d$ such that $\prod_{d=1}^{D} c_d = c$. Then we have

$$g(\boldsymbol{x}) = \prod_{d=1}^{D} c_d = c.$$

Thus, the TT form contains identity functions.

Note that CP is a special case of Tucker, hence the Tucker function class contains the identity function as well.

By the Stone-Weierstrass theorem (Fedorova, 2002), the CP function class $\mathcal{A}$ is dense in $C(\mathcal{X})$ (as well as the TT and Tucker function classes). Therefore, for any continuous multivariate function $f$ and $\frac{\epsilon}{2} > 0$, there exists a CP function $\sum_{r=1}^{R} (g_1(x_1))_r (g_2(x_2))_r \cdots (g_D(x_D))_r$ such that

$$\sup_{\boldsymbol{x} \in \mathcal{X}} \left| f(\boldsymbol{x}) - \sum_{r=1}^{R} (g_1(x_1))_r (g_2(x_2))_r \cdots (g_D(x_D))_r \right| < \frac{\epsilon}{2}.$$

Likewise, there exist TT function $\sum_{r_1=1}^{R_1} \sum_{r_2=1}^{R_2} \cdots \sum_{r_{D-1}=1}^{R_{D-1}} (g_1(x_1))_{1,r_1} (g_2(x_2))_{r_1,r_2} \cdots (g_D(x_D))_{r_{D-1},1}$ and Tucker function $\mathcal{C} \times_1 g_1(x_1) \times_2 \cdots \times_D g_D(x_D)$ such that

$$\sup_{\boldsymbol{x} \in \mathcal{X}} \left| f(\boldsymbol{x}) - \sum_{r_1=1}^{R_1} \sum_{r_2=1}^{R_2} \cdots \sum_{r_{D-1}=1}^{R_{D-1}} (g_1(x_1))_{1,r_1} (g_2(x_2))_{r_1,r_2} \cdots (g_D(x_D))_{r_{D-1},1} \right| < \frac{\epsilon}{2},$$

$$\sup_{\boldsymbol{x} \in \mathcal{X}} \left| f(\boldsymbol{x}) - \mathcal{C} \times_1 g_1(x_1) \times_2 \cdots \times_D g_D(x_D) \right| < \frac{\epsilon}{2},$$

where each component of the factor function $g_d(x_d)$ (i.e., $(g_d(x_d))_r$) is a single-output continuous function defined on compact set $\mathcal{X}_d$. The standard universal approximation theory (Leshno et al., 1993) states that for any $\delta > 0$ and any single-output continuous function $(g_d(x_d))_r$ defined on compact sets, there exists an MLP $(f'_{\Theta_d}(x_d))_r$ sampled in $\{\boldsymbol{W}_2\sigma(\boldsymbol{W}_1 x + \boldsymbol{b}) : \boldsymbol{W}_2 \in \mathbb{R}^{1 \times W}, \boldsymbol{W}_1 \in \mathbb{R}^{W \times 1}, \boldsymbol{b} \in \mathbb{R}^W, W \in \mathbb{N}_+\}$ with $\sigma$ a non-polynomial function such that

$$\sup_{x \in \mathcal{X}_d} |(g_d(x_d))_r - (f'_{\Theta_d}(x_d))_r| < \delta, \ r = 1, \cdots, R.$$

Let these $R$ single-output MLPs be $\{(f'_{\Theta_d}(x_d))_r = \boldsymbol{W}_2^r \sigma(\boldsymbol{W}_1^r x + \boldsymbol{b}^r)\}_{r=1}^{R}$. Therefore, we can construct a single $R$-output vector-valued MLP with the form

$$f_{\Theta_d}(x_d) = \boldsymbol{W}_2\sigma(\boldsymbol{W}_1 x + \boldsymbol{b}) \in \mathbb{R}^R,$$

where $\boldsymbol{W}_2 = \text{diag}(\boldsymbol{W}_2^1, \cdots, \boldsymbol{W}_2^R)$, $\boldsymbol{W}_1 = (\boldsymbol{W}_1^{1\top}, \cdots, \boldsymbol{W}_1^{R\top})^\top$, $\boldsymbol{b} = (\boldsymbol{b}^{1\top}, \cdots, \boldsymbol{b}^{R\top})^\top$ such that $\sup_{x \in \mathcal{X}_d} |(g_d(x_d))_r - (f_{\Theta_d}(x_d))_r| < \delta$ for each component $r = 1, \cdots, R$. We have thus extended the classical universal approximation theorem (Leshno et al., 1993) to vector-valued functions. Especially, each $g_d(x_d)$ can be well approximated (in a component-wise manner) by a vector-valued MLP $f_{\Theta_d} : \mathcal{X}_d \to \mathbb{R}^R$ sampled in the set $\{\boldsymbol{W}_2\sigma(\boldsymbol{W}_1 x + \boldsymbol{b}) : \boldsymbol{W}_2 \in \mathbb{R}^{\tilde{W} \times W}, \boldsymbol{W}_1 \in \mathbb{R}^{W \times 1}, \boldsymbol{b} \in \mathbb{R}^W, \tilde{W}, W \in \mathbb{N}_+\}$ with $\sigma$ a non-polynomial activation function.

For any continuous multivariate function $f : \mathcal{X} \to \mathbb{R}$, let the CP approximation be

$$g(\boldsymbol{x}) = \sum_{r=1}^{R} (g_1(x_1))_r (g_2(x_2))_r \cdots (g_D(x_D))_r,$$

such that $\sup_{\boldsymbol{x} \in \mathcal{X}} |f(\boldsymbol{x}) - g(\boldsymbol{x})| < \frac{\epsilon}{2}$. Let the CP SepNN approximation be:

$$f_\Theta(\boldsymbol{x}) = \sum_{r=1}^{R} (f_{\Theta_1}(x_1))_r (f_{\Theta_2}(x_2))_r \cdots (f_{\Theta_D}(x_D))_r,$$

with $\sup_{x_d \in \mathcal{X}_d} |(g_d(x_d))_r - (f_{\Theta_d}(x_d))_r| < \delta$ for $d = 1, \cdots, D$ and $r = 1, \cdots, R$. The total error is bounded by

$$|f(\boldsymbol{x}) - f_\Theta(\boldsymbol{x})| \leq |f(\boldsymbol{x}) - g(\boldsymbol{x})| + |g(\boldsymbol{x}) - f_\Theta(\boldsymbol{x})|$$

$$< \frac{\epsilon}{2} + \left| \sum_{r=1}^{R} \left( \prod_{d=1}^{D} (g_d(x_d))_r - \prod_{d=1}^{D} (f_{\Theta_d}(x_d))_r \right) \right|.$$

Since each $(g_d(x_d))_r$ and $(f_{\Theta_d}(x_d))_r$ is continuous on a compact set, they are bounded. Let $M$ be a common upper bound such that $|(g_d(x_d))_r| \leq M$ and $|(f_{\Theta_d}(x_d))_r| \leq M$ for all $d, r$, and $\boldsymbol{x} \in \mathcal{X}$. Using the generalized Cauchy-Schwarz inequality $\left| \prod_{d=1}^{D} a_d - \prod_{d=1}^{D} b_d \right| \leq \sum_{d=1}^{D} |a_d - b_d| \left( \prod_{j=1}^{d-1} |a_j| \right) \left( \prod_{j=d+1}^{D} |b_j| \right)$ we have that for each $r$:

$$\left| \prod_{d=1}^{D} (g_d(x_d))_r - \prod_{d=1}^{D} (f_{\Theta_d}(x_d))_r \right| \leq DM^{D-1} \max_d |(g_d(x_d))_r - (f_{\Theta_d}(x_d))_r| < DM^{D-1}\delta.$$

Thus we have

$$\left| \sum_{r=1}^{R} \left( \prod_{d=1}^{D} (g_d(x_d))_r - \prod_{d=1}^{D} (f_{\Theta_d}(x_d))_r \right) \right| < RDM^{D-1}\delta.$$

Choosing $\delta = \frac{\epsilon}{2RDM^{D-1}}$ we have

$$|f(\boldsymbol{x}) - f_\Theta(\boldsymbol{x})| < \frac{\epsilon}{2} + \frac{\epsilon}{2} = \epsilon,$$

and the approximation theory of CP SepNN is thus established.

For any continuous multivariate function $f : \mathcal{X} \to \mathbb{R}$, let the TT approximation be

$$g(\boldsymbol{x}) = \sum_{r_1=1}^{R_1} \cdots \sum_{r_{D-1}=1}^{R_{D-1}} (g_1(x_1))_{1,r_1} (g_2(x_2))_{r_1,r_2} \cdots (g_D(x_D))_{r_{D-1},1},$$

with $\sup_{\boldsymbol{x} \in \mathcal{X}} |f(\boldsymbol{x}) - g(\boldsymbol{x})| < \frac{\epsilon}{2}$. Let the TT SepNN approximation be

$$f_\Theta(\boldsymbol{x}) = \sum_{r_1=1}^{R_1} \cdots \sum_{r_{D-1}=1}^{R_{D-1}} (f_{\Theta_1}(x_1))_{1,r_1} (f_{\Theta_2}(x_2))_{r_1,r_2} \cdots (f_{\Theta_D}(x_D))_{r_{D-1},1},$$

with $\sup_{x_d \in \mathcal{X}_d} |(g_d(x_d))_{r,s} - (f_{\Theta_d}(x_d))_{r,s}| < \delta$ for all $d, r, s$. The total error is bounded by

$$|f(\boldsymbol{x}) - f_\Theta(\boldsymbol{x})| \leq |f(\boldsymbol{x}) - g(\boldsymbol{x})| + |g(\boldsymbol{x}) - f_\Theta(\boldsymbol{x})|$$

$$< \frac{\epsilon}{2} + \left| \sum_{r_1,\cdots,r_{D-1}} \left( \prod_{d=1}^{D} (g_d(x_d))_{r_{d-1},r_d} - \prod_{d=1}^{D} (f_{\Theta_d}(x_d))_{r_{d-1},r_d} \right) \right|,$$

where $r_0 = 1$ and $r_D = 1$. Let $M$ be a common upper bound for all components $(g_d(x_d))_{r_{d-1},r_d}, (f_{\Theta_d}(x_d))_{r_{d-1},r_d}$ on compact sets. The number of terms in the sum $\sum_{r_1,\cdots,r_{D-1}}$ is $\prod_{d=1}^{D-1} R_d$. Using the generalized Cauchy-Schwarz inequality $\left| \prod_{d=1}^{D} a_d - \prod_{d=1}^{D} b_d \right| \leq \sum_{d=1}^{D} |a_d - b_d| \left( \prod_{j=1}^{d-1} |a_j| \right) \left( \prod_{j=d+1}^{D} |b_j| \right)$, for each term we have the bound

$$\left| \prod_{d=1}^{D} (g_d(x_d))_{r_{d-1},r_d} - \prod_{d=1}^{D} (f_{\Theta_d}(x_d))_{r_{d-1},r_d} \right| < DM^{D-1}\delta.$$

Thus we have

$$|g(\boldsymbol{x}) - f_\Theta(\boldsymbol{x})| < \left( \prod_{d=1}^{D-1} R_d \right) DM^{D-1}\delta.$$

Choosing $\delta = \frac{\epsilon}{2(\prod_{d=1}^{D-1} R_d)DM^{D-1}}$ we have

$$|f(\boldsymbol{x}) - f_\Theta(\boldsymbol{x})| < \frac{\epsilon}{2} + \frac{\epsilon}{2} = \epsilon,$$

and the universal approximation theory of TT SepNN is established.

Let the Tucker approximation be

$$g(\boldsymbol{x}) = \mathcal{C} \times_1 g_1(x_1) \times_2 \cdots \times_D g_D(x_D),$$

with $\sup_{\boldsymbol{x} \in \mathcal{X}} |f(\boldsymbol{x}) - g(\boldsymbol{x})| < \frac{\epsilon}{2}$. Let the Tucker SepNN approximation be

$$f_\Theta(\boldsymbol{x}) = \mathcal{C} \times_1 f_{\Theta_1}(x_1) \times_2 \cdots \times_D f_{\Theta_D}(x_D),$$

with $\sup_{x_d \in \mathcal{X}_d} |(g_d(x_d))_r - (f_{\Theta_d}(x_d))_r| < \delta$ for all $d, r$. The total error is bounded by:

$$|f(\boldsymbol{x}) - f_\Theta(\boldsymbol{x})| \leq |f(\boldsymbol{x}) - g(\boldsymbol{x})| + |g(\boldsymbol{x}) - f_\Theta(\boldsymbol{x})|$$

$$< \frac{\epsilon}{2} + |\mathcal{C} \times_1 g_1(x_1) \times_2 \cdots \times_D g_D(x_D) - \mathcal{C} \times_1 f_{\Theta_1}(x_1) \times_2 \cdots \times_D f_{\Theta_D}(x_D)|.$$

Using the property of mode-$d$ product, we have

$$|g(\boldsymbol{x}) - f_\Theta(\boldsymbol{x})| \leq \sum_{r_1,\cdots,r_D} |\mathcal{C}_{r_1,\cdots,r_D}| \left| \prod_{d=1}^{D} (g_d(x_d))_{r_d} - \prod_{d=1}^{D} (f_{\Theta_d}(x_d))_{r_d} \right|.$$

Let $M$ be a common upper bound for all components $(g_d(x_d))_{r_d}, (f_{\Theta_d}(x_d))_{r_d}$ on compact sets, and assume each element of $\mathcal{C}$ is bounded by $N$. The number of terms in the summation is $\prod_{d=1}^{D} R_d$. For each term we have

$$\left| \prod_{d=1}^{D} (g_d(x_d))_{r_d} - \prod_{d=1}^{D} (f_{\Theta_d}(x_d))_{r_d} \right| < DM^{D-1}\delta.$$

Thus we have

$$|g(\boldsymbol{x}) - f_\Theta(\boldsymbol{x})| < \left( \prod_{d=1}^{D} R_d \right) NDM^{D-1}\delta.$$

Choosing $\delta = \frac{\epsilon}{2(\prod_{d=1}^{D} R_d)NDM^{D-1}}$, we have

$$|f(\boldsymbol{x}) - f_\Theta(\boldsymbol{x})| < \frac{\epsilon}{2} + \frac{\epsilon}{2} = \epsilon,$$

and the approximation theory of Tucker SepNN is established. Note that the approximation theory of Tucker SepNN can also be directly concluded by using the fact that CP is a special case of Tucker. The proof is finally completed. $\qquad \square$

## A.6   Proof of Lemma 1

*Proof.* The NTK is defined as

$$K_\Theta(\boldsymbol{x}, \boldsymbol{x}') = \langle \nabla_\Theta f_\Theta(\boldsymbol{x}), \nabla_\Theta f_\Theta(\boldsymbol{x}') \rangle.$$

The gradient could decompose as

$$\nabla_\Theta f_\Theta(\boldsymbol{x}) = (\nabla_{\Theta_1} f_\Theta(\boldsymbol{x}), \nabla_{\Theta_2} f_\Theta(\boldsymbol{x}), \cdots, \nabla_{\Theta_D} f_\Theta(\boldsymbol{x})).$$

Thus, the inner product becomes

$$K_\Theta(\boldsymbol{x}, \boldsymbol{x}') = \sum_{d=1}^{D} \langle \nabla_{\Theta_d} f_\Theta(\boldsymbol{x}), \nabla_{\Theta_d} f_\Theta(\boldsymbol{x}') \rangle.$$

For fixed $d$, the gradient with respect to $\Theta_d$ is

$$\nabla_{\Theta_d} f_\Theta(\boldsymbol{x}) = \frac{1}{\sqrt{R}} \sum_{r=1}^{R} \left( \prod_{d' \neq d} (f_{\Theta_{d'}}(x_{d'}))_r \right) \nabla_{\Theta_d} (f_{\Theta_d}(x_d))_r.$$

We first compute the inner product of factor gradients

$$\langle \nabla_{\Theta_d} f_\Theta(\boldsymbol{x}), \nabla_{\Theta_d} f_\Theta(\boldsymbol{x}') \rangle = \frac{1}{R} \sum_{r=1}^{R} \sum_{s=1}^{R} \left( \prod_{d' \neq d} (f_{\Theta_{d'}}(x_{d'}))_r \right) \left( \prod_{d' \neq d} (f_{\Theta_{d'}}(x'_{d'}))_s \right) \langle \nabla_{\Theta_d} (f_{\Theta_d}(x_d))_r, \nabla_{\Theta_d} (f_{\Theta_d}(x'_d))_s \rangle.$$

By definition, the inner product of gradients is the $(r, s)$-th element of the NTK matrix $\boldsymbol{K}_{\Theta_d}(x_d, x'_d)$, i.e., $\langle \nabla_{\Theta_d} (f_{\Theta_d}(x_d))_r, \nabla_{\Theta_d} (f_{\Theta_d}(x'_d))_s \rangle = (\boldsymbol{K}_{\Theta_d}(x_d, x'_d))_{r,s}$. Define the vector

$$\boldsymbol{a}_d(\boldsymbol{x}) = \left( \prod_{d' \neq d} (f_{\Theta_{d'}}(x_{d'}))_1, \prod_{d' \neq d} (f_{\Theta_{d'}}(x_{d'}))_2, \cdots, \prod_{d' \neq d} (f_{\Theta_{d'}}(x_{d'}))_R \right)^\top.$$

Then the summation above simplifies to

$$\frac{1}{R} \sum_{r=1}^{R} \sum_{s=1}^{R} (\boldsymbol{a}_d(\boldsymbol{x}))_r (\boldsymbol{a}_d(\boldsymbol{x}'))_s (\boldsymbol{K}_{\Theta_d}(x_d, x'_d))_{r,s} = \frac{1}{R} \boldsymbol{a}_d(\boldsymbol{x})^\top \boldsymbol{K}_{\Theta_d}(x_d, x'_d) \boldsymbol{a}_d(\boldsymbol{x}').$$

Therefore we have

$$\langle \nabla_{\Theta_d} f_\Theta(\boldsymbol{x}), \nabla_{\Theta_d} f_\Theta(\boldsymbol{x}') \rangle = \frac{1}{R} \boldsymbol{a}_d(\boldsymbol{x})^\top \boldsymbol{K}_{\Theta_d}(x_d, x'_d) \boldsymbol{a}_d(\boldsymbol{x}').$$

Summing over all $d$ yields the final NTK

$$K_\Theta(\boldsymbol{x}, \boldsymbol{x}') = \sum_{d=1}^{D} \langle \nabla_{\Theta_d} f_\Theta(\boldsymbol{x}), \nabla_{\Theta_d} f_\Theta(\boldsymbol{x}') \rangle = \frac{1}{R} \sum_{d=1}^{D} \boldsymbol{a}_d(\boldsymbol{x})^\top \boldsymbol{K}_{\Theta_d}(x_d, x'_d) \boldsymbol{a}_d(\boldsymbol{x}').$$

The proof is completed. $\qquad \square$

## A.7 Proof of Theorem 2

*Proof.* By Lemma 1, the NTK of the CP SepNN is given by

$$K_\Theta(\boldsymbol{x}, \boldsymbol{x}') = \frac{1}{R} \sum_{d=1}^{D} \boldsymbol{a}_d(\boldsymbol{x})^\top \boldsymbol{K}_{\Theta_d}(x_d, x_d') \boldsymbol{a}_d(\boldsymbol{x}').$$

We then analyze the behavior of this expression as $W \to \infty$ and $R \to \infty$.

We first analyze the behavior of each factor function $f_{\Theta_d}(x) = \frac{1}{\sqrt{W}} \boldsymbol{W}_{2,d} \sigma(\boldsymbol{W}_{1,d}x + \boldsymbol{b}_d)$. The $j$-th output component ($j = 1, \cdots, R$) of the network is

$$(f_{\Theta_d}(x))_j = \frac{1}{\sqrt{W}} \sum_{i=1}^{W} (\boldsymbol{W}_{2,d})_{j,i}\, \sigma((\boldsymbol{W}_{1,d})_i x + (\boldsymbol{b}_d)_i).$$

Let $z_i = (\boldsymbol{W}_{1,d})_i x + (\boldsymbol{b}_d)_i$. The gradients of $(f_{\Theta_d}(x))_j$ w.r.t. the parameters are

$$\frac{\partial (f_{\Theta_d}(x))_j}{\partial (\boldsymbol{W}_{2,d})_{j,i}} = \frac{1}{\sqrt{W}} \sigma(z_i), \text{ and } \frac{\partial (f_{\Theta_d}(x))_j}{\partial (\boldsymbol{W}_{2,d})_{j',i}} = 0, \ \ j' \neq j.$$

$$\frac{\partial (f_{\Theta_d}(x))_j}{\partial (\boldsymbol{W}_{1,d})_i} = \frac{1}{\sqrt{W}} (\boldsymbol{W}_{2,d})_{j,i}\, \dot{\sigma}(z_i)\, x.$$

$$\frac{\partial (f_{\Theta_d}(x))_j}{\partial (\boldsymbol{b}_d)_i} = \frac{1}{\sqrt{W}} (\boldsymbol{W}_{2,d})_{j,i}\, \dot{\sigma}(z_i).$$

The $(j, j')$-th element of the NTK matrix is

$$(\boldsymbol{K}_{\Theta_d}(x, x'))_{j,j'} = \sum_i \frac{\partial (f_{\Theta_d}(x))_j}{\partial (\boldsymbol{W}_{2,d})_{j,i}} \frac{\partial (f_{\Theta_d}(x'))_{j'}}{\partial (\boldsymbol{W}_{2,d})_{j',i}} + \sum_i \frac{\partial (f_{\Theta_d}(x))_j}{\partial (\boldsymbol{W}_{1,d})_i} \frac{\partial (f_{\Theta_d}(x'))_{j'}}{\partial (\boldsymbol{W}_{1,d})_i} + \sum_i \frac{\partial (f_{\Theta_d}(x))_j}{\partial (\boldsymbol{b}_d)_i} \frac{\partial (f_{\Theta_d}(x'))_{j'}}{\partial (\boldsymbol{b}_d)_i}.$$

For $j \neq j'$, all terms involve products of independent zero-mean variables (i.e., $(\boldsymbol{W}_{2,d})_{j,i}$ and $(\boldsymbol{W}_{2,d})_{j',i}$), hence $\mathbb{E}[(\boldsymbol{K}_{\Theta_d}(x, x'))_{j,j'}] = 0$. Thus, the NTK matrix is diagonal. We now compute the diagonal elements $(\boldsymbol{K}_{\Theta_d}(x, x'))_{j,j}$. For a fixed $j$, we have

$$(\boldsymbol{K}_{\Theta_d}(x, x'))_{j,j} = \sum_i \frac{\partial (f_{\Theta_d}(x))_j}{\partial (\boldsymbol{W}_{2,d})_{j,i}} \frac{\partial (f_{\Theta_d}(x'))_j}{\partial (\boldsymbol{W}_{2,d})_{j,i}} + \sum_i \frac{\partial (f_{\Theta_d}(x))_j}{\partial (\boldsymbol{W}_{1,d})_i} \frac{\partial (f_{\Theta_d}(x'))_j}{\partial (\boldsymbol{W}_{1,d})_i} + \sum_i \frac{\partial (f_{\Theta_d}(x))_j}{\partial (\boldsymbol{b}_d)_i} \frac{\partial (f_{\Theta_d}(x'))_j}{\partial (\boldsymbol{b}_d)_i}$$

$$= \sum_i \left( \frac{1}{W} \sigma(z_i)\sigma(z_i') \right) + \sum_i \left( \frac{1}{W} (\boldsymbol{W}_{2,d})_{j,i}^2 \dot{\sigma}(z_i)\dot{\sigma}(z_i')xx' \right) + \sum_i \left( \frac{1}{W} (\boldsymbol{W}_{2,d})_{j,i}^2 \dot{\sigma}(z_i)\dot{\sigma}(z_i') \right),$$

where $z_i = (\boldsymbol{W}_{1,d})_i x + (\boldsymbol{b}_d)_i$ and $z_i' = (\boldsymbol{W}_{1,d})_i x' + (\boldsymbol{b}_d)_i$. Combining these terms we have

$$(\boldsymbol{K}_{\Theta_d}(x, x'))_{j,j} = \frac{1}{W} \sum_i \sigma(z_i)\sigma(z_i') + \frac{1}{W} \sum_i (\boldsymbol{W}_{2,d})_{j,i}^2 \dot{\sigma}(z_i)\dot{\sigma}(z_i')(xx' + 1).$$

Taking the expectation over parameters we have

$$\mathbb{E}[(\boldsymbol{K}_{\Theta_d}(x, x'))_{j,j}] = \mathbb{E}\left( \frac{1}{W} \sum_i \sigma(z_i)\sigma(z_i') \right) + \mathbb{E}\left( \frac{1}{W} \sum_i (\boldsymbol{W}_{2,d})_{j,i}^2 \dot{\sigma}(z_i)\dot{\sigma}(z_i')(xx' + 1) \right).$$

Since $(\boldsymbol{W}_{2,d})_{j,i}$ is independent of $(\boldsymbol{W}_{1,d})_i$ and $(\boldsymbol{b}_d)_i$, and $\mathbb{E}[(\boldsymbol{W}_{2,d})_{j,i}^2] = 1$, we have

$$\mathbb{E}[(\boldsymbol{W}_{2,d})_{j,i}^2 \dot{\sigma}(z_i)\dot{\sigma}(z_i')] = \mathbb{E}[(\boldsymbol{W}_{2,d})_{j,i}^2]\mathbb{E}[\dot{\sigma}(z_i)\dot{\sigma}(z_i')] = \mathbb{E}[\dot{\sigma}(z_i)\dot{\sigma}(z_i')].$$

Thus we have

$$\mathbb{E}[(\boldsymbol{K}_{\Theta_d}(x, x'))_{j,j}] = \mathbb{E}\left( \frac{1}{W} \sum_i \sigma(z_i)\sigma(z_i') \right) + \mathbb{E}\left( \frac{1}{W} \sum_i \dot{\sigma}(z_i)\dot{\sigma}(z_i') \right)(xx' + 1).$$

As $W \to \infty$, by the law of large numbers, we get

$$\frac{1}{W} \sum_i \sigma(z_i)\sigma(z_i') \to \mathbb{E}_{w,b \sim \mathcal{N}(0,1)}[\sigma(wx + b)\sigma(wx' + b)],$$

$$\frac{1}{W} \sum_i \dot{\sigma}(z_i)\dot{\sigma}(z_i') \to \mathbb{E}_{w,b \sim \mathcal{N}(0,1)}[\dot{\sigma}(wx + b)\dot{\sigma}(wx' + b)].$$

Therefore we have

$$\lim_{W \to \infty} (\boldsymbol{K}_{\Theta_d}(x, x'))_{j,j} = \mathbb{E}_{w,b} \left( \sigma(wx + b)\sigma(wx' + b) \right) + \mathbb{E}_{w,b} \left( \dot{\sigma}(wx + b)\dot{\sigma}(wx' + b) \right) (xx' + 1)$$

$$= k(x, x')$$

by definition of $k(x, x')$. Since this limit is identical for all $j$, the NTK matrix $\boldsymbol{K}_{\Theta_d}(x, x')$ converges to $k(x, x')\boldsymbol{I}_R$. Substituting the diagonal form of $\boldsymbol{K}_{\Theta_d}$ into the NTK expression

$$K_\Theta(\boldsymbol{x}, \boldsymbol{x}') = \frac{1}{R} \sum_{d=1}^{D} k(x_d, x'_d) \sum_{r=1}^{R} (\boldsymbol{a}_d(\boldsymbol{x}))_r (\boldsymbol{a}_d(\boldsymbol{x}'))_r,$$

where $(\boldsymbol{a}_d(\boldsymbol{x}))_r = \prod_{d' \neq d} (f_{\Theta_{d'}}(x_{d'}))_r$. Consider the inner summation for each $d$:

$$V_d := \frac{1}{R} \sum_{r=1}^{R} (\boldsymbol{a}_d(\boldsymbol{x}))_r (\boldsymbol{a}_d(\boldsymbol{x}'))_r = \frac{1}{R} \sum_{r=1}^{R} \prod_{d' \neq d} (f_{\Theta_{d'}}(x_{d'}))_r (f_{\Theta_{d'}}(x'_{d'}))_r.$$

The factor MLPs are i.i.d. across the output components $r$ (see (Lee et al., 2018)), and the terms $(f_{\Theta_{d'}}(x_{d'}))_r$ and $(f_{\Theta_{d'}}(x'_{d'}))_r$ are also independent across $d'$. By the law of large numbers, as $R \to \infty$, we have

$$V_d \to \mathbb{E} \left( \prod_{d' \neq d} (f_{\Theta_{d'}}(x_{d'}))_r (f_{\Theta_{d'}}(x'_{d'}))_r \right) = \prod_{d' \neq d} \mathbb{E} \left( (f_{\Theta_{d'}}(x_{d'}))_r (f_{\Theta_{d'}}(x'_{d'}))_r \right).$$

As $W \to \infty$, by the neural network Gaussian process theory (Lee et al., 2018), we have that $f_{\Theta_{d'}}$ converges to a Gaussian process with covariance

$$\mathbb{E} \left( (f_{\Theta_{d'}}(x_{d'}))_r (f_{\Theta_{d'}}(x'_{d'}))_r \right) = \mathbb{E}_{w,b \sim \mathcal{N}(0,1)}[\sigma(wx_{d'} + b)\sigma(wx'_{d'} + b)] = c_{d'}(x_{d'}, x'_{d'}),$$

where $c_{d'}(x_{d'}, x'_{d'}) = \mathbb{E}_{w,b \sim \mathcal{N}(0,1)} \left( \sigma(wx_{d'} + b)\sigma(wx'_{d'} + b) \right)$ is by definition. Therefore, we have

$$V_d \to \prod_{d' \neq d} c_{d'}(x_{d'}, x'_{d'}),$$

and the overall NTK converges to

$$K_\Theta(\boldsymbol{x}, \boldsymbol{x}') \to \sum_{d=1}^{D} k(x_d, x'_d) \prod_{d' \neq d} c_{d'}(x_{d'}, x'_{d'}),$$

as stated in the theorem. Since the convergence of each factor MLP's NTK matrix is almost sure as $W \to \infty$, and the law of large numbers gives almost sure convergence as $R \to \infty$, the overall convergence is in almost sure regime. $\square$

## A.8 Proof of Corollary 1

*Proof.* We prove Corollary 1 by leveraging the results from Theorem 2 and Lemma 1. Recall that the CP SepNN is defined as

$$f_\Theta(x_1, \cdots, x_D) = \frac{1}{\sqrt{R}} \sum_{r=1}^{R} (f_{\Theta_1}(x_1))_r (f_{\Theta_2}(x_2))_r \cdots (f_{\Theta_D}(x_D))_r,$$

where each factor MLP $f_{\Theta_d} : \mathbb{R} \to \mathbb{R}^R$ has the architecture

$$f_{\Theta_d}(x_d) = \frac{1}{\sqrt{W}} \boldsymbol{W}_{2,d} \sigma(\boldsymbol{W}_{1,d} x_d + \boldsymbol{b}_d)$$

with parameters initialized independently from $\mathcal{N}(0, 1)$. From Lemma 1, the NTK of the SepNN is

$$K_\Theta(\boldsymbol{x}, \boldsymbol{x}') = \frac{1}{R} \sum_{d=1}^{D} \boldsymbol{a}_d(\boldsymbol{x})^\top \boldsymbol{K}_{\Theta_d}(x_d, x'_d) \boldsymbol{a}_d(\boldsymbol{x}'),$$

where $(\boldsymbol{a}_d(\boldsymbol{x}))_r = \prod_{d' \neq d} (f_{\Theta_{d'}}(x_{d'}))_r$ and $\boldsymbol{K}_{\Theta_d}(x_d, x'_d)$ is the NTK matrix of the $d$-th factor MLP. Consider the limit as the width $W \to \infty$ while the rank $R$ is fixed. From the proof of Theorem 2,

for each factor MLP $f_{\Theta_d}$, as $W \to \infty$, the NTK matrix $\boldsymbol{K}_{\Theta_d}(x_d, x'_d)$ converges almost surely to a deterministic diagonal matrix

$$\boldsymbol{K}_{\Theta_d}(x_d, x'_d) \to k(x_d, x'_d)\boldsymbol{I}_R,$$

where $k(x_d, x'_d) = c_d(x_d, x'_d) + \mathbb{E}_{w,b\sim\mathcal{N}(0,1)}(\dot{\sigma}(wx_d + b)\dot{\sigma}(wx'_d + b)(x_dx'_d + 1))$ and $c_d(x_d, x'_d) = \mathbb{E}_{w,b\sim\mathcal{N}(0,1)}(\sigma(wx_d + b)\sigma(wx'_d + b))$. Substituting this into the NTK expression

$$K_\Theta(\boldsymbol{x}, \boldsymbol{x}') \to \frac{1}{R}\sum_{d=1}^{D} k(x_d, x'_d)\sum_{r=1}^{R}(\boldsymbol{a}_d(\boldsymbol{x}))_r(\boldsymbol{a}_d(\boldsymbol{x}'))_r.$$

Define

$$V_d(\boldsymbol{x}, \boldsymbol{x}') := \frac{1}{R}\sum_{r=1}^{R}(\boldsymbol{a}_d(\boldsymbol{x}))_r(\boldsymbol{a}_d(\boldsymbol{x}'))_r = \frac{1}{R}\sum_{r=1}^{R}\prod_{d'\neq d}(f_{\Theta_{d'}}(x_{d'}))_r(f_{\Theta_{d'}}(x'_{d'}))_r.$$

Then we have

$$K_\Theta(\boldsymbol{x}, \boldsymbol{x}') \to \sum_{d=1}^{D} k(x_d, x'_d)V_d(\boldsymbol{x}, \boldsymbol{x}').$$

It remains to characterize the behavior of $V_d(\boldsymbol{x}, \boldsymbol{x}')$. The factor MLPs are initialized independently across dimensions $d'$, and for each fixed $d'$, the output components $(f_{\Theta_{d'}}(x_{d'}))_r$ for $r = 1, \cdots, R$ are independent and i.i.d. due to the independent initialization of $\boldsymbol{W}_{2,d'}$ (Lee et al., 2018). As $W \to \infty$, each output $(f_{\Theta_{d'}}(x_{d'}))_r$ converges to a Gaussian process with covariance function

$$\mathbb{E}\left((f_{\Theta_{d'}}(x_{d'}))_r(f_{\Theta_{d'}}(x'_{d'}))_r\right) = c_{d'}(x_{d'}, x'_{d'})$$

by the neural network Gaussian process theory (Lee et al., 2018). Consequently, as $W \to \infty$, $K_\Theta(\boldsymbol{x}, \boldsymbol{x}')$ converges in distribution to the stochastic kernel

$$\sum_{d=1}^{D} k(x_d, x'_d)V_d(\boldsymbol{x}, \boldsymbol{x}'),$$

where $V_d(\boldsymbol{x}, \boldsymbol{x}') = \frac{1}{R}\sum_{r=1}^{R}\prod_{d'\neq d}(f_{\Theta_{d'}}(x_{d'}))_r(f_{\Theta_{d'}}(x'_{d'}))_r$ and each $(f_{\Theta_{d'}}(x_{d'}))_r$ is a Gaussian process with covariance $c_{d'}(x_{d'}, x'_{d'})$. $\qquad\square$

### A.9 Proof of Lemma 2

*Proof.* From the construction of $\boldsymbol{M}_d$ in (8) we can see that

$$\boldsymbol{M}_1 = f_{\Theta_2}(\hat{\boldsymbol{x}}_2)\left(\mathcal{R}^\top\boldsymbol{S}_1 + \boldsymbol{S}_2\mathcal{R}^\top\right), \quad \boldsymbol{M}_2 = f_{\Theta_1}(\hat{\boldsymbol{x}}_1)\left(\boldsymbol{S}_1\mathcal{R} + \mathcal{R}\boldsymbol{S}_2\right),$$

where we have used the symmetry of $\boldsymbol{S}_d$. Let $\boldsymbol{A} = f_{\Theta_1}(\hat{\boldsymbol{x}}_1) \in \mathbb{R}^{R\times n}$, $\boldsymbol{B} = f_{\Theta_2}(\hat{\boldsymbol{x}}_2) \in \mathbb{R}^{R\times n}$. Then the output matrix of the SepNN is $\mathcal{Z}_\Theta = \boldsymbol{A}^\top\boldsymbol{B}$. The vectorized output (let $\boldsymbol{X} \in \mathbb{R}^{2\times n^2}$ be row-first batched inputs) is

$$\mathrm{vec}(\mathcal{Z}_\Theta^\top) = \mathrm{vec}(\boldsymbol{B}^\top\boldsymbol{A}) = (\boldsymbol{I}_n \otimes \boldsymbol{B}^\top)\mathrm{vec}(\boldsymbol{A}).$$

For parameters $\Theta_1$, the gradient given by classical NTK-based PGD is

$$\nabla_{\Theta_1} f_\Theta(\boldsymbol{X})^\top\tilde{\boldsymbol{S}}\boldsymbol{r} = \left(\frac{\partial\mathrm{vec}(\mathcal{Z}_\Theta^\top)}{\partial\Theta_1}\right)^\top\tilde{\boldsymbol{S}}\boldsymbol{r}.$$

Since $\frac{\partial\mathrm{vec}(\mathcal{Z}_\Theta^\top)}{\partial\Theta_1} = (\boldsymbol{I}_n \otimes \boldsymbol{B}^\top)\frac{\partial\mathrm{vec}(\boldsymbol{A})}{\partial\Theta_1}$, we have

$$\left(\frac{\partial\mathrm{vec}(\mathcal{Z}_\Theta^\top)}{\partial\Theta_1}\right)^\top = \left(\frac{\partial\mathrm{vec}(\boldsymbol{A})}{\partial\Theta_1}\right)^\top(\boldsymbol{I}_n \otimes \boldsymbol{B}).$$

Thus we have

$$\nabla_{\Theta_1} f_\Theta(\boldsymbol{X})^\top\tilde{\boldsymbol{S}}\boldsymbol{r} = \left(\frac{\partial\mathrm{vec}(\boldsymbol{A})}{\partial\Theta_1}\right)^\top(\boldsymbol{I}_n \otimes \boldsymbol{B})\tilde{\boldsymbol{S}}\boldsymbol{r}.$$

The gradient in SepPGD is

$$\nabla_{\Theta_1}\langle\boldsymbol{A}, \boldsymbol{M}_1\rangle = \left(\frac{\partial\mathrm{vec}(\boldsymbol{A})}{\partial\Theta_1}\right)^\top\mathrm{vec}(\boldsymbol{M}_1) = \left(\frac{\partial\mathrm{vec}(\boldsymbol{A})}{\partial\Theta_1}\right)^\top\mathrm{vec}(\boldsymbol{B}(\mathcal{R}^\top\boldsymbol{S}_1 + \boldsymbol{S}_2\mathcal{R}^\top)).$$

By the Kronecker product property $(\boldsymbol{C}^\top \otimes \boldsymbol{A})\mathrm{vec}(\boldsymbol{B}) = \mathrm{vec}(\boldsymbol{ABC})$, we can obtain

$$(\boldsymbol{I}_n \otimes \boldsymbol{B})\tilde{\boldsymbol{S}}\boldsymbol{r} = (\boldsymbol{I}_n \otimes \boldsymbol{B})(\boldsymbol{S}_1 \otimes \boldsymbol{I}_n + \boldsymbol{I}_n \otimes \boldsymbol{S}_2)\mathrm{vec}(\mathcal{R}^\top)$$
$$= (\boldsymbol{I}_n \otimes \boldsymbol{B})\mathrm{vec}(\mathcal{R}^\top \boldsymbol{S}_1 + \boldsymbol{S}_2\mathcal{R}^\top)$$
$$= \mathrm{vec}(\boldsymbol{B}\mathcal{R}^\top \boldsymbol{S}_1 + \boldsymbol{B}\boldsymbol{S}_2\mathcal{R}^\top)$$
$$= \mathrm{vec}(\boldsymbol{B}(\mathcal{R}^\top \boldsymbol{S}_1 + \boldsymbol{S}_2\mathcal{R}^\top)).$$

Therefore we have

$$\nabla_{\Theta_1} f_\Theta(\boldsymbol{X})^\top \tilde{\boldsymbol{S}}\boldsymbol{r} = \nabla_{\Theta_1}\langle \boldsymbol{A}, \boldsymbol{M}_1\rangle.$$

Similarly we can show that $\nabla_{\Theta_2} f_\Theta(\boldsymbol{X})^\top \tilde{\boldsymbol{S}}\boldsymbol{r} = \nabla_{\Theta_2}\langle \boldsymbol{B}, \boldsymbol{M}_2\rangle$. These results imply that

$$[\nabla_{\Theta_1}\langle f_{\Theta_1}(\hat{\boldsymbol{x}}_1), \boldsymbol{M}_1\rangle, \nabla_{\Theta_2}\langle f_{\Theta_2}(\hat{\boldsymbol{x}}_2), \boldsymbol{M}_2\rangle] = \nabla_\Theta f_\Theta(\boldsymbol{X})^\top \tilde{\boldsymbol{S}}\boldsymbol{r},$$

and hence the SepPGD is equivalent to the classical NTK-based PGD. $\square$

### A.10 Proof of Lemma 3

*Proof.* The NTK of the SepNN for two points $\boldsymbol{x} = (x_1, x_2)$ and $\boldsymbol{x}' = (x_1', x_2')$ is

$$K_\Theta(\boldsymbol{x}, \boldsymbol{x}') = \langle \nabla_{\Theta_1} f_\Theta(\boldsymbol{x}), \nabla_{\Theta_1} f_\Theta(\boldsymbol{x}')\rangle + \langle \nabla_{\Theta_2} f_\Theta(\boldsymbol{x}), \nabla_{\Theta_2} f_\Theta(\boldsymbol{x}')\rangle.$$

The gradients are

$$\nabla_{\Theta_1} f_\Theta(\boldsymbol{x}) = \frac{1}{\sqrt{R}}\sum_{r=1}^R (f_{\Theta_2}(x_2))_r \nabla_{\Theta_1}(f_{\Theta_1}(x_1))_r,$$

$$\nabla_{\Theta_2} f_\Theta(\boldsymbol{x}) = \frac{1}{\sqrt{R}}\sum_{r=1}^R (f_{\Theta_1}(x_1))_r \nabla_{\Theta_2}(f_{\Theta_2}(x_2))_r.$$

Thus we have

$$\langle \nabla_{\Theta_1} f_\Theta(\boldsymbol{x}), \nabla_{\Theta_1} f_\Theta(\boldsymbol{x}')\rangle = \frac{1}{R}\sum_{r,s=1}^R (f_{\Theta_2}(x_2))_r (f_{\Theta_2}(x_2'))_s \langle \nabla_{\Theta_1}(f_{\Theta_1}(x_1))_r, \nabla_{\Theta_1}(f_{\Theta_1}(x_1'))_s\rangle,$$

$$\langle \nabla_{\Theta_2} f_\Theta(\boldsymbol{x}), \nabla_{\Theta_2} f_\Theta(\boldsymbol{x}')\rangle = \frac{1}{R}\sum_{r,s=1}^R (f_{\Theta_1}(x_1))_r (f_{\Theta_1}(x_1'))_s \langle \nabla_{\Theta_2}(f_{\Theta_2}(x_2))_r, \nabla_{\Theta_2}(f_{\Theta_2}(x_2'))_s\rangle.$$

Therefore,

$$K_\Theta(\boldsymbol{x}, \boldsymbol{x}') = \frac{1}{R}\sum_{r,s=1}^R \Big( (f_{\Theta_2}(x_2))_r (f_{\Theta_2}(x_2'))_s \langle \nabla_{\Theta_1}(f_{\Theta_1}(x_1))_r, \nabla_{\Theta_1}(f_{\Theta_1}(x_1'))_s\rangle$$
$$+ (f_{\Theta_1}(x_1))_r (f_{\Theta_1}(x_1'))_s \langle \nabla_{\Theta_2}(f_{\Theta_2}(x_2))_r, \nabla_{\Theta_2}(f_{\Theta_2}(x_2'))_s\rangle \Big).$$

Let $\hat{\boldsymbol{x}}_d = ((\hat{\boldsymbol{x}}_d)_1, \cdots, (\hat{\boldsymbol{x}}_d)_n)^\top$ for $d = 1, 2$. Define $\boldsymbol{F}_d \in \mathbb{R}^{R\times n}$ with $(\boldsymbol{F}_d)_{r,i} = (f_{\Theta_d}((\hat{\boldsymbol{x}}_d)_i))_r$, so $(f_{\Theta_d}(\hat{\boldsymbol{x}}_d))_{r,:} \in \mathbb{R}^{1\times n}$ is the $r$-th row of $\boldsymbol{F}_d$. For $r, s = 1, \cdots, R$, define $\boldsymbol{K}_d^{(r,s)} \in \mathbb{R}^{n\times n}$ by

$$\left(\boldsymbol{K}_d^{(r,s)}\right)_{i,j} = \left\langle \nabla_{\Theta_d}(f_{\Theta_d}((\hat{\boldsymbol{x}}_d)_i))_r, \nabla_{\Theta_d}\left(f_{\Theta_d}((\hat{\boldsymbol{x}}_d)_j)\right)_s\right\rangle.$$

The NTK matrix $\boldsymbol{K} \in \mathbb{R}^{n^2\times n^2}$ has entries

$$\boldsymbol{K}_{(i,j),(i',j')} = K_\Theta\left(((\hat{\boldsymbol{x}}_1)_i, (\hat{\boldsymbol{x}}_2)_j), ((\hat{\boldsymbol{x}}_1)_{i'}, (\hat{\boldsymbol{x}}_2)_{j'})\right).$$

Substituting into the above expression:

$$\boldsymbol{K}_{(i,j),(i',j')} = \frac{1}{R}\sum_{r,s=1}^R \Big( \left(f_{\Theta_2}((\hat{\boldsymbol{x}}_2)_j)\right)_r \left(f_{\Theta_2}((\hat{\boldsymbol{x}}_2)_{j'})\right)_s \langle \nabla_{\Theta_1}(f_{\Theta_1}((\hat{\boldsymbol{x}}_1)_i))_r, \nabla_{\Theta_1}(f_{\Theta_1}((\hat{\boldsymbol{x}}_1)_{i'}))_s\rangle$$
$$+ (f_{\Theta_1}((\hat{\boldsymbol{x}}_1)_i))_r (f_{\Theta_1}((\hat{\boldsymbol{x}}_1)_{i'}))_s \left\langle \nabla_{\Theta_2}\left(f_{\Theta_2}((\hat{\boldsymbol{x}}_2)_j)\right)_r, \nabla_{\Theta_2}\left(f_{\Theta_2}((\hat{\boldsymbol{x}}_2)_{j'})\right)_s\right\rangle \Big).$$

According to the definitions of $\boldsymbol{F}_d$ and $\boldsymbol{K}_d^{(r,s)}$ we have

$$\boldsymbol{K}_{(i,j),(i',j')} = \frac{1}{R}\sum_{r,s=1}^R \left((\boldsymbol{F}_2)_{r,j}(\boldsymbol{F}_2)_{s,j'}\left(\boldsymbol{K}_1^{(r,s)}\right)_{i,i'} + (\boldsymbol{F}_1)_{r,i}(\boldsymbol{F}_1)_{s,i'}\left(\boldsymbol{K}_2^{(r,s)}\right)_{j,j'}\right).$$

Note that

$$(\boldsymbol{F}_2)_{r,j}(\boldsymbol{F}_2)_{s,j'} = \left( (f_{\Theta_2}(\hat{\boldsymbol{x}}_2))_{r,:}^\top (f_{\Theta_2}(\hat{\boldsymbol{x}}_2))_{s,:} \right)_{j,j'}$$

and

$$(\boldsymbol{F}_1)_{r,i}(\boldsymbol{F}_1)_{s,i'} = \left( (f_{\Theta_1}(\hat{\boldsymbol{x}}_1))_{r,:}^\top (f_{\Theta_1}(\hat{\boldsymbol{x}}_1))_{s,:} \right)_{i,i'}.$$

Therefore, the first term in $\boldsymbol{K}_{(i,j),(i',j')}$ corresponds to

$$\boldsymbol{K}_1^{(r,s)} \otimes \left( (f_{\Theta_2}(\hat{\boldsymbol{x}}_2))_{r,:}^\top (f_{\Theta_2}(\hat{\boldsymbol{x}}_2))_{s,:} \right),$$

and the second term in $\boldsymbol{K}_{(i,j),(i',j')}$ corresponds to

$$\left( (f_{\Theta_1}(\hat{\boldsymbol{x}}_1))_{r,:}^\top (f_{\Theta_1}(\hat{\boldsymbol{x}}_1))_{s,:} \right) \otimes \boldsymbol{K}_2^{(r,s)}.$$

Summing over $r, s$ and scaling by $1/R$ gives

$$\boldsymbol{K} = \frac{1}{R} \sum_{r,s=1}^R \left( \boldsymbol{K}_1^{(r,s)} \otimes \left( (f_{\Theta_2}(\hat{\boldsymbol{x}}_2))_{r,:}^\top (f_{\Theta_2}(\hat{\boldsymbol{x}}_2))_{s,:} \right) + \left( (f_{\Theta_1}(\hat{\boldsymbol{x}}_1))_{r,:}^\top (f_{\Theta_1}(\hat{\boldsymbol{x}}_1))_{s,:} \right) \otimes \boldsymbol{K}_2^{(r,s)} \right),$$

as required. □

### A.11 Proof of Theorem 3

*Proof.* According to Lemma 1, the NTK of $f_\Theta$, defined as $K_\Theta(\boldsymbol{x}, \boldsymbol{x}') := \langle \nabla_\Theta f_\Theta(\boldsymbol{x}), \nabla_\Theta f_\Theta(\boldsymbol{x}') \rangle$ for $\boldsymbol{x} = (x_1, \cdots, x_D)$ and $\boldsymbol{x}' = (x_1', \cdots, x_D')$, is given by

$$K_\Theta(\boldsymbol{x}, \boldsymbol{x}') = \frac{1}{R} \sum_{d=1}^D \boldsymbol{a}_d(\boldsymbol{x})^\top \boldsymbol{K}_{\Theta_d}(x_d, x_d') \boldsymbol{a}_d(\boldsymbol{x}'), \tag{12}$$

where $\boldsymbol{K}_{\Theta_d}(x_d, x_d') \in \mathbb{R}^{R \times R}$ is the NTK matrix of the $d$-th factor MLP $f_{\Theta_d}$ with elements $(\boldsymbol{K}_{\Theta_d}(x_d, x_d'))_{r,s} = \langle \nabla_{\Theta_d} (f_{\Theta_d}(x_d))_r, \nabla_{\Theta_d} (f_{\Theta_d}(x_d'))_s \rangle$, and $\boldsymbol{a}_d(\boldsymbol{x})$ is a vector defined by $\boldsymbol{a}_d(\boldsymbol{x}) = \left( \prod_{d' \neq d} (f_{\Theta_{d'}}(x_{d'}))_1, \prod_{d' \neq d} (f_{\Theta_{d'}}(x_{d'}))_2, \cdots, \prod_{d' \neq d} (f_{\Theta_{d'}}(x_{d'}))_R \right)^\top \in \mathbb{R}^R$. In the infinitely small under standard gradient descent optimizer with infinitely smaller learning rate, the dynamics of network predictions $\boldsymbol{u}(t) = (f_{\Theta(t)}(\boldsymbol{x}_1), \cdots, f_{\Theta(t)}(\boldsymbol{x}_n))^\top \in \mathbb{R}^n$ under standard gradient descent optimizer would follow $\frac{d\boldsymbol{u}(t)}{dt} = -\boldsymbol{K}(t)(\boldsymbol{u}(t) - \boldsymbol{y})$, where $\boldsymbol{K}(t) \in \mathbb{R}^{n \times n}$ is the NTK matrix over $\{\boldsymbol{x}_i\}$. The dynamics of each network weight $w(t)$ under standard gradient descent optimizer with infinitely smaller learning rate would follow $\frac{dw(t)}{dt} = -\sum_i (f_{\Theta(t)}(\boldsymbol{x}_i) - y_i) \frac{df_{\Theta(t)}(\boldsymbol{x}_i)}{dw(t)}$.

For a fixed $d$, denote $\boldsymbol{a}_i^t = \boldsymbol{a}_d(\boldsymbol{x}_i) \in \mathbb{R}^R$, where $\boldsymbol{a}_d(\boldsymbol{x}_i)$ are defined using the network weights at time $t$, i.e., $\Theta(t)$ (since the asymptotic behavior of each $\boldsymbol{a}_d(\boldsymbol{x}_i)$ is the same, we can discard the subscript $d$ for simplicity). Denote $\boldsymbol{K}_{i,j}^t = \boldsymbol{K}_{\Theta_d(t)}((\boldsymbol{x}_i)_d, (\boldsymbol{x}_j)_d) \in \mathbb{R}^{R \times R}$. We bound the difference $K_{\Theta(t)}(\boldsymbol{x}_i, \boldsymbol{x}_j) - K_{\Theta(0)}(\boldsymbol{x}_i, \boldsymbol{x}_j)$ using

$$|K_{\Theta(t)}(\boldsymbol{x}_i, \boldsymbol{x}_j) - K_{\Theta(0)}(\boldsymbol{x}_i, \boldsymbol{x}_j)| = \left| \frac{1}{R} \sum_d \left( \boldsymbol{a}_i^t \boldsymbol{K}_{i,j}^t \boldsymbol{a}_j^t - \boldsymbol{a}_i^0 \boldsymbol{K}_{i,j}^t \boldsymbol{a}_j^t + \boldsymbol{a}_i^0 \boldsymbol{K}_{i,j}^t \boldsymbol{a}_j^t - \boldsymbol{a}_i^0 \boldsymbol{K}_{i,j}^0 \boldsymbol{a}_j^0 \right) \right|$$

$$\leq \frac{1}{R} \sum_d \left| (\boldsymbol{a}_i^t - \boldsymbol{a}_i^0) \boldsymbol{K}_{i,j}^t \boldsymbol{a}_j^t \right| + \left| \boldsymbol{a}_i^0 (\boldsymbol{K}_{i,j}^t \boldsymbol{a}_j^t - \boldsymbol{K}_{i,j}^0 \boldsymbol{a}_j^0) \right| \tag{13}$$

$$\leq \frac{1}{R} \sum_d \left| (\boldsymbol{a}_i^t - \boldsymbol{a}_i^0) \boldsymbol{K}_{i,j}^t \boldsymbol{a}_j^t \right| + \left| \boldsymbol{a}_i^0 \boldsymbol{K}_{i,j}^t (\boldsymbol{a}_j^t - \boldsymbol{a}_j^0) \right| + \left| \boldsymbol{a}_i^0 (\boldsymbol{K}_{i,j}^t - \boldsymbol{K}_{i,j}^0) \boldsymbol{a}_j^0 \right|.$$

We next show that the variance of the movements $|\boldsymbol{a}_i^t - \boldsymbol{a}_i^0|, |\boldsymbol{a}_j^t - \boldsymbol{a}_j^0|, |\boldsymbol{K}_{i,j}^t - \boldsymbol{K}_{i,j}^0|$ and $\boldsymbol{a}_j^t, \boldsymbol{a}_i^0, \boldsymbol{a}_j^0, \boldsymbol{K}_{i,j}^t$ are bounded by some constants related to $R, W$.

Let the loss function be $\mathcal{L} = \frac{1}{2}(f_\Theta(\boldsymbol{x}_i) - y_i)^2$. Then the dynamic of each weight matrix $\boldsymbol{W}_{2,d}$ is

$$
\begin{aligned}
\frac{d(\boldsymbol{W}_{2,d})_{r,s}}{dt} &= -\frac{\partial \mathcal{L}}{\partial (\boldsymbol{W}_{2,d})_{r,s}} \\
&= -\sum_i (f_\Theta(\boldsymbol{x}_i) - y_i)\frac{\partial f_\Theta(\boldsymbol{x}_i)}{\partial (\boldsymbol{W}_{2,d})_{r,s}} \\
&= -\sum_i (\underbrace{f_\Theta(\boldsymbol{x}_i)}_{\text{Var. } O(1)} - y_i)\frac{1}{\sqrt{R}}\left(\underbrace{\prod_{d' \neq d}(f_{\Theta_{d'}}(\boldsymbol{x}_i)_{d'})_r}_{\text{Var. } O(1)}\right)\frac{1}{\sqrt{W}}\underbrace{\sigma(z_s)}_{\text{Var. } O(1)} \\
&= O(\frac{1}{\sqrt{RW}})X,
\end{aligned}
$$

where $z_s = (\boldsymbol{W}_{1,d})_s x + (\boldsymbol{b}_d)_s$ and $X$ is a r.v. with variance $O(1)$, and $\prod_{d' \neq d}(f_{\Theta_{d'}}(\boldsymbol{x}_i)_{d'})_r$ has variance $O(1)$ by the i.i.d. $\mathcal{N}(0,1)$ weight assumption and the scaling factor $\frac{1}{\sqrt{W}}$ following the neural network Gaussian process theory (Lee et al., 2018). Also, we have used the fact that $D$ is a fixed constant dimension. Therefore $\frac{d(\boldsymbol{W}_{2,d})_{r,s}}{dt}$ has variance $O(\frac{1}{RW})$. Similarly, we have

$$
\begin{aligned}
\frac{d(\boldsymbol{W}_{1,d})_s}{dt} &= -\sum_i (f_\Theta(\boldsymbol{x}_i) - y_i)\frac{1}{\sqrt{W}}\frac{1}{\sqrt{R}}(\boldsymbol{x}_i)_d \underbrace{\sum_r \left(\prod_{d' \neq d}(f_{\Theta_{d'}}(\boldsymbol{x}_i)_{d'})_r\right)(\boldsymbol{W}_{2,d})_{r,s}\dot\sigma(z_s)}_{\text{Var. } O(R)} \\
&= O(\frac{1}{\sqrt{W}})\frac{1}{\sqrt{R}}Y,
\end{aligned}
$$

where $Y$ has variance $O(R)$, and thus $\frac{1}{\sqrt{R}}Y$ has variance $O(1)$. Therefore $\frac{d(\boldsymbol{W}_{1,d})_s}{dt}$ has variance $O(\frac{1}{W})$. Similarly, $\frac{d(\boldsymbol{b}_d)_s}{dt}$ can be written as $O(\frac{1}{\sqrt{W}})\frac{1}{\sqrt{R}}Y$ for some r.v. $Y$ of variance $O(R)$. For each weight $w(t)$, its movement is $|w(t) - w(0)| \leq \int_0^t |\frac{w(\tau)}{d\tau}|d\tau$. Therefore, the network prediction change $\|f_{\Theta_d(t)}((\boldsymbol{x}_i)_d) - f_{\Theta_d(0)}((\boldsymbol{x}_i)_d)\|_{\ell_2}$ is bounded by

$$
\begin{aligned}
\|f_{\Theta_d(t)}((\boldsymbol{x}_i)_d) - f_{\Theta_d(0)}((\boldsymbol{x}_i)_d)\|_{\ell_2} &= \frac{1}{\sqrt{W}}\|\boldsymbol{W}_{2,d}(t)\sigma(\boldsymbol{W}_{1,d}(t)(\boldsymbol{x}_i)_d + \boldsymbol{b}_d(t)) - \boldsymbol{W}_{2,d}(0)\sigma(\boldsymbol{W}_{1,d}(0)(\boldsymbol{x}_i)_d + \boldsymbol{b}_d(0))\|_{\ell_2} \\
&\leq \frac{1}{\sqrt{W}}\|\underbrace{(\boldsymbol{W}_{2,d}(t) - \boldsymbol{W}_{2,d}(0))}_{\text{Var. } O(t)}\underbrace{\sigma(\boldsymbol{W}_{1,d}(t)(\boldsymbol{x}_i)_d + \boldsymbol{b}_d(t))}_{\text{Var. } O(W)}\|_{\ell_2} \\
&\quad + \frac{1}{\sqrt{W}}\|\underbrace{\boldsymbol{W}_{2,d}(0)}_{\text{Var. } O(RW)}\underbrace{(\sigma(\boldsymbol{W}_{1,d}(t)(\boldsymbol{x}_i)_d + \boldsymbol{b}_d(t)) - \sigma(\boldsymbol{W}_{1,d}(0)(\boldsymbol{x}_i)_d + \boldsymbol{b}_d(0)))}_{\text{Var. } O(t)}\|_{\ell_2},
\end{aligned}
$$

where we have used the Lipschitz smoothness of $\sigma$ and the small movement of each $(\boldsymbol{W}_{1,d})_s$ and $(\boldsymbol{W}_{2,d})_{r,s}$ to compose the total movement by summing over the variance. This implies that the total movement $\|f_{\Theta_d(t)}((\boldsymbol{x}_i)_d) - f_{\Theta_d(0)}((\boldsymbol{x}_i)_d)\|_{\ell_2}$ has variance not larger than $O(Rt)$, and thus each rank component's movement $|(f_{\Theta_d(t)}((\boldsymbol{x}_i)_d))_r - (f_{\Theta_d(0)}((\boldsymbol{x}_i)_d))_r|$ has variance not larger than $O(t)$. According to the proof of Theorem 2, we have each factor MLP's NTK formulation:

$$
(\boldsymbol{K}_{\Theta_d}(x, x'))_{j,j} = \frac{1}{W}\sum_i \sigma(z_i)\sigma(z_i') + \frac{1}{W}\sum_i (\boldsymbol{W}_{2,d})_{j,i}^2 \dot\sigma(z_i)\dot\sigma(z_i')(xx' + 1), \tag{14}
$$

where $z_i = (\boldsymbol{W}_{1,d})_i x + (\boldsymbol{b}_d)_i$ and $z_i' = (\boldsymbol{W}_{1,d})_i x' + (\boldsymbol{b}_d)_i$. According to (14), for two points $x, x'$, the change of factor NTK matrix is

$$|(\boldsymbol{K}_{\Theta_d(t)}(x,x'))_{j,j} - (\boldsymbol{K}_{\Theta_d(0)}(x,x'))_{j,j}| \leq \frac{1}{W} \sum_i |\sigma(z_i(t))\sigma(z_i'(t)) - \sigma(z_i(0))\sigma(z_i'(0))|$$

$$+ \frac{1}{W} \sum_i |(\boldsymbol{W}_{2,d}(t))_{j,i}^2 \dot{\sigma}(z_i(t))\dot{\sigma}(z_i'(t)) - (\boldsymbol{W}_{2,d}(0))_{j,i}^2 \dot{\sigma}(z_i(0))\dot{\sigma}(z_i'(0))||xx' + 1|$$

$$= \frac{1}{W} \sum_i |\underbrace{\sigma(z_i(t))\sigma(z_i'(t)) - \sigma(z_i(t))\sigma(z_i'(0))}_{\text{Var. } O(\frac{t}{W})} + \underbrace{\sigma(z_i(t))\sigma(z_i'(0)) - \sigma(z_i(0))\sigma(z_i'(0))}_{\text{Var. } O(\frac{t}{W})} |$$

$$+ \frac{1}{W} \sum_i |\underbrace{(\boldsymbol{W}_{2,d}(t))_{j,i}^2 \dot{\sigma}(z_i(t))\dot{\sigma}(z_i'(t)) - (\boldsymbol{W}_{2,d}(t))_{j,i}^2 \dot{\sigma}(z_i(0))\dot{\sigma}(z_i'(0))}_{\text{Var. } O(\frac{t}{W})}$$

$$+ \underbrace{(\boldsymbol{W}_{2,d}(t))_{j,i}^2 \dot{\sigma}(z_i(0))\dot{\sigma}(z_i'(0)) - (\boldsymbol{W}_{2,d}(0))_{j,i}^2 \dot{\sigma}(z_i(0))\dot{\sigma}(z_i'(0))}_{\text{Var. } O(\frac{t}{RW})} ||xx' + 1|.$$

Therefore, the change $|(\boldsymbol{K}_{\Theta_d(t)}(x,x'))_{j,j} - (\boldsymbol{K}_{\Theta_d(0)}(x,x'))_{j,j}|$ would have variance not larger than $O(\frac{t}{W})$ by summing over $i$ and the scaling factor $\frac{1}{W}$. In the infinite width limit, the factor NTK matrix $\boldsymbol{K}_{\Theta_d}(x,x') \in \mathbb{R}^{R\times R}$ is diagonal. Hence the factor NTK matrix $\boldsymbol{K}_{\Theta_d}(x,x') \in \mathbb{R}^{R\times R}$ change norm $\|\boldsymbol{K}_{\Theta_d(t)}(x,x') - \boldsymbol{K}_{\Theta_d(0)}(x,x')\|_{op}$ has variance not larger than $O(\frac{t}{W})$ in terms of operator norm (the largest diagonal element). Similarly, the NTK matrix norm $\|\boldsymbol{K}_{\Theta_d(t)}(x,x')\|_{op}$ has variance not larger than $O(1)$ from the operator norm of a matrix. The vector $\boldsymbol{a}_d(\boldsymbol{x})$ is defined as $\boldsymbol{a}_d(\boldsymbol{x}) = \left( \prod_{d'\neq d}(f_{\Theta_{d'}}(x_{d'}))_1, \cdots, \prod_{d'\neq d}(f_{\Theta_{d'}}(x_{d'}))_R \right)^\top$. Each element has variance $O(1)$, so the norm $\|\boldsymbol{a}_d(\boldsymbol{x})\|_{\ell_2}$ has variance not larger than $O(R)$. Its change norm $\|\boldsymbol{a}_d^t(\boldsymbol{x}) - \boldsymbol{a}_d^0(\boldsymbol{x})\|_{\ell_2}$ has variance not larger than $O(t)$ by the argument before and the neural network Gaussian process (Lee et al., 2018). By substituting all the variance bounds into (13), we see that $\frac{1}{R}\left|(\boldsymbol{a}_i^t - \boldsymbol{a}_i^0)\boldsymbol{K}_{i,j}^t\boldsymbol{a}_j^t\right|$ has variance not larger than $\frac{1}{R^2}O(t)O(1)O(R) = O(\frac{t}{R})$. $\frac{1}{R}\left|\boldsymbol{a}_i^0\boldsymbol{K}_{i,j}^t(\boldsymbol{a}_j^t - \boldsymbol{a}_j^0)\right|$ has variance not larger than $\frac{1}{R^2}O(R)O(1)O(t) = O(\frac{t}{R})$. $\frac{1}{R}\left|\boldsymbol{a}_i^0(\boldsymbol{K}_{i,j}^t - \boldsymbol{K}_{i,j}^0)\boldsymbol{a}_j^0\right|$ has variance $\frac{1}{R^2}O(R)O(\frac{t}{W})O(R) = O(\frac{t}{W})$. Hence the total error has variance not larger than $O(\frac{t}{R}) + O(\frac{t}{W})$. When $W, R \to \infty$, such variance vanishes and the total error $|K_{\Theta(t)}(\boldsymbol{x}_i, \boldsymbol{x}_j) - K_{\Theta(0)}(\boldsymbol{x}_i, \boldsymbol{x}_j)| \to 0$ almost surely. This implies that $K_{\Theta(t)}(\boldsymbol{x}_i, \boldsymbol{x}_j) \to K_{\Theta(0)}(\boldsymbol{x}_i, \boldsymbol{x}_j)$ for all $\boldsymbol{x}_i, \boldsymbol{x}_j$ in training data. $\qquad\square$

### A.12 DETAILED EXPERIMENTAL SETTINGS AND MORE RESULTS

**KRR.** Using the KRR of NTK could mimic the behavior of infinite-width neural networks because the training dynamic of neural networks with infinite width can be characterized by their corresponding NTK. Following (Geifman et al., 2024), we perform KRR by using the gradients of the neural network w.r.t. parameters as the feature function of the kernel. We test both MLP and CP SepNN, and compare the CP SepPGD with the classical NTK-based PGD, the modified spectrum kernel (MSK) (Geifman et al., 2024). Following (Geifman et al., 2024), we consider both noisy (standard deviation 0.01) and noiseless cases. The convergence behavior is measured by testing MSE during training. Because the efficiency advantage of SepNN and SepPGD comes from the lower complexity in an iteration, we plot the convergence curve w.r.t. execution time rather than iteration number. All experiments are conducted on an RTX 2080TI GPU. All experiments are conducted in the full-batch optimization setting to align with theories.

Specifically, given an initialization of network parameters $\Theta_0 \in \mathbb{R}^P$ (with $P$ denoting the number of parameters), let $\phi(\boldsymbol{x}) := \nabla_{\Theta_0} f_{\Theta_0}(\boldsymbol{x})$ and $h(\boldsymbol{x}, \Theta') := \langle \Theta', \phi(\boldsymbol{x}) \rangle$. Let $\boldsymbol{X}$ denote the batched inputs and $\boldsymbol{y}$ the batched labels. The KRR w.r.t. the preconditioned loss is

$$\min_{\Theta' \in \mathbb{R}^p} \frac{1}{2} \left\| \boldsymbol{S}^{1/2}(h(\boldsymbol{X}, \Theta') - \boldsymbol{y}) \right\|_2^2 + \frac{1}{2}\gamma \|\Theta'\|_{\ell_2}^2, \tag{15}$$

where $\boldsymbol{S}$ is the preconditioner in the NTK-based PGD. We use gradient descent to optimize the parameters $\Theta'$ to perform the KRR by using the NTK at initialization. We utilize the Adam optimizer (Kingma & Ba, 2015) with learning rate 0.001 and weight decay 0.0001, where the weight decay serves as the regularization term ($\ell_2$-norm of weight vector $\Theta'$) in KRR. The rank of the bivariate

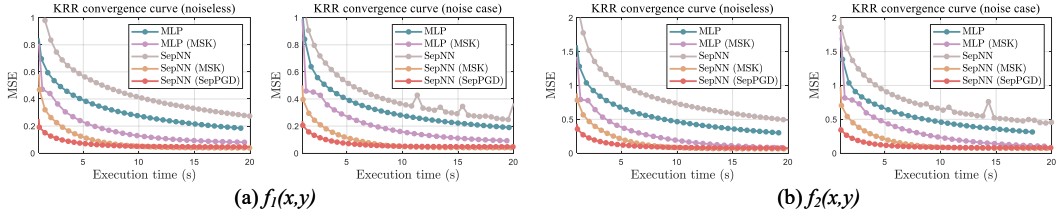

Figure 5: Convergence curves for KRR on data $f_1(x, y)$ and $f_2(x, y)$.

CP SepNN is set as $100$. The network width is set as $200$. The depth of MLPs is set to 3. The preconditioner of each method is updated once every 10 iterations. In the modulation function $g(\lambda)$, we set $k = 5$ for SepPGD and $k = 25$ for MSK (fined-tuned using trial-and-error). The activation function in MLPs and factor MLPs in SepNNs is set as ReLU for KRR. We use the following analytical functions from (Lee et al., 1997) to perform the KRR testing.

$$f_1(x, y) = \frac{1}{3} \exp\left(-\frac{81}{4}((x - 1/2)^2 + (y - 1/2)^2)\right), \tag{16}$$

$$f_2(x, y) = \frac{1.25 + \cos(5.4y)}{6 + 6(3x - 1)^2}, \tag{17}$$

$$f_3(x, y) = \frac{1}{9}(\tanh(9 - 9x - 9y) + 1). \tag{18}$$

We randomly select $30 \times 30 = 900$ training points and $30 \times 30 = 900$ testing points on grids in the interval $[-1, 1] \times [-1, 1]$. We add Gaussian noise with standard deviation $0.01$ on the training set in the noise case. Results are averaged across five random seeds. The result in Fig. 2 lists the convergence curves for $f_3(x, y)$. The results (convergence curves during training, i.e., MSE on testing sets during training) for $f_1$ and $f_2$ are shown in Fig. 5.

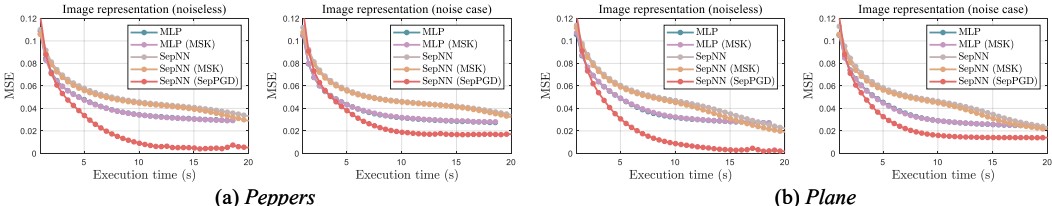

Figure 6: Convergence curves for image representation on downsampled data *Peppers* and *Plane*.

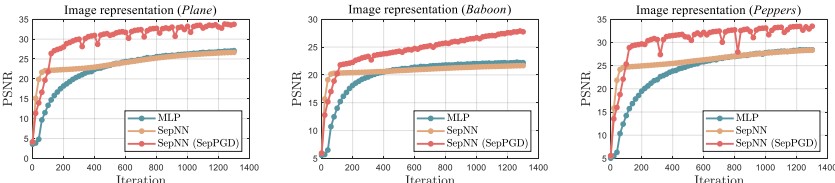

Figure 7: Convergence curves for image representation using PSNR vs. iteration.

**Image Representation and Recovery.** For image representation, we consider using the SepNN to approximate the image by feeding the coordinates of images into the SepNNs and output the image values, a convention in INRs (Sitzmann et al., 2020; Shi et al., 2025). We use the image datasets *Baboon*, *Peppers*, and *Plane*[4] ($512 \times 512 \times 3$) as shown in Figs. 10-12.

The experiments contain two parts. In the first part, we run MLP, MLP (MSK), CP SepNN, CP SepNN (MSK), and CP SepNN (SepPGD) for image representation (Fig. 2(b) and Fig. 6) on downsampled images with smaller resolution to fit the computational overhead, since the NTK calculation in MSK is expensive for images of large sizes. In the second part, we run MLP, SepNN, and SepNN (SepPGD) for image representation and recovery (inpainting) (Fig. 3 and Figs. 8-12) on the original image sizes. For the original size, the MSK run out of memory on our GPU and hence we omit

---

[4]https://sipi.usc.edu/database/database.php

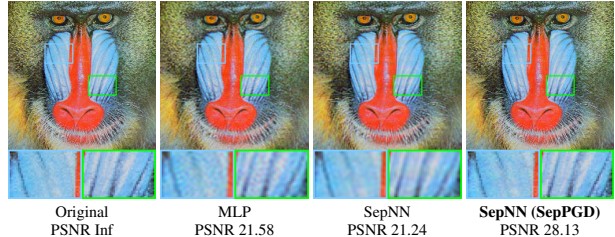

Figure 8: Visual results for image representation via INR and SepPGD on *Baboon*

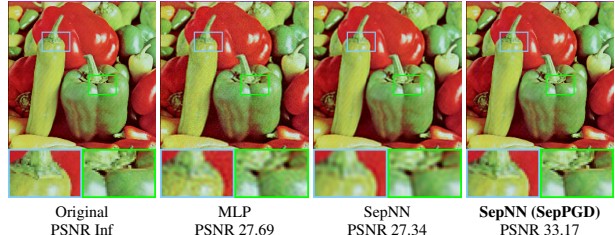

Figure 9: Visual results for image representation via INR and SepPGD on *Peppers*.

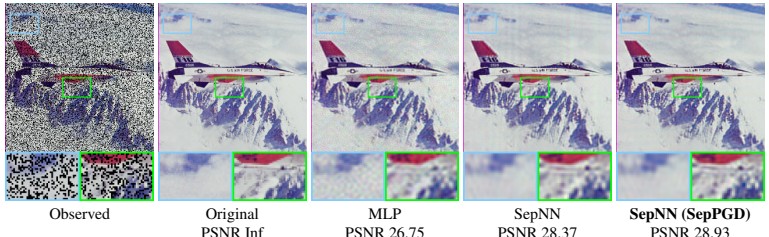

Figure 10: Visual results for image inpainting via INR and SepPGD on *Plane*.

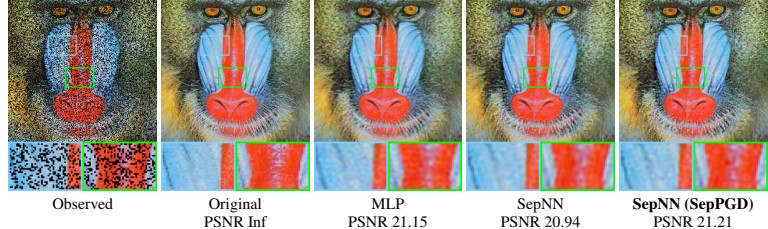

Figure 11: Visual results for image inpainting via INR and SepPGD on *Baboon*.

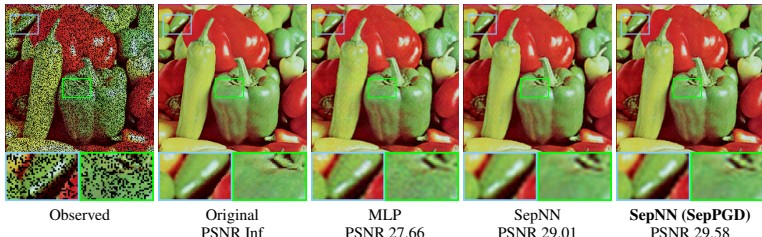

Figure 12: Visual results for image inpainting via INR and SepPGD on *Peppers*.

the comparison with MSK on original sized images. The network width is set to 150 for smaller images, a smaller value since the NTK calculation for MLP and MSK runs out of memory for larger width. We train the neural networks using INRs by feeding the image coordinates into the network and output the corresponding image value for image representation, following INR works (Sitzmann et al., 2020; Shi et al., 2025). For image recovery (i.e., image inpainting), we randomly sample 30% points in the image as incompleted entries. We then train the models on other 70% known pixels and predict the values of incompleted entries using the trained model for image inpainting. For the first part using smaller images, we report the image representation MSE w.r.t. execution time using convergence curves (Fig. 2(b) and Fig. 6) to show the convergence superiority of SepPGD. For the

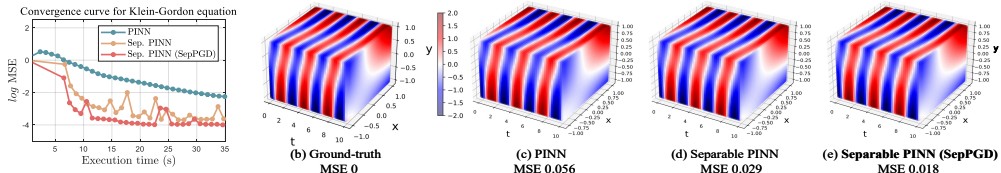

Figure 13: Convergence curves and results for the 3D Klein-Gordon equation using PINNs.

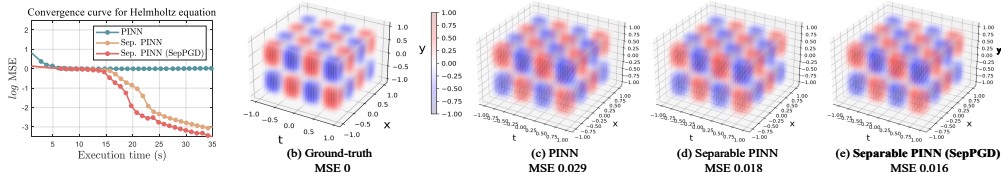

Figure 14: Convergence curves and results for the 3D Helmholtz equation using PINNs.

second part using larger images, we present visual results to display the recovery of image details and textures (Fig. 3 and Figs. 8-12).

In the first part using smaller images, we consider both noisy (standard deviation $0.01$) and noiseless cases. The convergence behavior is measured by image MSE during training. Because the efficiency advantage of SepNN and SepPGD comes from the lower complexity in an iteration, we plot the convergence curve w.r.t. execution time rather than iteration number (While we have also reported the convergence curves w.r.t. iteration for larger images; see Fig. 7). All experiments are conducted on an RTX 2080TI GPU. For all experiments of image representation, we utilize the Adam optimizer (Kingma & Ba, 2015) with learning rate $0.0001$ and without weight decay. For smaller images, the rank of the bivariate CP SepNN is set as $150$. The network width is set as $150$. For original sized images, the rank and network width are set to $500$. The depth of MLPs is set to $3$. The preconditioner of each method is updated once every 10 iterations. The images are fitted channel-by-channel using bivariate SepNNs. In the modulation function $g(\lambda)$, we set $k = 10$ for smaller images and $k = 70$ for original sized images. For MSK, we set $k = 25$ in the modulation function (fine-tuned using trial-and-error). All MLPs and factor MLPs in SepNNs are parameterized by the SIREN structure (Sitzmann et al., 2020), a well-studied method in INRs. For original sized images, we run 1300 iterations for all methods to ensure fair comparisons between their convergence behavior.

Convergence curves in Fig. 2(b) list the result for *Baboon*. The convergence curves for *Peppers* and *Plane* are shown in Fig. 6. We note that in these figures, MLP (MSK) does not obviously improve the convergence of MLP. This is because the MSK under full-batch condition requires generally larger computational costs for image data, resulting similar convergence speed as compared with the pure MLP. Image representation results in Fig. 3 and Figs. 8-9 (under the same iteration number for different methods) show that SepPGD effectively alleviates the spectral bias of SepNN to improve convergence, thus better capturing image details and structures. Figs. 10-12 (under the same iteration number for different methods) show that SepPGD does not affect the generalization ability of the SepNN for image inpainting, even leads to improvements in most time. This shows the robustness and effectiveness of the proposed SepPGD for preserving the generalization while accelerating training convergence. In Fig. 7, we show the PSNR vs. iteration curves using original sized images. SepPGD clearly enhances the convergence speed and alleviates spectral bias of the SepNN by observing the convergence curves.

**Surface Representation.** The 3D surface representation in Fig. 3 is conducted on the *Thai statue* dataset (Saragadam et al., 2023; Shi et al., 2025) ($304 \times 512 \times 262$ occupancy cube). In the modulation function $g(\lambda)$, we set $k = 20$ for SepPGD. All MLPs and factor MLPs in SepNNs are parameterized by the SIREN structure (Sitzmann et al., 2020), a well-studied method in INRs. We run 1000 iterations for all methods to ensure fair comparisons between their convergence behavior. We utilize the Adam optimizer (Kingma & Ba, 2015) with learning rate $0.005$ and without weight decay. The rank of the CP SepNN is set as $200$. The network width is set as $256$. The depth of MLPs is set to $3$. The preconditioner of each method is updated once every 10 iterations. Here, the CP SepNN and SepPGD are the three-dimensional case in Definition 1 ($D = 3$). Since the MSK run out of memory for full-batch settings in our platform, we do not compare this method for surface representation. Although we can run the MSK using mini-batch versions (Shi et al., 2025) (i.e., by

sampling a part of training samples in each iteration), we remark that our theory and algorithm are mainly derived for full-batch settings, and hence comparing with mini-batch methods is not mandatory. In contrast, our SepPGD can be efficiently implemented for full-batch algorithms by virtue of the $O(nD)$ complexity. Deriving the training dynamics and implicit regularization of mini-batch algorithms are promising directions for future researches.

**PINN.** Following the framework of separable PINNs as introduced in (Cho et al., 2023), our experiments are performed on the 3D diffusion, Helmholtz, and Klein-Gordon equations (Figs. 4, 13, and 14). For the detailed formulations of these equations, we refer readers to equations (51)–(59) in (Cho et al., 2023). Unlike the loss function used in (Cho et al., 2023), the physics-informed loss in our experiments contains four components: the PDE residual loss, the initial condition loss, the boundary condition loss, and an additional data loss. Here, the data loss corresponds to training the PINN model using a set of grid-based training points, which enables the application of the proposed SepPGD algorithm via the $\ell_2$-norm of the data loss. We also apply SepPGD to the initial and boundary condition losses. For the PDE residual loss, which involves derivatives, we do not employ the SepPGD algorithm, as extending PGD to derivative-based losses requires substantially different algorithmic treatment. We leave this promising direction to future work.

In the modulation function $g(\lambda)$, we set $k = 5$ in SepPGD for the data loss in PINNs, while setting $k = 2$ for boundary and initial losses after trial-and-error parameter tuning. All MLPs and factor MLPs in SepNNs are parameterized by the SIREN structure (Sitzmann et al., 2020) to enhance the convergence for PINNs. We run 5000 iterations for all methods to ensure fair comparisons between their convergence behavior. We utilize the Adam optimizer (Kingma & Ba, 2015) with learning rate 0.005 and without weight decay. The rank of the CP SepNN is set as 64. The network width is set as 64 as well. The preconditioners for the three loss functions (data, boundary, and initial loss functions) are computed at initialization and remain fixed throughout the training process to reduce computational cost, since we need to update three preconditioners for the PINN. For each example, we sample $16 \times 16 \times 16$ grid collocation and training points, while evaluating the trained models on $30 \times 30 \times 30$ grid testing points to test the generalization ability. The number of layers in all MLPs is set to 4 by following (Cho et al., 2023). The results in Figs. 4, 13, and 14 show that SepPGD has substantial potential in accelerating the convergence of separable PINNs.

**LLM use.** LLMs were rarely used in this paper to polish writing. All technical contents and conceptualizations were made by humans.

