# OpenReview forum: "Separable Neural Networks: Approximation Theory, NTK Regime, and Preconditioned Gradient Descent"
_ICLR.cc/2026/Conference — ICLR 2026 Poster_

### Official Review · Reviewer_WM7X · 2025-10-22

**Soundness:** 3
**Presentation:** 2
**Contribution:** 3
**Rating:** 6
**Confidence:** 3

**Summary:**

The paper studies real valued separable neural networks (SepNNs), an alternative architecture which combines some univariate functions linearly to represent multivariate functions. SepNNs combine univariate factor functions $f_{\theta_i}(x_i)$, with linearity $L$ (equation 1). Authors considered three cases of this $L$ function, CP, TT, Tucker.
The paper build up on three claims:
1. Similar universal approximation guarantees of neural networks in terms of expressivity for SepNNs.
2. Analyzing neural tangent kernels in the case where rank or width goes to infinity and it establishes that under infinite width and rank, NTK converges to a deterministic kernel and under infinite width and finite fixed rank it converges to a random kernel
3. Using theoretical analysis of NTK they propose a training algorithm (SepPGD) that reshapes the spectrum with O(nD) preconditioning cost on n grids per axis reaching faster convergence in tasks KRR, INRs, PINNs.

**Strengths:**

The most interesting thing about this paper is how they use their theoretical analysis on optimization of SepNNs and the limitations they found to come up with a prescription as a training algorithm.

Their universal approximation theorem nicely shows the expressivity power of SepNNs. Also, the proof sketch is nicely and understandably stated in the main body and it’s intuitive enough for readers. (Theorem 1).

They use NTK to study optimization regimes under asymptotic assumptions (Theorem2, lemma1) and they characterize when training dynamics are kernel-like vs. stochastic.

They use spectral bias already known results to argue that SepNN suffers from a similar problem, this is not particularly novel but the discussion of spectral bias and how they propose a new training algorithm inspired by similar works for SepNN is nice.

The training algorithm (SepPGD) is a slightly complicated but good-sounding idea. The authors support this algorithm with nice theoretical analysis showing equivalence to full NTK-PGD step with smoothened eigenvalue spectrum which improves convergence.

**Weaknesses:**

In terms of motivation, I found the introduction to be a bit lacking. The authors don’t clearly discuss the advantages of SepNNs.
More thoroughly, the motivation for SepNNs is not well positioned against established efficient architectures. The paper frames SepNNs as a new paradigm but does not convincingly argue why this approach matters beyond reinterpreting classical separability. Without this context, the work risks appearing incremental from the perspective of practical ML.

The authors mentioned computational complexity: the paper mentions that SepNNs need only $$O(nD) $$ computations (n being number of data and D being dimension). They compare it to computational complexity of NN being $$O(n^D)$. This is true in terms of unique function evaluations but is does not imply faster training since regular models for example a CNN do their computation of all the dimension D in parallel. My impression is that SepNNs’ benefits are limited to high dimensional, low sample problems, with structured scientific problems in terms of learnable parameters and memory efficiency; but the authors don’t discuss that.

Theorem 1, does not say anything about rank, for example there’s no rate describing rank R must grow with target smoothness of dimension or something similar to that. This limits the theorem's relevance for practical rank selection.

The paper lacks theory for fixed-rank regimes, which is the setting that actually matters in practice. The entire motivation mentioned for using SepNNs is to have low rank structure to achieve parameter efficiency, scalability and interpretability, but the theory for this is limited in terms of convergence guarantees, or bound for generalization errors.

The random NTK in the fixed rank case is mathematically similar to random features maps. It would be nice for authors to make this connection and use the well-studied random feature maps. This can address the lack between asymptotic assumptions in theorem and finite rank networks used in experiments.

The preconditioner choices appear to be heuristics and there is no discussion/guarantee on what their condition number might be, or their possible effect on generalization.

The experiment focuses on grid structured problems; adding other experiments, with non-grid tasks (e.g. tabular inputs) might strengthen things. This is just a suggestion, rather than a requirement.

Some reproducibility details are missing.

Some experiments:
Ablations study on sensitivity to rank, also k in the eigen-leveling and update frequency, choice of activation and width.
Some baseline are missing specially fourier feature maps.
Some experiments on robustness to noise and overfitting behaviors of SepPGD

The claimed efficiency of SepPGD neglects the high upfront cost of constructing the preconditioner. The claim of $O(nD)$ refers to applying the preconditioner while constructing them seems expensive. Also it requires a lot of memory to store these pre-conditioned matrices $O(Dn^2)$ which makes the method not scalable for a higher amount of data.

What if the target is not well-separable? It would be nice to have approximation-error curves vs true separation rank of the target (synthetics with controllable TT/Tucker rank). Or some similar attempt to address the real-world misspecification gap.

**Questions:**

I find the definition of TT and Tucker not well placed in the article and equation 2, 3 lacks some notations, since it’s not obvious what are R and the tensor product and what are $f_{\theta_i}_{i_{i-1}, r_i}$, and $C$, maybe they can be moved to theorem 1 ot a preliminaries section.

The claim in remark 1 is not obvious to me, how can the results be extended to multi layers?

Line 337 is incomplete, there seems to be a missing sentence.

What is the computational cost for calculating the preconditioner?

In the fixed rank regime is it possible to bound $$||K(t) - K(0)|| over finite time with small step sizes?

How sensitive is SepPHD to the choice of k?

Could Lemma 2 be extended to non-grid inputs?

---

> ### Author Response · Authors · 2025-11-19
> **Response to Reviewer WM7X**
>
> ***To Reviewer WM7X***
>
> We sincerely thank the reviewer for the valuable and constructive comments! We have carefully incorporated additional clarifications in response to the feedback. Please see the details below.
>
> >**W1: In terms of motivation, I found the introduction to be a bit lacking. The authors don’t clearly discuss the advantages of SepNNs. More thoroughly, the motivation for SepNNs is not well positioned against established efficient architectures. The paper frames SepNNs as a new paradigm but does not convincingly argue why this approach matters beyond reinterpreting classical separability. Without this context, the work risks appearing incremental from the perspective of practical ML.**
>
> Thanks for your comment. We agree that the efficiency advantages of SepNNs need to be more clearly positioned against classical efficient architectures. Certainly, many efficient architectures have been proposed for deep neural networks (DNNs). Among these, a classical line of work employs tensor decomposition to factorize the weights of DNNs, thereby reducing the number of parameters. This category is indeed the most closely related to SepNNs. However, SepNNs are motivated by a fundamentally different idea. Rather than decomposing the network weights, the principle of SepNNs is to separate the input vector into multiple smaller inputs and process each input using factor networks. This design uniquely improves efficiency in scenarios involving coordinate-based neural networks and function evaluations on grids by reusing the factor outputs. Such a structure is particularly advantageous in applications like implicit neural representations (INRs), which map coordinates to pixel values, and physics-informed neural networks (PINNs), which map coordinates to physical fields. Beyond these, SepNNs have shown promising utility in a range of scientific domains, including separable operator learning [R1], wireless communication [R2], Earth science [R3], and bioinformatics [R4] (Note that in these applications, SepNNs are utilized to efficiently evaluate grid points). We therefore believe that the structure and efficiency of SepNNs hold important value across a variety of practical machine learning applications. We will incorporate the additional clarifications on the motivations and advantages of SepNNs in the Introduction Section of the revised manuscript. Thanks.
>
> [R1] Separable Operator Networks, TMLR
>
> [R2] Constructing 4D Radio Map in LEO Satellite Networks with Limited Samples, INFOCOM
>
> [R3] Full-Waveform Inversion With Velocity Model Low-Rank Implicit Neural Representation, TGRS
>
> [R4] GNTD: reconstructing spatial transcriptomes with graph-guided neural tensor decomposition informed by spatial and functional relations, Nature Communications
>
> >**W2: The authors mentioned computational complexity: the paper mentions that SepNNs need only $O(nD)$ computations ($n$ being number of data and $D$ being dimension). They compare it to computational complexity of NN being $O(n^D)$. This is true in terms of unique function evaluations but is does not imply faster training since regular models for example a CNN do their computation of all the dimension $D$ in parallel. My impression is that SepNNs’ benefits are limited to high dimensional, low sample problems, with structured scientific problems in terms of learnable parameters and memory efficiency; but the authors don’t discuss that.**
>
> Thank you for this insightful comment. You are right that the computational advantage of SepNNs is most relevant in certain structured settings, and does not directly imply faster training in all scenarios compared to architectures like CNNs. We would like to clarify that SepNNs are primarily designed for problems where the input is a set of coordinate points on a grid, such as in INRs and PINNs. In these cases, the input is not a high-dimensional signal like an image, but rather a low-dimensional coordinate space. CNNs, while highly efficient for image-like data through localized convolution and parallelism, are less naturally suited to such coordinate-based inputs, as they rely on spatial locality and translation invariance, which may not align with the structure of the problem. In contrast, SepNNs exploit the separable structure of the grid to reduce the number of unique function evaluations from $O(n^D)$ (as in a standard MLP applied naively to all grid points) to $O(nD)$, resulting in faster training on grid coordinate points. This makes SepNNs particularly advantageous in training coordinate networks on grids, such as image representation using INRs and evaluating grid collocation points in PINNs. We agree that the benefits of SepNNs are most pronounced in these structured settings, and we will revise the Introduction Section to more clearly describe the scope of such efficiency.

---

> > ### Author Response · Authors · 2025-11-19
> > **Response to Reviewer WM7X**
> >
> > >**W3: Theorem 1, does not say anything about rank, for example there’s no rate describing rank $R$ must grow with target smoothness of dimension or something similar to that. This limits the theorem's relevance for practical rank selection.**
> >
> > Thank you for raising this point. Indeed, our current theoretical analysis establishes the universal approximation capability of SepNNs and does not yet provide explicit approximation error rates in terms of network rank. In future work, we can consider function spaces spanned by Fourier bases and leveraging the decay properties of Fourier coefficients [R5] to derive approximation error bounds with respect to the rank, in order to provide theoretical guidance for practical rank selection.
> >
> > [R5] Russell L. Herman, Generalized Fourier Series and Function Spaces, Chapman and Hall/CRC
> >
> > >**W4: The paper lacks theory for fixed-rank regimes, which is the setting that actually matters in practice. The entire motivation mentioned for using SepNNs is to have low rank structure to achieve parameter efficiency, scalability and interpretability, but the theory for this is limited in terms of convergence guarantees, or bound for generalization errors.**
> >
> > Thank you for raising this point. Indeed, our current theoretical analysis does not yet provide comprehensive convergence guarantees or generalization error bounds for the fixed-rank regime. We agree that such analysis is crucial for a deeper understanding of the training and generalization behaviors of SepNNs. Still, we believe the present theoretical framework offers a valuable initial insight, as it establishes a convergence analysis for SepNNs under infinite rank regime. In fact, by applying probability bounds related to the law of large numbers with respect to the rank parameter $R$, we anticipate that an $\epsilon$-convergence bound on the training residual could be derived for a fixed rank $R = O(1/\epsilon^2)$, in a manner analogous to classical probability convergence bound analyses for wide MLPs (see, e.g., Theorem 4.1 in [R6]). In future work, we can leverage more advanced tools from stochastic NTK analysis and explore training dynamics beyond the convergent NTK regime [R7], to provide a more comprehensive theoretical understanding on convergence and generalization of SepNNs under small rank. Thanks.
> >
> > [R6] Fine-Grained Analysis of Optimization and Generalization for Overparameterized Two-Layer Neural Networks, ICML
> >
> > [R7] On the linearity of large non-linear models: when and why the tangent kernel is constant, NeurIPS
> >
> > >**W5: The random NTK in the fixed rank case is mathematically similar to random features maps. It would be nice for authors to make this connection and use the well-studied random feature maps. This can address the lack between asymptotic assumptions in theorem and finite rank networks used in experiments.**
> >
> > Thanks for this constructive comment! Indeed, in the fixed-rank regime, the factor MLPs of the SepNN converge to random Gaussian processes. This implies that their outputs serve as random feature maps for each input. The SepNN then performs a tensor product of these random features. This connection indeed allows us to consider the well-established theoretical framework of random feature models [R8] to analyze the potential property of SepNNs under fixed-rank settings, providing a way to derive non-asymptotic convergence rates and generalization bounds for fixed-rank SepNN. This is a great idea that can be investigated in future research. Also, we will discuss this connection and its theoretical implications in the revised manuscript to strengthen the foundation for the fixed-rank SepNNs used in our experiments. Thanks.
> >
> > [R8] Random features for large-scale kernel machines, NeurIPS

---

> > > ### Author Response · Authors · 2025-11-19
> > > **Response to Reviewer WM7X**
> > >
> > > >**W6: The preconditioner choices appear to be heuristics and there is no discussion/guarantee on what their condition number might be, or their possible effect on generalization.**
> > >
> > > Thanks. First, we would like to clarify that the preconditioner in our work is designed to modulate the eigenvalue distribution of the NTK matrix $K$ to accelerate training convergence. By applying the modulation function $g(\lambda)$ to smooth the top $k$ eigenvalues, the resulting preconditioned product matrix $K {S}$ is guaranteed to have a condition number no larger than $K$. This is because the dominant large eigenvalues are reduced, leading to a more uniform spectral distribution. In future work, we can derive quantitative bounds on the condition number of the modulated NTK matrix by leveraging its eigenvalue decay properties. For instance, tools from [R9] on the eigenvalue decay rates of kernel functions could be employed. We regard this as a promising direction for further research. Regarding the effect of the preconditioner on generalization, existing studies (e.g., [R10, R11]) suggest that slowing the eigenvalue decay of the kernel matrix can introduce an implicit regularization effect, helping prevent overfitting and improving generalization. While the current manuscript does not include a rigorous theoretical analysis of generalization, the above references provide a way for establishing formal generalization guarantees for SepPGD. We will include these discussions in the revised manuscript. Thanks!
> > >
> > > [R9] On the Eigenvalue Decay Rates of a Class of Neural-Network Related Kernel Functions Defined on General Domains, JMLR
> > >
> > > [R10] Controlling the Inductive Bias of Wide Neural Networks by Modifying the Kernel’s Spectrum, TMLR
> > >
> > > [R11] Benign overfitting in ridge regression, JMLR
> > >
> > > >**W7: The experiment focuses on grid structured problems; adding other experiments, with non-grid tasks (e.g. tabular inputs) might strengthen things. This is just a suggestion, rather than a requirement.**
> > >
> > > Thank you for this suggestion. We have further conducted experiments on non-grid tasks using INRs. Specifically, we randomly sampled 500 non-grid points along with their corresponding function values ($f_1(x,y), f_2(x,y), f_3(x,y)$ from Eqs. (16)–(18) in the manuscript) to form the non-grid training dataset. We then train a SepNN to fit these non-grid samples, as shown in the Table below. For non-grid data, the proposed SepPGD gradient is formulated as $\nabla_{\Theta_d} \langle f_{d}(X_{d,:}), M_d \rangle$, where $M_d$ is constructed by some Einstein products between factor outputs, $S_d$, and training residual $r$ (see our response to your comment Q7). This formulation reduces to point-wise evaluations and yields similar efficiency and effectiveness to classical NTK-based preconditioning methods, such as the modifying spectrum kernel (MSK). Therefore, it is reasonable that SepNN (SepPGD) performs comparably to SepNN (MSK) on non-grid data, while both outperform the original SepNN. The efficiency advantage of SepPGD comes primarily from the separable structure of grid inputs, which is a prevalent training configuration in applications such as image grids in INRs and grid collocation points in PINNs.
> > >
> > > Table: MSE performance of SepPGD for non-grid data $f_1(x,y),f_2(x,y),f_3(x,y)$ by learning function representations using INRs.
> > > |Data| $f_1(x,y)$| $f_2(x,y)$| $f_3(x,y)$|
> > > |-|-|-|-|
> > > | SepNN | 0.020| 0.030 |  0.075 |
> > > | SepNN (MSK) | 0.014| 0.024 | 0.058 |
> > > | SepNN (SepPGD) | 0.015| 0.021 | 0.062 |

---

> > > > ### Author Response · Authors · 2025-11-19
> > > > **Response to Reviewer WM7X**
> > > >
> > > > >**W8: Some reproducibility details are missing. Some experiments: Ablations study on sensitivity to rank, also k in the eigen-leveling and update frequency, choice of activation and width. Some baseline are missing specially Fourier feature maps. Some experiments on robustness to noise and overfitting behaviors of SepPGD.**
> > > >
> > > > Thanks. Sensitivity analysis to rank $R$ and $k$ in the eigenvalue modulation function are shown in the Tables below (i.e., Table 3 in the manuscript Appendix). For varying ranks $R$, our SepPGD consistently enhances the performance of SepNN. We find that setting $k > 60$ in this case effectively mitigates the spectral bias in SepNN by adjusting the NTK spectrum. In particular, values of $k \in \\{60, 70, 80,90\\}$ produce a desirable effect for enhancing convergence efficiency. Our SepPGD method is not overly sensitive to the choice of $k$.
> > > >
> > > > Table: Sensitivity analysis (PSNR) of rank $R$ for image representation using INRs.
> > > > | Rank $R$ | 100   | 200   | 300   | 400   | 500   | 600   | 700   |
> > > > |-|-|-|-|-|-|-|-|
> > > > | SepNN  | 26.36 | 26.74 | 26.54 | 26.61 | 26.48 | 26.49 | 26.45 |
> > > > | SepNN (SepPGD) | 31.59 | 32.69 | 33.41 | 33.65 | 33.30 | 33.54 | 33.40 |
> > > >
> > > > Table: Sensitivity analysis (PSNR) of $k$ for image representation using INRs.
> > > > |$k$| 30   | 40   | 50   | 60   | 70   | 80   | 90   |
> > > > |-|-|-|-|-|-|-|-|
> > > > | SepNN       | 26.48 | 26.48 | 26.48 | 26.48 | 26.48 | 26.48 | 26.48 |
> > > > | SepNN (SepPGD) | 26.79 | 26.95 | 29.65 | 33.04 | 33.30 | 34.28 | 34.21 |
> > > >
> > > > The sensitivity analysis of our SepPGD w.r.t. the preconditioner update frequency is shown in the Table below. SepPGD is relatively not sensitive to the preconditioner update frequency. This is because the NTK matrix would not change abruptly during training and thus the preconditioner is effective for NTK eigenvalue modulation across a number of iterations.
> > > >
> > > > Table. PSNR performance on image representation using INRs with different preconditioner update frequency (number of iterations between preconditioner updates).
> > > > |Update frequency|10|100|300|500|700|900|1100|
> > > > |-|-|-|-|-|-|-|-|
> > > > |SepNN|26.48|26.48|26.48|26.48|26.48|26.48|26.48|
> > > > |SepNN (SepPGD)| 33.30|33.29|33.16|33.12|33.09|33.06|33.06|
> > > >
> > > > The performance of SepPGD with different activation functions as well as the Fourier feature mapping [R12] is shown in the Table below. SepPGD improves the convergence speed of the SepNN baseline with different activation functions and Fourier encoding structures.
> > > >
> > > > Table. PSNR performance on image representation using INRs with different activation function/network structure.
> > > > |Activation|Sin|Cos|ReLU|LeakyReLU|Fourier feature+ReLU|Fourier feature+ LeakyReLU |
> > > > |-|-|-|-|-|-|-|
> > > > |SepNN|26.48|21.83|18.20|18.20|30.89|30.94|
> > > > |SepNN (SepPGD)|33.30|30.06|20.02|20.74|40.49|40.51|
> > > >
> > > > [R12] Fourier Features Let Networks Learn High Frequency Functions in Low Dimensional Domains, NeurIPS
> > > >
> > > > The performance with different width is shown in the Table below. With a sufficient width (e.g., width$>200$), the SepPGD is effective in accelerating the convergence of SepNN.
> > > >
> > > > Table. PSNR performance on image representation using INRs with different network width.
> > > > |Width|100| 200| 300| 400| 500| 600| 700|
> > > > |-|-|-|-|-|-|-|-|
> > > > |SepNN|22.46|23.88|25.21|26.19|26.48|27.08|27.20|
> > > > |SepNN (SepPGD)| 22.79|31.46|32.75|33.22|33.30|33.89|34.35|
> > > >
> > > > The experimental results for the image representation under additive Gaussian noise are shown in the Table below. Under moderate to low noise deviation, SepPGD demonstrates robustness and improves the denoising performance of SepNN. However, when the noise deviation is large, SepPGD tends to overfit to the noise components. This behavior is expected, as SepPGD accelerates convergence toward high-frequency details, which can cause it to fit high-frequency noise more rapidly when the noise level is high and no explicit regularizations are applied. In future work, we can introduce suitable regularizations (e.g., total variation) to enhance robustness in noisy scenarios.
> > > >
> > > > Table. PSNR performance on image representation using INRs with Gaussian noise.
> > > > |Noise standard deviation|0.01|0.03|0.05|0.07|0.09|
> > > > |-|-|-|-|-|-|
> > > > |SepNN|26.40|26.35|26.27|25.85|25.29|
> > > > |SepNN (SepPGD)|33.26|31.08|28.20|26.35|25.06|

---

> > > > > ### Author Response · Authors · 2025-11-19
> > > > > **Response to Reviewer WM7X**
> > > > >
> > > > > >**W9: The claimed efficiency of SepPGD neglects the high upfront cost of constructing the preconditioner. The claim of $O(nD)$ refers to applying the preconditioner while constructing them seems expensive. Also it requires a lot of memory to store these pre-conditioned matrices $O(Dn^2)$ which makes the method not scalable for a higher amount of data.**
> > > > >
> > > > > Thank you for this important comment. Yes, the $O(nD)$ complexity mainly refers to the application of the preconditioner during each iteration. We would like to further clarify that the preconditioner construction stage in SepPGD is also more efficient than classical NTK-based PGD methods under the grid input setting.
> > > > > Specifically, SepPGD constructs the preconditioner by calculating the NTK matrix and performing eigenvalue decomposition for $D$ factor matrices, each of size $n \times n$. In contrast, classical NTK-based PGD requires the same operations on a single large matrix of size $n^D \times n^D$. Therefore, the preconditioner construction complexity for SepPGD scales as $O(D(n^3 + n^2P))$ (note that NTK matrix calculation consumes $O(n^2P)$ and eigenvalue decomposition consumes $O(n^3)$), where $P$ is the number of network parameters, while the classical NTK-based PGD method scales as $O(n^{3D} + n^{2D}P)$. We therefore see that SepPGD is more efficient in preconditioner construction. In practice, the preconditioner is updated once every 10 iterations, and hence both the complexity of preconditioner construction and application are important for efficient training. Regarding memory, SepPGD requires $O(Dn^2)$ storage for preconditioning matrices as suggested by the reviewer. This is more efficient than the $O(n^{2D})$ storage needed by the classical NTK-based PGD method, which stores an $n^D \times n^D$ preconditioner matrix. In practice, this $O(Dn^2)$ storage of SepPGD is relatively modest compared to other storage components of neural network optimization, such as gradient storage. We have included these discussions in terms of computational and storage complexity of the proposed SepPGD method to enhance completeness of the revised manuscript. Thanks.
> > > > >
> > > > > >**W10: What if the target is not well-separable? It would be nice to have approximation-error curves vs true separation rank of the target (synthetics with controllable TT/Tucker rank). Or some similar attempt to address the real-world misspecification gap.**
> > > > >
> > > > > Thanks for this constructive suggestion. As recommended, we employ SVD truncation to synthesize images with varying ranks, and then apply a SepNN model with a fixed rank of $R=100$ to represent these images via INRs. The relationship between the approximation error (MSE) and the true separation rank of the target image is given in the table below. It can be observed that as the image rank increases, the approximation error of SepNN also grows. In contrast, SepPGD effectively enhances convergence efficiency (and consequently reduces the approximation error) across different data ranks.
> > > > >
> > > > > Table. Approximation error (MSE) vs. true separation rank of the target image data (Setting the model rank $R=100$) using a bivariate CP SepNN.
> > > > > |Data rank|100|200|300|400|500|
> > > > > |-|-|-|-|-|-|
> > > > > |SepNN| 0.052 | 0.061 | 0.063| 0.064 |0.066|
> > > > > |SepNN (SepPGD)| 0.021 | 0.030 | 0.035 | 0.036 | 0.037 |
> > > > >
> > > > > >**Q1: I find the definition of TT and Tucker not well placed in the article and equation 2, 3 lacks some notations, since it’s not obvious what are $R$ and the tensor product and what are $f_{\theta_i}(x_i)_{r_{i-1}, r_i}$ and $\mathcal C$, maybe they can be moved to theorem 1 or a preliminaries section.**
> > > > >
> > > > > Thanks. As suggested, we have provided additional clarifications for the notations used in TT and Tucker models. Especially, ${\mathcal C}\in{{\mathbb R}}^{R_1\times\cdots\times R_D}$ denotes the core tensor in the Tucker decomposition model, $(R_1,\cdots,R_{D})$ denotes the TT or Tucker rank of the model, $\times_d:{\mathbb R}^{R_1\times\cdots\times R_D}\times{\mathbb R}^{R_d}\to{\mathbb R}^{R_1\times\cdots \times R_{d-1}\times R_{d+1}\times\cdots\times R_D}$ denotes the tensor product between a tensor and a vector. More detailed definitions are provided in the Introduction Section of the revised manuscript for completeness. Thanks.

---

> > > > > > ### Author Response · Authors · 2025-11-19
> > > > > > **Response to Reviewer WM7X**
> > > > > >
> > > > > > >**Q2: The claim in remark 1 is not obvious to me, how can the results be extended to multi layers?**
> > > > > >
> > > > > > Thanks. For SepNNs with multi-layer factor neural networks, we can use the corresponding NTK analysis for multi-layer neural networks to study their theoretical property [R13]. Especially, [R13] has shown that the NTK of a multi-layer neural network (beyond two-layer) converges to a deterministic kernel under infinite width. Their results are based on the recursive formulation of the covariance matrix of hidden layers in a multi-layer network (cf. Theorem 3.1 in [R13]). By borrowing this type of technique, we believe the theoretical result for SepNN (Theorem 2 in our manuscript) can be readily extended to multi-layer MLPs by replacing the two-layer factor NTK $k(x_d,x_d’)$ with multi-layer NTK formulations. Thanks.
> > > > > >
> > > > > > [R13] On exact computation with an infinitely wide neural net, NeurIPS
> > > > > >
> > > > > > >**Q3: Line 337 is incomplete, there seems to be a missing sentence.**
> > > > > >
> > > > > > Thanks for your careful reading. The incomplete expression was likely due to sentence segmentation across adjacent pages. We have revised the formatting to prevent any potential confusion.
> > > > > >
> > > > > > >**Q4: What is the computational cost for calculating the preconditioner?**
> > > > > >
> > > > > > Thanks. SepPGD constructs the preconditioner by calculating the NTK matrix and performing eigenvalue decomposition for $D$ factor matrices, each of size $n \times n$. In contrast, the classical NTK-based PGD requires the same operations on a single large matrix of size $n^D \times n^D$. Therefore, the preconditioner construction complexity for SepPGD scales as $O(D(n^3 + n^2P))$ (note that NTK matrix calculation consumes $O(n^2P)$ and eigenvalue decomposition consumes $O(n^3)$), where $P$ is the number of network parameters, while the classical NTK-based PGD method scales as $O(n^{3D} + n^{2D}P)$. Hence, SepPGD is more efficient in preconditioner construction.
> > > > > >
> > > > > > >**Q5: In the fixed rank regime is it possible to bound $||K(t) - K(0)||$ over finite time with small step sizes?**
> > > > > >
> > > > > > Yes, it is possible to bound the NTK matrix change norm $||K(t) - K(0)||$ by Chebyshev-type inequality through leveraging the variance of $||K(t) - K(0)||$ related to $O(\frac{t}{R})$ with a fixed $R$ and small step sizes. Indeed, when $R\to\infty$, the variance vanishes and $K(t)$ converges to $K(0)$ almost surely under infinitely small step sizes. With a fixed $R$ and small step sizes, we can instead use the Chebyshev inequality to quantify the error $||K(t) - K(0)||$ using probability bounds related to $R$ and the step size.
> > > > > >
> > > > > > >**Q6: How sensitive is SepPGD to the choice of $k$?**
> > > > > >
> > > > > > Thanks. The sensitivity analysis to $k$ is shown in the Table below (i.e., Table 3 in the manuscript Appendix). We find that setting $k > 60$ in this case effectively mitigates the spectral bias in SepNN by adjusting the NTK spectrum. In particular, values of $k \in \\{60, 70, 80,90\\}$ produce a desirable effect for enhancing convergence efficiency. Our SepPGD method is not overly sensitive to the choice of $k$.
> > > > > >
> > > > > > Table: Sensitivity analysis (PSNR) of $k$ for image representation using INRs.
> > > > > > |$k$| 30   | 40   | 50   | 60   | 70   | 80   | 90   |
> > > > > > |-|-|-|-|-|-|-|-|
> > > > > > | SepNN       | 26.48 | 26.48 | 26.48 | 26.48 | 26.48 | 26.48 | 26.48 |
> > > > > > | SepNN (SepPGD) | 26.79 | 26.95 | 29.65 | 33.04 | 33.30 | 34.28 | 34.21 |
> > > > > >
> > > > > > >**Q7: Could Lemma 2 be extended to non-grid inputs?**
> > > > > >
> > > > > > Yes, Lemma 2 can be extended to non-grid inputs. Especially, if we construct the preconditioner as $S=\sum_d S_d$, then the SepPGD gradient $\nabla_{\Theta_d}\langle f_{d}(X_{d,:}), M_d\rangle$ (where $M_d$ is constructed by some Einstein products between factor outputs, $S_d$, and training residual $r$) is equivalent to the classical NTK-based PGD update $\nabla_\Theta f_\Theta(X)^T Sr$ for non-grid inputs $X \in{\mathbb R}^{D\times n}$ (element-wise evaluation for $n$ non-grid points in $D$-dimensional space without Kronecker product structure). We will include this discussion under Lemma 2 in the revised manuscript.
> > > > > >
> > > > > > Once again, we thank the reviewer for these insightful comments, which greatly help to enhance the comprehensiveness of our manuscript and can guide our efforts toward further theoretical investigation. We will include the additional discussions in the revised manuscript, and all revisions will be highlighted in blue for your convenience. Should you need further information, please let us know. We look forward to hearing from your feedback!

---

### Official Review · Reviewer_3E8r · 2025-10-27

**Soundness:** 3
**Presentation:** 3
**Contribution:** 3
**Rating:** 8
**Confidence:** 4

**Summary:**

This work makes contributions to theoretically understanding SepNN. First, using Weierstrass-based approximation and universal approximation theory, they prove that SepNN can approximate any multivariate function with arbitrary precision, confirming its representation completeness. Second, they derive the neural tangent kernel (NTK) regimes for SepNN, showing that the NTK of infinite-width SepNN converges to a deterministic (or random) kernel under infinite (or fixed) decomposition rank, with corresponding convergence and spectral bias characterization. Third, they propose an efficient separable preconditioned gradient descent (SepPGD) for optimizing SepNN, which alleviates the spectral bias of SepNN by provably adjusting its NTK spectrum. The SepPGD enjoys an efficient order of complexity and is much more efficient than previous neural network PGD methods. Extensive experiments for kernel ridge regression, image and surface representation using INRs, and numerical PDEs using PINNs validate the efficiency of SepNN and the effectiveness of SepPGD for alleviating spectral bias.

**Strengths:**

The paper is well written. Ideas are well presented.

**Weaknesses:**

No obvious weakness.

**Questions:**

No question.

---

> ### Author Response · Authors · 2025-11-19
> **Response to Reviewer 3E8r**
>
> We sincerely thank the reviewer for the valuable comments!

---

### Official Review · Reviewer_A9Bg · 2025-11-01

**Soundness:** 3
**Presentation:** 3
**Contribution:** 2
**Rating:** 4
**Confidence:** 3

**Summary:**

This paper provides a comprehensive theoretical and algorithmic framework for Separable Neural Networks (SepNNs), a class of efficient models that factorize a multivariate function into a combination of univariate functions. The main contributions are:
1. Approximation Theory:Proving that SepNNs are universal approximators, capable of approximating any continuous multivariate function with arbitrary precision.
2. NTK Analysis:Deriving the Neural Tangent Kernel (NTK) for SepNNs, characterizing their training dynamics and inherent spectral bias.
3. Efficient Algorithm:Proposing a novel Separable Preconditioned Gradient Descent (SepPGD)method that effectively alleviates spectral bias with significantly lower computational cost than existing preconditioning methods.

**Strengths:**

The authors performed an analysis of the sepNN parallel to the fully-connected neural networks. They built the uniform approximation theorem, the NTK regime, and the gradient descent.

**Weaknesses:**

My major concern with this work is that I fail to see the potential impact of developing this theory for separable neural networks (SepNNs). After several years of developments in NTK research, this type of work feels somewhat routine.

If the author could explain a little bit on the potential application and the near future goal of this theory, I would be happy to increase the score.

**Questions:**

Same to the weakness.

---

> ### Author Response · Authors · 2025-11-13
> **Response to Reviewer A9Bg**
>
> ***To Reviewer A9Bg***
>
> We sincerely thank the reviewer for the valuable and constructive comments! We have carefully incorporated additional discussions and clarifications in response to the feedback, particularly regarding the potential impact and applications of our work. Please see the details below.
>
> >**W1: My major concern with this work is that I fail to see the potential impact of developing this theory for separable neural networks (SepNNs). After several years of developments in NTK research, this type of work feels somewhat routine.**
>
> Thank you for raising this concern. We appreciate the opportunity to further elaborate on the potential impact of our proposed theory and method. Below, we highlight its significance from three perspectives.
>
> 1.	**Current Applications.** Separable neural networks (SepNNs) have been attracting growing attention in various applications due to their efficient structure, such as in implicit neural representations (INRs) and physics-informed neural networks (PINNs). Motivated by our theoretical analysis of the NTK for SepNNs, we introduced the separable preconditioned gradient descent (SepPGD) algorithm to accelerate the training convergence of such networks. Numerical experiments demonstrate that SepPGD can effectively speed up convergence in applications involving INRs and PINNs (e.g., in image and surface representation and numerical PDEs). This is particularly important for applications in computer vision and graphics community that rely on INRs, where alleviating spectral bias has been a long-standing challenge in continuously representing images, surfaces, and neural fields [R1,R2]. This is also important for numerical PDEs using PINNs, where accurately capturing high-frequency details is essential, and thus developing methods to accelerate convergence for high-frequency components would be appreciated [R3]. Therefore, we believe that our theory and the corresponding algorithm have the potential to benefit these fields and address related challenges.
>
> 2.	**Potential Applications.** Moreover, SepNNs are also increasingly being adopted across diverse scientific domains by leveraging their efficient structure and interpretability through structural constraints (such as factor modeling that decomposes complex representations into compact, interpretable factor components). Representative applications include separable neural operators [R4], medical image analysis using SepNNs [R5], wireless communications (e.g., building 4D radio maps with SepNNs [R6]), urban computing (e.g., urban time series modeling using SepNNs [R7]), Earth sciences (e.g., full waveform inversion using SepNNs [R8]), and bioinformatics (e.g., transcriptomics analysis using SepNNs [R9]). Given this broad relevance, understanding the theoretical approximation ability, training behavior (e.g., NTK regime), and addressing the potential optimization challenge of SepNNs are believed to be valuable and important for these practical applications that rely on SepNNs.
>
> 3.	**Near Future Goal.** From a mathematical perspective, the theory may inspire several promising future research directions. First, SepNNs are inherently related to decomposition-based algebraic structures such as nonnegative matrix factorization (NMF) and tensor decomposition. Therefore, our NTK analysis for SepNNs provides a theoretical tool to bridge neural network optimization with classical NMF, tensor decomposition, and related scientific computing topics. This connection may inspire extending matrix and tensor algebra into the functional analysis setting. Especially, we can combine the SepNN NTK theory with the classical tensor functional optimization [R10] to design functional algorithms by utilizing neural networks as powerful learners of mode functions, establishing the corresponding theoretical guarantee (e.g., algorithmic error bounds) for this potential model by using the training dynamic characterized by the SepNN NTK. Furthermore, the NTK theory can be naturally extended to separable operator learning [R4], providing a theoretical tool for learning mappings between function spaces, which is of broad interest in analysis and applied math. Moreover, based on the NTK analysis, another future research line is to study adaptive feature learning regime [R11] induced by SepNNs. This is particularly interesting for SepNNs since SepNNs can learn disentangled representations, making them suited to extract interpretable features from multimodal data.

---

> > ### Author Response · Authors · 2025-11-13
> > **References**
> >
> > We will include these discussions about the potential impact and applications of our work into the “Conclusions and Discussions” Section of the revised manuscript, and all revisions will be highlighted in blue for your convenience. Should you need further information, please let us know. We look forward to hearing from your feedback!
> >
> > [R1] Implicit Neural Representations with Periodic Activation Functions, NeurIPS
> >
> > [R2] Inductive Gradient Adjustment for Spectral Bias in Implicit Neural Representation, ICML
> >
> > [R3] On understanding and overcoming spectral biases of deep neural network learning methods for solving PDEs, Journal of Computational Physics
> >
> > [R4] Separable Operator Networks, TMLR
> >
> > [R5] Patch-based Reconstruction for Unsupervised Dynamic MRI using Learnable Tensor Function with Implicit Neural Representation, arXiv
> >
> > [R6] Constructing 4D Radio Map in LEO Satellite Networks with Limited Samples, INFOCOM
> >
> > [R7] Collaborative Imputation of Urban Time Series through Cross-city Meta-learning, arXiv
> >
> > [R8] Full-Waveform Inversion With Velocity Model Low-Rank Implicit Neural Representation, TGRS
> >
> > [R9] GNTD: reconstructing spatial transcriptomes with graph-guided neural tensor decomposition informed by spatial and functional relations, Nature Communications
> >
> > [R10] Guaranteed Functional Tensor Singular Value Decomposition, JASA
> >
> > [R11] Towards a Statistical Understanding of Neural Networks: Beyond the Neural Tangent Kernel Theories, arXiv

---

> > > ### Comment · Reviewer_A9Bg · 2025-11-21
> > >
> > > Thanks for your clarification. I am satisfied with your response and would like to increase the score accordingly.

---

> > > > ### Author Response · Authors · 2025-11-21
> > > > **Response to Reviewer A9Bg**
> > > >
> > > > We thank the reviewer again for the constructive comments and suggestions!

---

### Official Review · Reviewer_dMQY · 2025-11-01

**Soundness:** 3
**Presentation:** 3
**Contribution:** 3
**Rating:** 6
**Confidence:** 3

**Summary:**

The paper begins by establishing the universal approximation theory for three classes of separable neural networks (SepNNs): the CP type, TT type, and Tucker type. The proofs are built on classical results (Stone-Weierstrass, universal approximation theorems) and well-motivated in the context of separable function spaces. The results extend beyond the bivariate approximation in (Cho et al., 2023) to general D-variate functions.

Further, the paper prove that, under the asymptotic regime of infinite network width and infinite rank, the NTK of a CP type SepNN converges to a deterministic kernel, provide an analogous result to that of a standard MLP. In contrast, under infinite width and fixed rank, the NTK converges to a stochastic kernel defined via Gaussian processes.

Finally, the paper introduces the SepPGD training algorithm that exploits the Kronecker product structure of SepNN NTKs to reduce computational complexity from $\mathcal{O}(n^D)$ to $\mathcal{O}(nD)$.

**Strengths:**

1. To the best of my knowledge, this is the first work to formally establish the universal approximation theorem for SepNNs, including CP, TT, and Tucker variants. Although the result may not be entirely surprising given known approximation capabilities of neural networks, it fills a gap in the theoretical understanding of SepNNs.

2. I find the introduction of the SepPGD algorithm to be a significant contribution to the application of SepNN, even though no theoretica guarantees is given, its efficiency is demonstrated through various numerical experiments.

**Weaknesses:**

1. While the universal approximation theory for SepNNs is novel, the paper does not provide explicit approximation error rates as a function of network size. In particular, it does not show how the approximation error scales with respect to rank or width.
2. The analysis of the NTK is restricted to the CP-type SepNN, whereas analogous results for TT and Tucker types are not provided.

**Questions:**

I have some additional questions:

1. The manuscript briefly mentions that SepNN is closely related to the Kolmogorov-Arnold network (KAN). Since KAN is a timely and widely-discussed topic these days, could the authors elaborate more on this connection? Specifically, can KAN be regarded as a special caseof a SepNN under the definitions used in this paper? If so, how do the function composition structures in KAN (i.e., sums of univariate functions composed with multivariate linear transforms) relate to the separable architectures (e.g., CP, TT, Tucker) defined here?

2. I may have missed this in the manuscript, but how does SepPGD perform with varying eigenvalue modulation functions $g(\lambda)$? Is the choice critical?

---

> ### Author Response · Authors · 2025-11-14
> **Response to Reviewer dMQY**
>
> ***To Reviewer dMQY***
>
> We sincerely thank the reviewer for the valuable and constructive comments! We have carefully incorporated additional discussions and clarifications in response to the feedback, including the discussions on KAN and the sensitivity of modulation functions. Please see the details below.
>
> >**W1: While the universal approximation theory for SepNNs is novel, the paper does not provide explicit approximation error rates as a function of network size. In particular, it does not show how the approximation error scales with respect to rank or width.**
>
> Thank you for raising this point. Indeed, our current theoretical analysis establishes the universal approximation capability of SepNNs and does not yet provide explicit approximation error rates in terms of network rank or width. In future work, we can consider function spaces spanned by Fourier bases and leveraging the decay properties of Fourier coefficients [R1] to derive approximation error bounds with respect to the rank. This could be achieved by formulating the Fourier expansion in the SepNN form [R2]. Moreover, we can try to characterize the approximation ability of SepNNs w.r.t. width by borrowing approximation theory for narrow neural networks [R3] in future research.
>
> >**W2: The analysis of the NTK is restricted to the CP-type SepNN, whereas analogous results for TT and Tucker types are not provided.**
>
> Thank you for raising this point. While this work considers the CP-type SepNN (which is the most classical SepNN type) to perform the NTK analysis, we believe the NTK analysis can be extended to TT- and Tucker-type SepNNs in a similar fashion. For TT-type SepNNs, the derivation would closely follow that of CP, relying on tensor products of NTKs corresponding to factor MLPs. For Tucker-type SepNNs, analogous NTK results are also expected to hold, provided the asymptotic behavior and boundedness of the core tensor $\mathcal{C}$ are appropriately treated. We regard this extension as a valuable direction for future research. Thanks.
>
> >**Q1: The manuscript briefly mentions that SepNN is closely related to the Kolmogorov-Arnold network (KAN). Since KAN is a timely and widely-discussed topic these days, could the authors elaborate more on this connection? Specifically, can KAN be regarded as a special case of a SepNN under the definitions used in this paper? If so, how do the function composition structures in KAN (i.e., sums of univariate functions composed with multivariate linear transforms) relate to the separable architectures (e.g., CP, TT, Tucker) defined here?**
>
> Thanks for this constructive comment. Yes, KAN is closely related to SepNN. Especially, each layer of KAN can be viewed as a type of SepNN. Specifically, consider the input vector ${x}=[x_{(1)},x_{(2)},\cdots,x_{(n)}]^T$. A single KAN layer [R4] (take single-output as an example), denoted as $\Phi(\cdot):\mathbb{R}^n\rightarrow\mathbb{R}$, is defined as
>
> $$
> \Phi({x}) = \sum_{j=1}^n \phi_{1,j}(x_{(j)}),
> $$
>
> where each univariate function $\phi_{1,j}(\cdot):\mathbb{R}\rightarrow\mathbb{R}$ is parameterized as
>
> $$
> \phi_{1,j}(x_{(j)}) = \omega^{1,j} \left[ b(x_{(j)}) + \sum_{k=1}^K c_k^{1,j} B_k(x_{(j)}) \right].
> $$
>
> Here, $\{c_k^{1,j}\}$ for $k=1,\dots,K$ are trainable parameters, and $\{B_k(\cdot)\}$ are spline basis functions as used in [R4].
>
> We observe that the output $\Phi({x})$ is a linear combination (specifically, summation) of learnable univariate functions $\phi_{1,j}(x_{(j)})$, which shares a similar spirit with SepNNs. Therefore, each KAN layer can be viewed as a SepNN, though it does not directly correspond to the CP, TT, or Tucker types. The distinction lies in how the linear combination is realized. In KAN, it is implemented via simple addition of univariate functions, while in the SepNNs discussed in our manuscript, the combination is realized via tensor product (e.g., CP, TT, or Tucker).
>
> The most significant commonality between KAN and SepNN is that both architectures learn univariate functions and compose multivariate functions through linear combinations of these univariate components. This characteristic fundamentally differentiates KAN from conventional MLPs, and similarly distinguishes SepNNs from their non-separable counterparts. We will include these discussions related to KAN in the revised manuscript. Thanks.

---

> > ### Author Response · Authors · 2025-11-14
> > **Response to Reviewer dMQY**
> >
> > >**Q2: I may have missed this in the manuscript, but how does SepPGD perform with varying eigenvalue modulation functions $g(\lambda)$? Is the choice critical?**
> >
> > As suggested, we conducted a sensitivity analysis on the modulation function $ g(\lambda) $. Specifically, we compared two forms of $ g(\lambda) $: $ g_1(\lambda_i) = \lambda_k $ for $ i \leq k $, and $ g_1(\lambda_i) = \lambda_i $ for $ i > k $ (the version used in the paper); $ g_2(\lambda_i) = \sqrt{\lambda_i} $ for $ i \leq k $, and $ g_2(\lambda_i) = \lambda_i $ for $ i > k $. We also varied the number of modulated eigenvalues $ k $ to assess sensitivity. The results in the table below demonstrate that SepPGD remains effective under both modulation functions, and $ g_1(\lambda) $ yields better performance. Moreover, the method shows robustness to the choice of $ k $ within a certain range (e.g., $ k \in (60, 90) $). These findings indicate that with a suitable modulation function that smooths the eigenvalue distribution, the proposed SepPGD method is not overly sensitive to the specific form of $ g(\lambda) $.
> >
> > Table. PSNR performance on image representation using INRs with different modulation functions.
> >
> > | $ k $       | 30   | 40   | 50   | 60   | 70   | 80   | 90   | 100  |
> > |-|-|---|------|------|------|------|------|------|
> > | w/o SepPGD  | 26.48| 26.48| 26.48| 26.48| 26.48| 26.48| 26.48| 26.48|
> > | $ g_1(\lambda) $ | 26.79| 26.95| 29.65| 33.04| 33.30| 34.28| 34.21| 34.06|
> > | $ g_2(\lambda) $ | 26.92| 27.70| 27.91| 30.81| 31.58| 31.74| 31.95| 30.23|
> >
> > We will include the additional discussions in the revised manuscript, and all revisions will be highlighted in blue for your convenience. Should you need further information, please let us know. We look forward to hearing from your feedback!
> >
> > [R1] Russell L. Herman, Generalized Fourier Series and Function Spaces, Chapman and Hall/CRC
> >
> > [R2] Nikos Kargas, Nicholas D. Sidiropoulos, Supervised Learning and Canonical Decomposition of Multivariate Functions, TSP
> >
> > [R3] Patrick Kidger, Terry Lyons, Universal Approximation with Deep Narrow Networks, COLT
> >
> > [R4] Liu et al., KAN: Kolmogorov–Arnold Networks, ICLR

---

### Author Response · Authors · 2025-11-19
**We have uploaded a revised manuscript**

Dear reviewers,

In response to your valuable suggestions, we have revised the manuscript accordingly and provided additional clarifications in the **uploaded revised manuscript**. These include further clarifications of the potential applications of our work, motivations of SepNNs, more complete definitions of notations, and additional computational complexity analysis.

Due to space constraints, we have also included extended discussions in Section A.1 of the manuscript Appendix, including more discussions on the efficiency advantages of SepNNs, discussions on theoretical analyses and future directions, relation to KAN, hyperparameter sensitivity analysis, and experiments on non-grid inputs.

For easy reference, all revisions have been highlighted in blue in the revised manuscript. Should you need further information, please let us know. We sincerely appreciate your time and valuable efforts!

---

### Author Response · Authors · 2025-11-29
**Brief summary of the discussion phase**

Dear AC and Reviewers,

We sincerely appreciate the time and effort you have dedicated to reviewing our manuscript. We fully understand the additional workload and challenges brought about by the current situation, and we truly appreciate your continued efforts.

In the hope of facilitating your decision-making process, we would like to provide a brief summary of the discussion phase:

**Reviewer dMQY** raised questions regarding the theoretical analysis, relations to the Kolmogorov-Arnold network, and performance with different eigenvalue modulation functions. We addressed these questions in our response, after which the reviewer maintained a positive score of **6**.

**Reviewer A9Bg** raised questions regarding the potential applications of our work. Following our clarifications, the reviewer explicitly confirmed that: *“I am satisfied with your response and would like to increase the score accordingly”*. The score was updated from 4 to **6** prior to the reverting process.

**Reviewer 3E8r** expressed satisfaction with our work, raised no further questions, and maintained a positive score of **8** throughout the process.

**Reviewer WM7X** raised questions regarding the motivation of SepNNs, the theoretical analysis, the computational and memory efficiency, and the completeness of experiments such as ablation studies and non-grid validations. These points were addressed in our response, after which the reviewer maintained a positive score of **6**.

As a result, the final scores before the reverting process were **6 / 6 / 8 / 6**. All revisions and additional clarifications provided during the discussion have been carefully incorporated into the updated manuscript, with changes highlighted in blue for your convenience. We hope this summary could assist your re-evaluation process.

Additionally, we would like to affirm that the discussion of our paper was conducted in a fully legitimate, scientific, and anonymous manner. The authors did not engage in any form of non-scientific activity, deanonymization, or collusion throughout the process.

---

### Meta-Review · Area_Chair_XWf7 · 2026-01-06

**Summary:**

The paper aims to study the architecture design from a theoretical standpoint. The architecture studies relates with the CP and Tucker tensor decompositions, which are standard decompositions. It introduces a theoretical framework for Separable Neural Networks (SepNNs), establishing their universal approximation capabilities and characterizing their Neural Tangent Kernel (NTK) regimes. It proposes a learning gradient-based algorithm that exploits the Kronecker product structure of the NTK to mitigate spectral bias, with significantly lower complexity than standard preconditioning methods.

**Reviewer Concerns:**

The reviewers raise few concerns, including the reasoning behind the architecture, and the scope of the NTK analysis. More concretely, the reviewers raise the following concerns:

1. lack of explicit approximation error rates
2. the results for the tucker analysis are not fully provided
3. the fact that NTK results have been achieved for many types of networks, including for convolutional networks, graph networks, polynomial networks, etc, so the reviewer mentions this analysis is "routine".
4. Among the major concerns of reviewers is that "the motivation for SepNNs is not well positioned against established efficient architectures".
5. Some experimental concerns on reproducibility.

**Reviewer Scores:**

I believe out of the concerns raised above, some of them might be addressed already by the rebuttal. Concretely, the rebuttal offers new experiments and elaborates on the experimental setting, so I do think some of those concerns would have been raised. It seems that the responses of the reviewers were also positive after the rebuttal. Lastly, some of the concerns, such as the explicit approximation rates, are challenging and might be a component of a different paper.

---

### Decision · Program_Chairs · 2026-01-26

Accept (Poster)